# Bacteria loaded with glucose polymer and photosensitive ICG silicon-nanoparticles for glioblastoma photothermal immunotherapy

Rong Sun[1,2], Mingzhu Liu[1,2], Jianping Lu[1,2], Binbin Chu[1], Yunmin Yang[1], Bin Song[1], Houyu Wang [1] ✉ & Yao He [1] ✉

Bacteria can bypass the blood-brain barrier (BBB), suggesting the possibility of employment of bacteria for combating central nervous system diseases. Herein, we develop a bacteria-based drug delivery system for glioblastoma (GBM) photothermal immunotherapy. The system, which we name as 'Trojan bacteria', consists of bacteria loaded with glucose polymer and photosensitive ICG silicon-nanoparticles. In an orthotopic GBM mouse model, we demonstrate that the intravenously injected bacteria bypass the BBB, targeting and penetrating GBM tissues. Upon 808 nm-laser irradiation, the photothermal effects produced by ICG allow the destruction of bacterial cells and the adjacent tumour cells. Furthermore, the bacterial debris as well as the tumour-associated antigens promote antitumor immune responses that prolong the survival of GBM-bearing mice. Moreover, we demonstrate the residual bacteria are effectively eliminated from the body, supporting the potential therapeutic use of this system.

Central nervous system (CNS) tumors are still a significant cause of morbidity and mortality throughout the world. For example, glioblastoma (GBM) is the most aggressive brain tumor, with more than 15,000 deaths each year in the United States alone, and a 5-year survival rate of less than 10%[1–4]. One key challenge in the therapy of GBM is to develop a high-efficiency drug delivery system (DDS) bypassing the blood–brain barrier (BBB), which is impermeable to most drugs[5–7]. For example, the levels of antibodies in the brain are only 0.01–0.1% of those in plasma after parenteral administration[8, 9]. To deliver drugs into brain, invasive and non-invasive technologies have been proposed[10–14]. The invasive approaches include deep brain stimulation, intracerebral grafts, direct brain injection, intrathecal brain delivery, and so forth. Due to the huge risks and pains caused by the invasive approaches, people are turning more and more attention to the non-invasive approaches, which include receptor-mediated transcytosis, the use of neurotropic viruses, exosomes, nanoparticles and so on[15–17]. Among these non-invasive strategies, the nanoparticulate drug delivery systems have shown promising clinical value in GBM therapy due

to their appealing properties such as high drug loading efficiency, spatial- and temporal-controlled drug release, real-time visualization during the therapeutic process and so forth[18–26]. Despite these elegant works, there are still no clinically approved nanoparticle-based therapies against GBM owing to the following hurdles: (1) the intrinsic properties of nanoparticles such as size, surface charge, and opsonization can influence uptake of them into phagocytes, thereby preventing their entry into the brain; (2) the premature release of payload might occur during systemic circulation since therapeutic agents are generally loaded on nanoparticles through charge interaction or hydrophobic interaction; (3) the nanoparticles even crossing BBB still hardly penetrate deeply into GBM tissues owing to the high interstitial fluid pressure of GBM tissues, thus hampering their therapeutic effects[5, 15, 27, 28].

On the other aspect, bacteria recently have achieved encouraging outcomes in cancer therapy[29–33]. Conceptually, bacteria for cancer treatment are independent of 'genetic makeup', featuring superior merits over conventional treatments including intrinsic tumor

[1]Suzhou Key Laboratory of Nanotechnology and Biomedicine, Institute of Functional Nano & Soft Materials & Collaborative Innovation Center of Suzhou Nano Science and Technology (NANO-CIC), Soochow University, Suzhou 215123, China. [2]These authors contributed equally: Rong Sun, Mingzhu Liu, Jianping Lu. ✉e-mail: houyuwang@suda.edu.cn; yaohe@suda.edu.cn

navigation ability, tumor tissue-penetration ability as well as gene packaging ability. Remarkably, it has been well demonstrated that bacteria can cross the BBB transcellularly, paracellularly, and/or in infected phagocytes[34–36], which is the basis of bacterial systems for the treatment of GBM. However, GBM therapies with live bacteria are still few, since they encounter numerous challenges, such as difficulty in precise control of drug release, inadequate stimulation to immune responses, and potential bacterial toxicity (e.g., bacteremia)[35–37]. Intriguingly, there is increasing enthusiasm for developing bacteria-nanoparticles hybrid systems for the drug delivery against other types of cancer[38, 39]. However, in these systems, the nanoparticles are generally loaded on the surface of bacteria, and thus they still suffer from the afore-mentioned drawbacks often associated with free nanoparticles. Moreover, the surface-loaded nanoparticles might break the integrity of the capsule of bacteria, and the intact capsule can prevent fusion of bacteria with lysosomes, which is necessary for traversal of the BBB as live bacteria[34, 40]. Taken together, bacteria-nanoparticles hybrid system across BBB for fighting GBM has not yet been achieved.

To this end, we herein develop a Trojan bacteria system to delivery therapeutics into brain for photothermal immunotherapy of GBM. The therapeutics are composed of glucose polymer (GP)-conjugated and indocyanine green (ICG)-loaded silicon nanoparticles (GP-ICG-SiNPs) (Fig. 1a). The GP-ICG-SiNPs can be selectively and robustly internalized by the facultatively anaerobic bacteria (e.g., attenuated *Salmonella typhimurium* VNP20009 (VNP), *Escherichia coli* 25922 (EC)) through the bacteria-specific ATP-binding cassette (ABC) transporter to form the Trojan bacteria system[41–46] (Fig. 1b). Compared with free therapeutics hardly entering the brain and penetrating GBM tissue, the constructed Trojan bacteria can take therapeutics together across the BBB, target GBM and then penetrate the GBM tissue more deeply. Under 808-nm irradiation, the ICG molecules loaded on SiNPs could convert light energy into sufficient heat to destruct tumor cells, promoting the release of tumor-associated antigens (TAAs)[47–51]. Meanwhile, the produced heat could lysis host bacterial cells, promoting the release of diverse pathogen-associated molecular patterns (PAMPs). The PAMPs could promote the activation of innate immune cells like macrophages and natural killer (NK) cells, leading to innate antitumour

immunity. On the other aspect, the as-resultant PAMPs could promote tumor-infiltrating frequencies of activated CD8$^+$ T cells, eliciting potent adaptive antitumor immunity[47–51]. We demonstrate that the Trojan bacteria exhibit better therapeutic effects toward GBM compared with the pristine bacteria. We anticipate the proposed Trojan bacteria system will catalyze innovative therapies for various CNS diseases, and GBM treatment in the work only serves as an initial proof of principle.

## Results

### Design of Trojan bacteria system

As schematically illustrated in the synthetic route of GP-ICG-SiNPs (Supplementary Fig. 1), the GP molecules are firstly conjugated to the SiNPs surface based on the Schiff base reaction between the aldehyde groups of GP and the amino SiNPs[41]. A series of experiments were carried out to demonstrate the successful preparation of GP-ICG-SiNPs including transmission electron microscopy (TEM), dynamic light scattering (DLS), UV-vis absorbance (UV), photoluminescence (PL) and flow cytometry. As shown in the TEM image in Supplementary Fig. 2a, b, GP-ICG-SiNPs appear as spherical particles with a narrow size distribution of ~4.1 nm, which is slightly larger than that of bare SiNPs (e.g., ~2.7 nm).The DLS spectra in Supplementary Fig. 2c reveal the hydrodynamic diameter of GP-ICG-SiNPs is ~6.5 nm, also larger than the hydrodynamic diameter of bare SiNPs (~3.6 nm). As shown in Supplementary Fig. 2d, three distinct peaks located at 320 nm (assigned to SiNPs), 731 nm, and 790 nm (assigned to ICG) exist in the absorption spectrum of GP-ICG-SiNPs, confirming the successful loading of ICG molecules. As further revealed in Supplementary Fig. 2e, upon the treatment of phenol-sulfuric acid, the absorption spectrum of GP-ICG-SiNPs displays a new peak at 490 nm since the linked GP is hydrolyzed into furfural derivative, followed by the formation of furfural resin with phenol[52]. The new appearing peak confirms the successful conjugation of GP molecules. The amounts of linked GP and loaded ICG onto SiNPs can be quantified based on the corresponding calibration absorption curves (Supplementary Fig. 2f–i), respectively. Supplementary Fig. 2j, k give the PL spectra of GP-ICG-SiNPs under the excitation of 405 nm or 780 nm, respectively. Typically, the two characteristic emission peaks

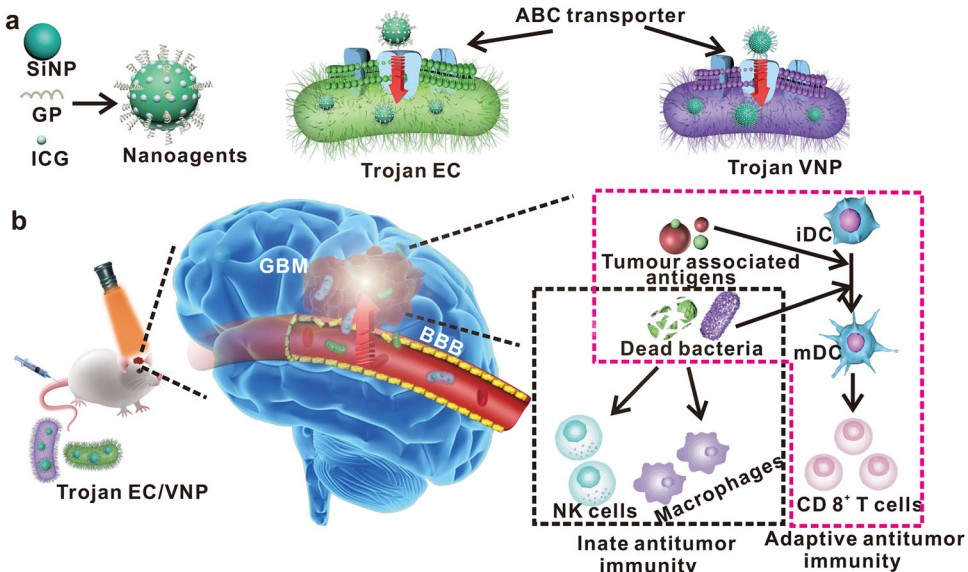

**Fig. 1 | Schematic illustration of Trojan bacteria crossing blood–brain barrier for photothermal immunotherapy of glioblastoma. a** A scheme illustrating the construction of Trojan bacteria system. The nanoagents composed of silicon nanoparticle (SiNP) modified with glucose polymer (GP) (e.g, *poly[4-O-(α-D-glucopyranosyl)-D-glucopyranose]*) and indocyanine green (ICG) were internalized into bacterial cells (e.g., attenuated *Salmonella typhimurium* VNP20009 (VNP), *Escherichia coli* 25922 (EC)) through the bacteria-specific ATP-binding cassette (ABC) transporter to form the Trojan bacteria system (Trojan EC/VNP). **b** A scheme illustrating Trojan bacteria system crossing the blood–brain barrier (BBB), targeting and penetrating glioblastoma (GBM) tissues, followed by light-triggered photothermal immunotherapy of GBM in vivo. The cartoons are created by Dr. Houyu Wang.

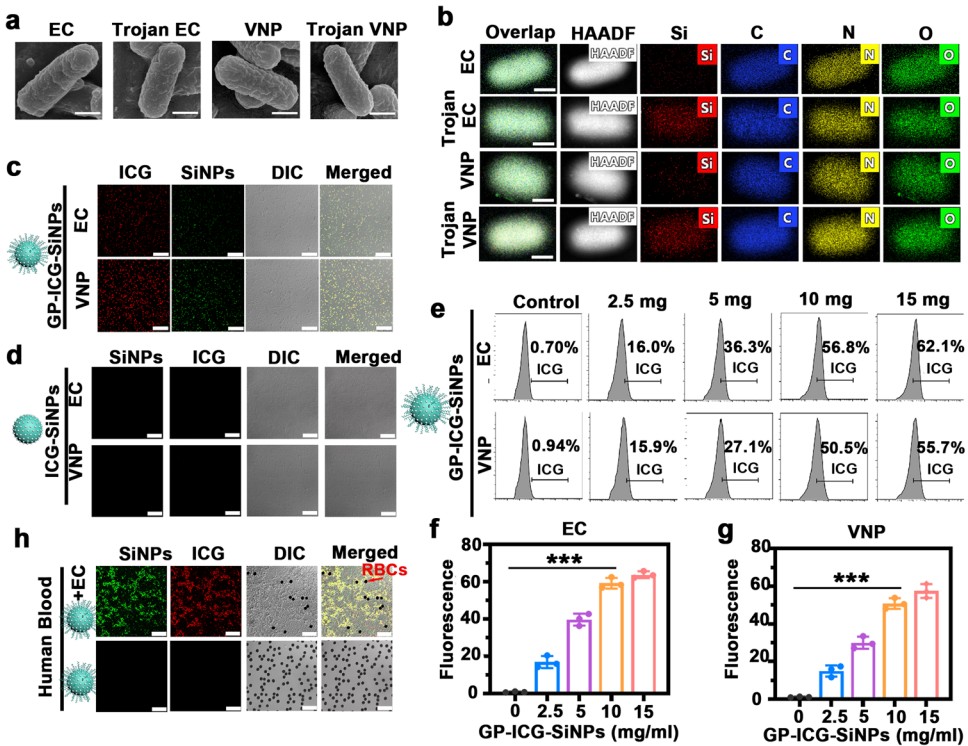

**Fig. 2 | Construction and characterization of Trojan bacteria system. a** SEM images of EC, Trojan EC, VNP, and Trojan VNP. Scale bars: 200 nm. **b** Elemental mapping in HAADF-STEM images of EC, Trojan EC, VNP, and Trojan VNP. Scale bars: 500 nm. **c** CLSM images of EC and VNP incubated with GP-ICG-SiNPs. Scale bars: 25 μm. **d** CLSM images of EC and VNP incubated with ICG-SiNPs. Scale bars: 25 μm. All imaging experiments were repeated three times with similar results. **e**–**g** Flow cytometry analysis of the uptake rates of EC and VNP incubated with different concentrations of GP-ICG-SiNPs for 2 h (**e**) and corresponding quantitative analysis of uptake rates of different concentrations of GP-ICG-SiNPs by EC (**f**) and VNP (**g**). ***$p = 4.80 \times 10^{-6}$ (**f**), ***$p = 7.46 \times 10^{-6}$ (**g**). **h** CLSM images of the mixture of human blood and EC after incubation with GP-ICG-SiNPs. Arrows indicate red blood cells (RBCs). Scale bars: 25 μm. All imaging experiments were repeated three times with similar results. The EC or VNP were incubated with the synthesized nanoagents ([SiNPs] = 12 mg/mL, [ICG] = 600 μg/mL) at 37 °C for 2 h. After incubation, the treated bacteria were rinsed with PBS buffer for several times. The bacterial cell concentration is $\sim 1.0 \times 10^7$ CFU. All error bars represent the standard deviation determined from three independent assays. Statistical significance is calculated *via* one-way analysis of variance (ANOVA) with a Tukey post hoc test. Data are presented as mean values +/− SD ($n = 3$). Source data are provided as a Source data file.

located at 485 nm and 810 nm are, respectively, corresponding to SiNPs and ICG. The photothermal curves in Supplementary Fig. 2l suggest that the temperature of GP-ICG-SiNPs solutions reaches 64.0 °C during a 300-s 808-nm laser exposure when the loading concentration of ICG is or more than 150 μg/mL. ICG molecules tend to aggregate at high concentration, which may lead to the degradation and self-quenching of ICG, possibly resulting in unchanged temperature with increasing the ICG concentration. These results together demonstrate the successful modification of ICG molecules with SiNPs.

To test the generality of the proposed Trojan bacteria strategy, two representative bacteria of VNP and EC were selected for the following experiments. The VNP or EC were incubated with the synthesized GP-ICG-SiNPs at 37 °C for 2 h, and then washed with phosphate-buffered saline (PBS) buffer for several times. As revealed in the scanning electron microscope (SEM) images in Fig. 2a, the surface and the morphology of Trojan bacteria are similar to that of pristine bacteria. As further confirmed by elemental mapping in high-angle annular dark field-scanning transmission electron microscope (HAADF-STEM) images (Fig. 2b), the silicon element exists only in Trojan EC or Trojan VNP rather than in pure EC or VNP. SEM and TEM images together prove that the prepared therapeutics indeed enter bacteria rather than nonspecifically absorb on the bacterial surface. Moreover, Trojan bacteria have the same growth curve compared with the untreated bacteria (Supplementary Fig. 3a), and their survival rate could maintain as high as 90% or more (Supplementary Fig. 3b), indicating the growth and activity of host bacteria would not be greatly influenced by the internalized therapeutics.

As shown in the confocal laser scanning microscope (CLSM) images in Fig. 2c, the green fluorescence from SiNPs (first column, $\lambda_{ex} = 405$ nm, $\lambda_{em} = 500$–550 nm) and the red fluorescence from ICG (second row, $\lambda_{ex} = 633$ nm, $\lambda_{em} = 700$–800 nm) could be simultaneously observed in the Trojan bacteria. On the contrary, no fluorescent signals are detected when EC or VNP are treated with ICG-SiNPs under the same conditions (Fig. 2d). Quantitatively, the uptake efficiency of nanoagents by EC or VNP cells was further determined by flow cytometry. As revealed in Fig. 2e, the uptake efficiency of GP-ICG-SiNPs by EC or VNP after 2 h of incubation is gradually rising when increasing the concentration of GP-ICG-SiNPs. Typically, when the GP-ICG-SiNPs is 10 mg/mL, the uptake efficiency is 56.8% for EC and 50.5% for VNP. If further enhancing the concentration to 15 mg/mL, the uptake efficiency does not improve significantly, e.g., 62.1% for EC and 55.7% for VNP. It suggests that the saturated state of the uptake of GP-ICG-SiNPs by bacteria has achieved when the concentration of GP-ICG-SiNPs is 10 mg/mL. As such, 10 mg/mL GP-ICG-SiNPs is employed in the following experiments. The fluorescence of Trojan bacteria would not decrease when the concentration of the incubated GP-ICG-SiNPs is above 15 mg/mL (Fig. 2f, g). Analogously, the near-infrared (NIR)-induced thermolysis of Trojan bacteria would not interfere when the saturated state has been achieved.

## Trojan bacteria are ABC transporter pathway-dependent
We further constructed the bacterial mutants including a deletion mutant for delta-lamB (ΔlamB) and a deletion mutant for delta-malE (ΔmalE) to confirm GP-ICG-SiNPs entering bacteria was through the

ABC transporter pathway. First, the results of Sanger sequencing (Supplementary Notes) demonstrated the successful construction of ΔlamB and ΔmalE. The constructed ΔlamB or ΔmalE were incubated with GP-ICG-SiNPs. As shown in Supplementary Fig. 4, we did not observe any fluorescence in GP-ICG-SiNPs treated bacteria mutants. We also performed the competition assay to test the mechanism, in which EC or VNP were first incubated with 0, 5, and 20 mg/mL of GP and then incubated with GP-ICG-SiNPs. As indicated in Supplementary Fig. 5, the fluorescence of bacteria becomes gradually weak with the increase of GP concentrations. To testify the selectivity of GP-ICG-SiNPs for bacteria over mammalian cells, the mixture of human blood and bacteria were incubated with GP-ICG-SiNPs for 2 h. As shown in Fig. 2h and Supplementary Fig. 6, fluorescence signals are only observed in EC or VNP rather than in red blood cells (RBCs). These results together demonstrate that GP-ICG-SiNPs can be internalized into bacteria to form the Trojan system via the bacteria-specific ABC transporter pathway.

## Trojan bacteria against tumor in vitro

We next studied the photothermal ability of Trojan bacteria. Under 808-nm laser irradiation, the constructed Trojan bacteria (Trojan EC or Trojan VNP) could heat up to 55 °C within 400 s, slightly lower than the temperature achieved by the equivalent free GP-ICG-SiNPs (Fig. 3a). To obtain the equivalent amount of NPs, the detected absorbance of ICG should be kept the same between free GP-ICG-SiNPs (containing 8 mg/kg ICG) and Trojan bacteria (GP-ICG-SiNPs (containing 8 mg/kg ICG) internalized into -1 × 10^7 CFU bacteria). Accordingly, the bacterial cell viability gradually decreases as the temperature increases from 48 to 55 °C (Fig. 3b). In particular, the bacterial cell viability is only 37% at 52 °C. Meanwhile, the overall morphology of the Trojan bacteria begins to rupture when the temperature rises to 48 °C, as shown in the SEM image in Fig. 3c. These results manifest that the constructed Trojan bacteria feature good photothermal activity.

Next, we investigated whether the photothermal effects produced by the Trojan bacteria could destroy glioblastoma G422 cells. In live-dead cell staining, the red fluorescence is only observed in most of Trojan bacteria-treated G422 cells (Fig. 3d). In MTT assay, under the irradiation of 808 nm for 5 min, the cell viability of G422 cells dramatically decreases to less than -20% when they are incubated with Trojan bacteria or equivalent free GP-ICG-SiNPs for 6 h, much lower than the 100% of other control groups ($p < 0.001$) (Fig. 3e). On the contrary, the cell viability of G422 cells treated with EC under laser irradiation (EC + laser) for 5 min maintains at 93%, and treated with VNP under laser irradiation (VNP + laser) for 5 min maintains at 89%. In addition, alternative glioblastoma lines (e.g., GL261) and non-GBM tumor cells (e.g., HeLa, 4T1) have been employed to assess the general efficiency of the Trojan bacteria system in vitro (Supplementary Fig. 7).

In the constructed Trojan bacteria system, tumor-associated antigens and bacterial residues produced by PTT might trigger an effective immune response. To testify this hypothesis, we designed a transwell system to study this effect in vitro. As shown in Fig. 3f, G422 cells with different treatments are placed in the upper chamber of the transwell system, and DCs collected from the bone marrow of female Balb/c mice about 6−8 weeks old are seeded in the lower chamber. After that, the maturation of DCs (CD11c^+, MHC II^+) are evaluated by flow cytometry[53, 54]. The details of antibody panel for spectral flow cytometry analyses were listed in Supplementary Table 1. As revealed in Supplementary Fig. 8 and Fig. 3g, compared with the PBS group, the level of DCs maturation can be significantly improved when G422 cells were treated with Trojan bacteria system under 808-nm irradiation (e.g., 53.3% DC maturation in Trojan EC + laser group, 58.0% DC maturation in Trojan VNP + laser group). Moreover, the synergy coefficient was calculated to be -0.75 for the combination of Trojan EC and

laser; and -0.78 for the combination of Trojan VNP and laser, indicating this combination exhibited synergy in the maturation of DCs.

## In vivo behaviors of bacteria in mice

Before cancer treatment, we have systematically studied the behavior of bacteria in mice after tail vein injection. We first determined the safe dose of bacteria injected into the mice. The body weights were measured for healthy mice injected with EC or VNP at -1 × 10^6, -1 × 10^7, and -1 × 10^8 CFU (Fig. 4a). Specially, at the high dose of -1 × 10^8 CFU EC or VNP, the mouse body weights rapidly dropped, and one or two of five mice died, respectively, in the EC or VNP-treated groups, implying the severe toxicity of bacteria at such a high dose. Under lower doses, i.e., -1 × 10^6 and -1 × 10^7 CFU per mouse, the mouse body weights did not change significantly, and no mice died in the corresponding group. Therefore, EC or VNP at a moderate dose (-1 × 10^7 CFU per mouse) with a tolerable side effect were employed in the following experiments.

To reveal the in situ and real-time location of bacteria in vivo, we transformed the pRSETB-mCherry plasmids into EC (mCherry@EC) or VNP (mCherry@VNP) to express red fluorescence protein of mCherry (Supplementary Fig. 9). Afterward, the female health Balb/c mice were injected with these engineered VNP or EC through the tail vein at the dose of -1 × 10^7 CFU per mouse. The mice were then sacrificed at 12, 24, 72, 120, and 360 h after intravenous injection to obtain their main organs (e.g., brain, heart, liver, spleen, lung, and kidney), followed by the detection of red fluorescence of mCherry by an in vivo optical imaging system (IVIS Lumina III). As shown in Fig. 4b−d, the fluorescence signal mainly exists in the liver, and is gradually weakened over time, which is basically undetectable at 15 days. We then homogenized the extracted organs, serially diluted (10-fold), and plated them on solid LB agar plates. In consistent with the results of ex vivo images, bacteria mainly accumulated in the liver and were quickly cleared from all extracted organs. The total elimination was basically achieved at 15 days (Fig. 4e−g).

To further ensure the safety of bacterial injection, routine blood tests including complete blood and serum biochemical analysis were performed on the tested dose, i.e., -1 × 10^7 CFU per mouse. As revealed in Supplementary Tables 2 and 3, all serum biochemical parameters data were within the normal range on the first day after bacterial injection, except for an increase in glutamic-pyruvic transaminase and a decrease in white blood cell, platelet count, alkaline phosphatase and blood urea nitrogen. On the fifth day after bacterial injection, these levels of changed indicators returned to normal ranges. Moreover, we performed routine blood tests on the tumor-bearing mice after 16 days of Trojan bacteria injection (-1 × 10^7 CFU per mouse). As revealed in Supplementary Table 4, the values of all indicators of routine blood tests in Trojan bacteria groups were in normal range. Additionally, we tested the CSF cytokine (e.g., IL-6, IL-10) levels in the tumor-bearing mice after Trojan bacteria injection. As revealed in Supplementary Fig. 10, while IL-6 and IL-10 levels would enhance slightly from the fifth day of treatment, they would return to normal ranges on the 25th day after treatment. Furthermore, the internal temperature of treated mice maintained at -37 °C. Collectively, these results indicated that the acute inflammation caused by Trojan bacterial infection was mild and tolerated by the mice and did not develop chronic toxicity.

## Trojan bacteria crossing BBB, targeting and penetrating glioblastoma

Next, we performed a series of experiments to demonstrate the constructed Trojan bacteria could cross the BBB. We first built an in vitro human brain microvascular endothelial cell (HBMEC) model to investigate whether the Trojan bacteria could cross the BBB (Fig. 5a)[55,56]. The construction of HBMEC model was evaluated by the measurement of transepithelial electrical resistance (TEER) (Supplementary Fig. 11a). Experimentally, Trojan bacteria were inoculated with HBMEC cells in

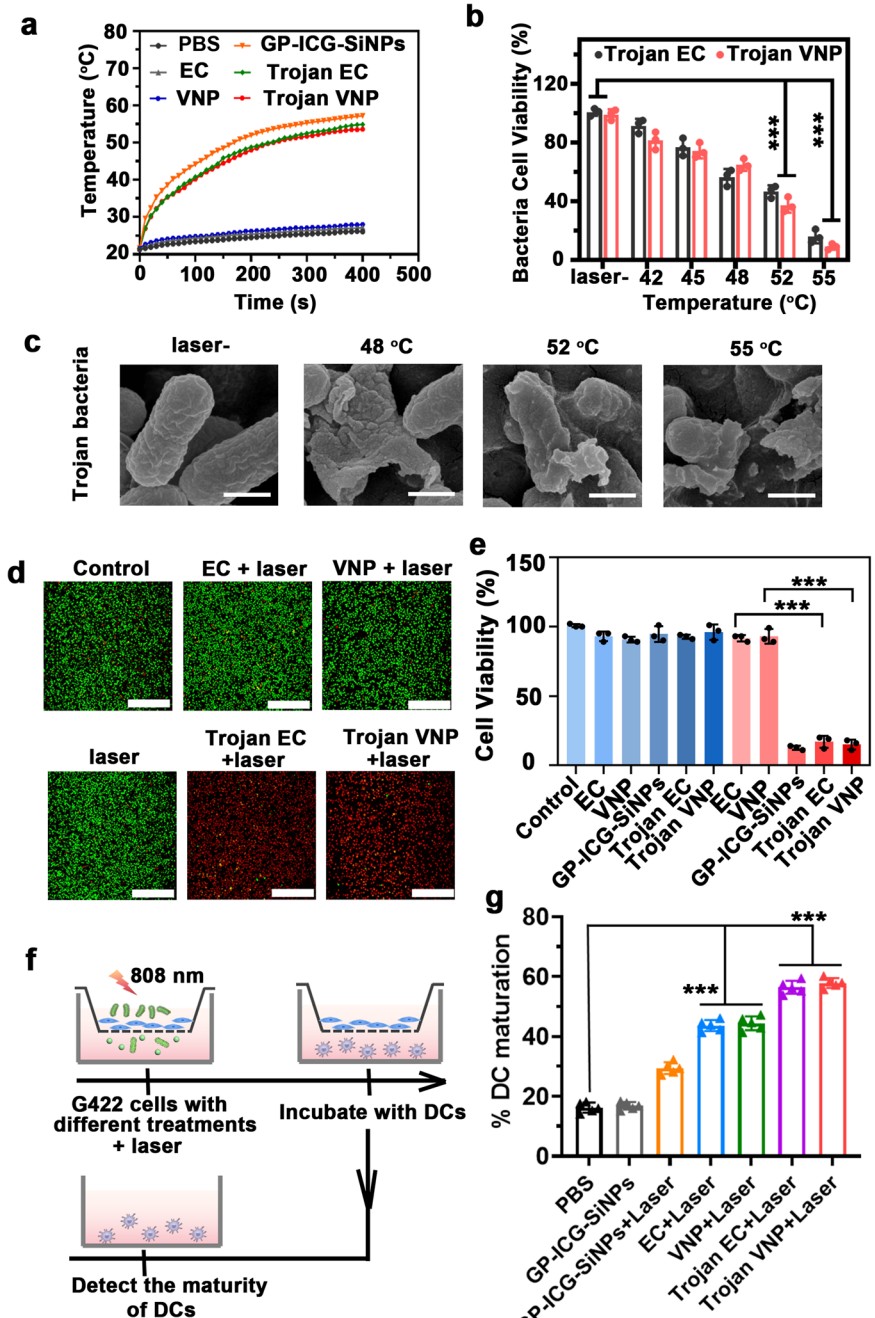

**Fig. 3 | Trojan bacteria system against tumor in vitro. a** Photothermal heating curves of PBS, GP-ICG-SiNPs, EC, VNP, Trojan EC, and Trojan VNP under the irradiation of NIR laser (808 nm, 1.2 W/cm²). **b** The viability of Trojan EC or Trojan at different temperatures (mean ± SD, $n = 3$). ***$p = 6.27 \times 10^{-6}$ for laser- vs. 52 °C, ***$p = 3.90 \times 10^{-7}$ for laser- vs. 55 °C (Trojan EC); ***$p = 2.57 \times 10^{-6}$ for laser- vs. 52 °C, ***$p = 2.29 \times 10^{-7}$ for laser- vs. 55 °C (Trojan VNP). **c** SEM images of Trojan bacteria at different temperatures. Scale bars: 200 nm. All imaging experiments were repeated three times with similar results. **d** Confocal fluorescence images of G422 cells stained with calcein-AM (CAM) and PI after treated with PBS, EC, VNP, GP-ICG-SiNPs, Trojan EC, and Trojan VNP with or without laser irradiation for 5 min (808 nm, 1.2 W/cm²). Scale bars: 500 μm. All imaging experiments were repeated three times with similar results. **e** The viability of G422 cells treated with EC, VNP, GP-ICG-SiNPs, Trojan EC, and Trojan VNP with or without laser irradiation for 5 min (808 nm, 1.2 W/cm²) (mean ± SD, $n = 3$). ***$p = 9.64 \times 10^{-4}$ for EC vs. Trojan EC, ***$p = 2.96 \times 10^{-5}$ for VNP vs. Trojan VNP. **f** Schematic diagram of the transwell system. G422 cells were cultured in the upper chamber and DCs were cultured in the lower chamber. **g** Quantification of the maturation of DCs post different treatments as indicated in the transwell system (mean ± SD, $n = 3$). The bacterial cell concentration is ~10⁷ CFU. ***$p = 8.20 \times 10^{-9}$ for PBS vs. EC + Laser, ***$p = 2.00 \times 10^{-8}$ for PBS vs. VNP + Laser, ***$p = 8.00 \times 10^{-10}$ for PBS vs. Trojan EC + Laser, ***$p = 1.80 \times 10^{-10}$ for PBS vs. Trojan VNP + Laser. Statistical significance was calculated via one-way analysis of variance (ANOVA) with a Tukey post hoc test. Source data are provided as a Source data file.

the upper chamber (apical chamber) of transwell at a dose of ~8 × 10⁴ CFU/well, followed by collecting 10 μL of culture medium from the lower chamber at 1, 2, 3, and 4 h, respectively. As revealed in Supplementary Fig. 11b, the value of TEER of HBMEC BBB model after co-incubating with EC or VNP kept relatively stable, indicating EC or

VNP would not influence the integrity of the BBB. The penetration rate of Trojan bacteria crossing the BBB was determined by counting the number of colonies on the LB solid medium the next day (Supplementary Fig. 11c). As revealed in Fig. 5b, the penetration rate of EC and VNP increases gradually with time, climbing to 49.7% and 60% at 4 h,

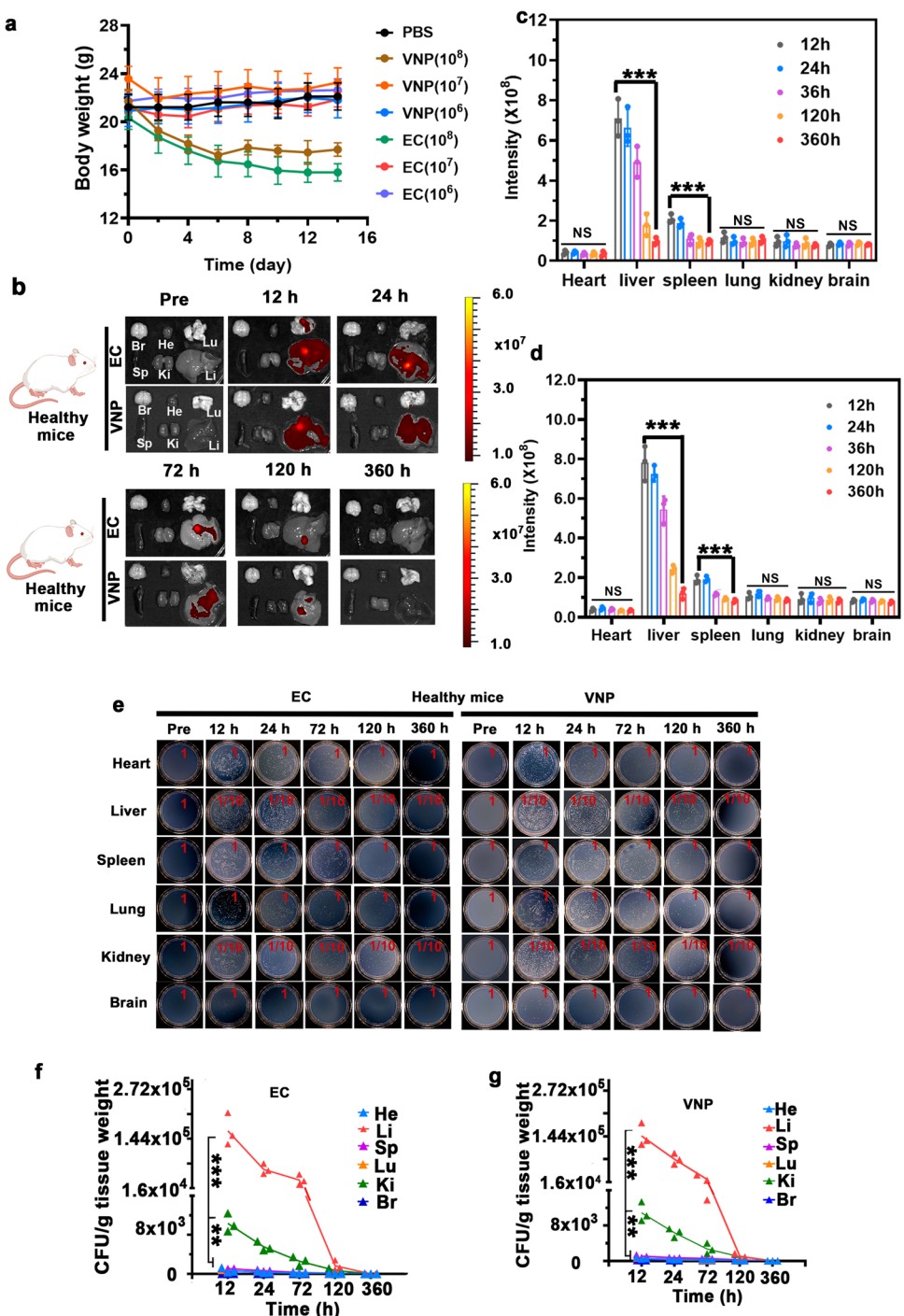

**Fig. 4 | In vivo behaviors of bacteria in mice. a** Average body weights of healthy mice injected with EC or VNP with different concentrations. Data are presented as mean values +/− SD (*n* = 3). **b**–**d** Ex vivo fluorescence images of major organs (heart, liver, spleen, lung, kidney, and brain) of healthy mice after the intravenous injection with mCherry@EC and mCherry@VNP at the dose of -1 × 10⁷ CFU for 12, 24, 72, 120, and 360 h (**b**) and corresponding fluorescence intensity in mCherry@EC group (**c**) and mCherry@VNP group (**d**). ***p = 8.81 × 10⁻⁶ for 360 h vs. 12 h (liver), ***p = 3.28 × 10⁻⁴ for 360 h vs. 12 h (spleen) (**c**). ***p = 2.98 × 10⁻⁷ for 360 h vs. 12 h (liver), ***p = 9.77 × 10⁻⁶ for 360 h vs. 12 h (spleen) (**d**). Data are presented as mean values +/− SD (ns means no significance, *n* = 3). **e**–**g** Homogenates of major organs of healthy mice after intravenous injection with mCherry@EC (left) and mCherry@VNP (right) for 12, 24, 72, 120, and 360 h cultured on the solid LB agar (*n* = 3) (**e**) and corresponding quantification of bacterial colonization in mCherry@EC group (**f**) and mCherry@VNP group (**g**). ***p = 1.60 × 10⁻¹⁰ for liver vs. brain, **p = 9.0 × 10⁻³ for kidney vs. brain (**f**). ***p = 1.60 × 10⁻⁷ for liver vs. brain, **p = 9.30 × 10⁻³ for kidney vs. brain (**g**). Data are presented as mean values +/− SD (*n* = 3). All error bars represent the standard deviation determined from three independent assays. Statistical significance is calculated *via* one-way analysis of variance (ANOVA) with a Tukey post hoc test. Source data are provided as a Source data file.

respectively. On the other aspect, we also constructed an additional in vitro BBB model made of bEnd.3 cells to testify the general feasibility of Trojan bacteria crossing BBB in the in vitro model (Supplementary Fig. 12).

Afterward, the female Balb/c mice with in situ GBM were injected with these mCherry@VNP or mCherry@EC through the tail vein at the dose of -1 × 10⁷ CFU per mouse. As revealed in ex vivo images in Fig. 5c−e, the red fluorescence of bacteria could be found in GBM after

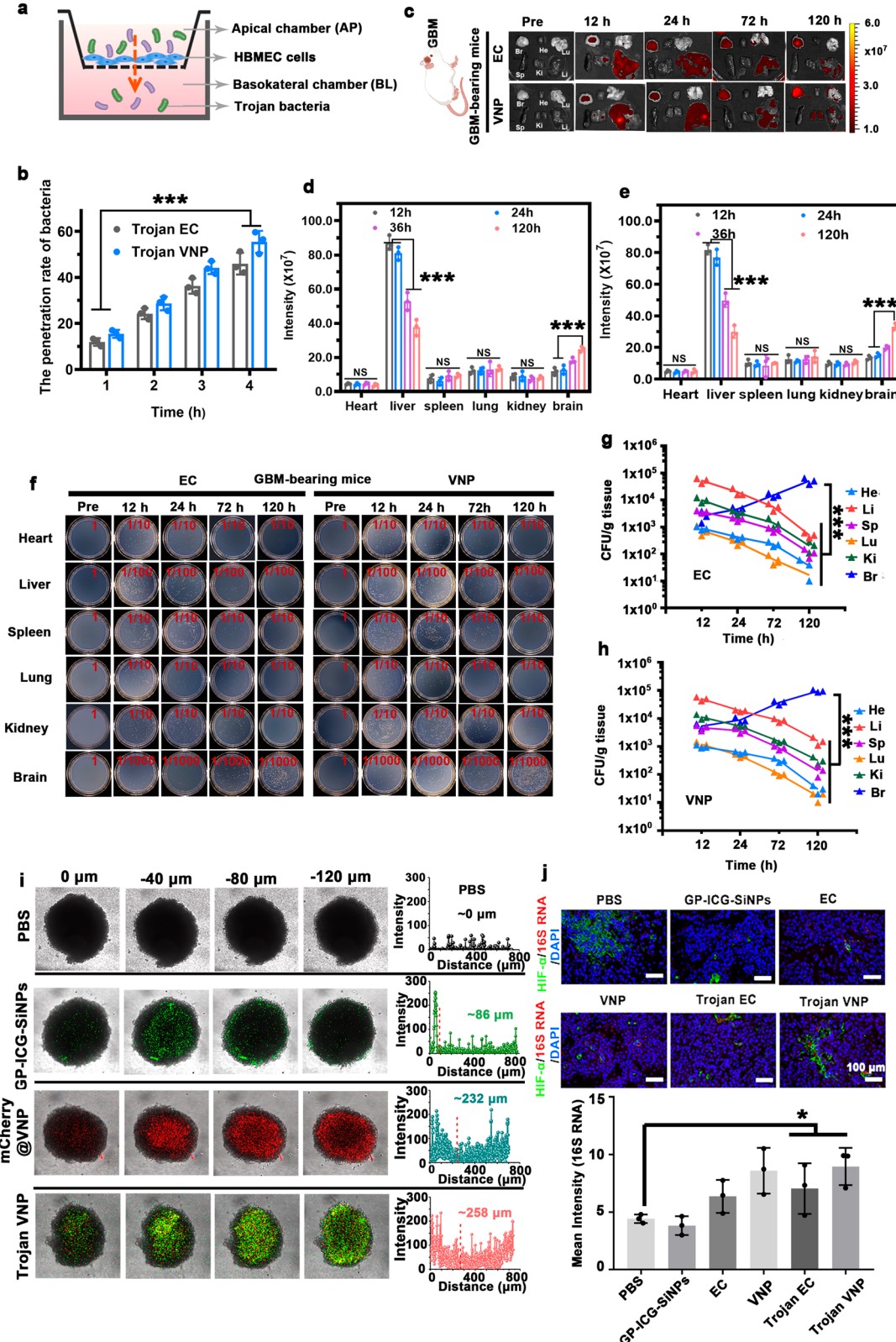

12 h of bacteria injection, gradually increasing following the time. By contrast, the fluorescence from bacteria in other organs like liver decreases exponentially following time. Such difference might be resulted from the selective proliferation of bacteria in the hypoxic, immunosuppressive, and biochemically unique glioblastomas microenvironment[33, 57–59]. And through the results of the plates, the amount of EC or VNP in brain counted in each plate was much higher

than that in other organs. Specifically, the CFU of bacteria in brain reached its peak up to ~$0.51 \times 10^6$ CFU/g for EC and ~$0.74 \times 10^6$ CFU/g for VNP at the 120 h post injection with bacteria (Fig. 5f–h). We also used the agar plate assay to quantitatively study the bacterial distribution in control mice injected with Trojan bacteria. As shown in Supplementary Fig. 13, as expected, Trojan bacteria mainly accumulated in the liver and were quickly cleared from all extracted organs,

**Fig. 5 | Trojan bacteria crossing BBB, targeting and penetrating GBM.**
**a** Schematic diagram of in vitro BBB model for evaluating whether the Trojan bacteria could cross the BBB. **b** The corresponding penetration rates of Trojan EC or Trojan VNP at 1, 2, 3, and 4 h. ***$p = 4.80 \times 10^{-5}$ for 1 h vs. 4 h (Trojan EC), ***$p = 5.34 \times 10^{-5}$ for 1 h vs. 4 h (Trojan VNP). Data are presented as mean values +/− SD ($n = 3$). **c–e** Ex vivo fluorescence images of major organs (heart, liver, spleen, lung, kidney, and brain) of GBM-bearing mice after intravenous injection with mCherry@EC and mCherry@VNP at the dose of ~1 × $10^7$ CFU for 12, 24, 72, and 120 h (**c**) and corresponding fluorescence intensity in mCherry@EC group (**d**) and mCherry@VNP group (**e**). ***$p = 6.63 \times 10^{-5}$ for 36 h vs. 12 h (liver), ***$p = 2.66 \times 10^{-4}$ for 36 h vs. 24 h (liver), ***$p = 4.09 \times 10^{-6}$ for 120 h vs. 12 h (liver), ***$p = 1.12 \times 10^{-5}$ for 120 h vs. 24 h (liver), ***$p = 2.16 \times 10^{-4}$ for 120 h vs. 12 h (brain), ***$p = 3.78 \times 10^{-4}$ for 120 h vs. 24 h (brain) (**d**). ***$p = 5.44 \times 10^{-5}$ for 36 h vs. 12 h (liver), ***$p = 1.84 \times 10^{-4}$ for 36 h vs. 24 h (liver), ***$p = 1.24 \times 10^{-6}$ for 120 h vs. 12 h (liver), ***$p = 2.86 \times 10^{-6}$ for 120 h vs. 24 h (liver), ***$p = 1.16 \times 10^{-6}$ for 120 h vs. 12 h (brain), ***$p = 2.42 \times 10^{-6}$ for 120 h vs. 24 h (brain) (**e**). Data are presented as mean values +/− SD (ns means no significance, $n = 3$). **f–h** Homogenates of major organs of GBM-bearing mice after intravenous injection with mCherry@EC (left) and mCherry@VNP (right) for 12, 24, 72, and 120 h cultured on the solid LB agar ($n = 3$) (**f**) and corresponding quantification of bacterial colonization in mCherry@EC group (**g**) and mCherry@VNP group (**h**). ***$p = 6.33 \times 10^{-7}$ for brain vs. heart, ***$p = 6.43 \times 10^{-7}$ for brain vs. liver, ***$p = 6.34 \times 10^{-7}$ for brain vs. spleen, ***$p = 6.33 \times 10^{-7}$ for brain vs. lung, ***$p = 6.36 \times 10^{-7}$ for brain vs. kidney (**g**). ***$p = 2.30 \times 10^{-4}$ for brain vs. heart, ***$p = 2.30 \times 10^{-4}$ for brain vs. liver, ***$p = 2.30 \times 10^{-4}$ for brain vs. spleen, ***$p = 2.30 \times 10^{-4}$ for brain vs. lung, ***$p = 2.30 \times 10^{-4}$ for brain vs. kidney (**h**). Data are presented as mean values +/− SD ($n = 3$). **i** Confocal images and corresponding distribution profiles of fluorescence intensity along the diameter of 3D tumor microspheres with different treatments as indicated. **j** The in situ hybridization fluorescence image of GBM tissues and corresponding fluorescence intensity of 16 S rRNA probe in different groups. *$p = 0.023$ for Trojan EC vs. PBS, *$p = 0.018$ for Trojan VNP vs. PBS. Data are presented as mean values +/− SD ($n = 3$). The nucleus, hypoxic zone, and bacteria were stained with DAPI (blue), anti-HIF-α antibody (green) and 16S rRNA probe (red), respectively. Scale bars: 100 μm. Source data are provided as a Source data file.

which was in consistent with the data on control mice injected with pure bacteria (Fig. 4b–g).

With an aim to study the intratumoural transport of constructed system, we first constructed an ex vivo model of three-dimensional cultured multicellular spheroids (MCSs) with the diameter of ~750 μm. The MCSs were made of U87MG cells rather than G422 cells since we found that G422 cells were not as good as U87MG cells in the formation of MCSs, and were easy to disperse. The PBS, GP-ICG-SiNPs (8 mg/kg ICG), mCherry@VNP (~1.0 × $10^7$ CFU) and Trojan bacteria (GP-ICG-SiNPs (8 mg/ml ICG) internalized into ~1.0 × $10^7$ CFU mCherry@VNP) were co-incubated with U87MG MCSs for 12 h, respectively. Fluorescence signals at different depths of MCSs were collected by CLSM. The three-dimensional confocal images show that the MCSs treated by Trojan bacteria display distinct green (from SiNPs) and red (from mCherry@VNP) fluorescence signals at both the edge and internal space, with a penetration depth of ~260 μm (Fig. 5i). In addition, the distribution of Trojan bacteria in MCSs is basically consistent with that of pure mCherry@VNP, but much deeper than that of free GP-ICG-SiNPs. These results indicate that the constructed Trojan bacteria could penetrate deep GBM tissues in vitro.

To verify the constructed Trojan bacteria could penetrate the deep GBM tissues in vivo, the female Balb/c mice with in situ GBM were intravenously injected with PBS, GP-ICG-SiNPs, EC, VNP, Trojan EC, or Trojan VNP, respectively. Afterward, the excision and section of deep GBM tissue were performed at 12 h post injection, followed by hypoxia-inducible factor-α (HIF-α) analysis and the bacterial 16S rRNA fluorescence in situ hybridization (FISH) analysis. The expression level of HIF-α indicates the hypoxia situation in GBM tissues and the expression of 16S rRNA indicates the location of injected bacteria in GBM tissues. As revealed in Fig. 5j, we can observe distinct green fluorescence signals of HIF-α in all groups, suggesting the high hypoxia level of deep GBM tissues. In addition, the distinct red fluorescence signals of 16S rRNA are only found in EC, VNP, Trojan EC and Trojan VNP-treated groups, suggesting Trojan bacteria with hypoxia-targeting ability indeed penetrate GBM tissues in vivo.

### Trojan bacteria-induced photothermal immunotherapy

Based on the proven ability of Trojan EC and Trojan VNP to cross the BBB, target, and penetrate GBM, we next investigated the photothermal immune efficacy of Trojan bacteria in the treatment of orthotopic GBM-bearing mice. As schematically illustrated in Fig. 6a, the orthotopic tumor model was constructed by in situ inoculation of ~8 × $10^5$ Luc-G422 cells per mouse at day −7. After the in situ GBM model was successfully constructed, GBM-bearing mice were intravenously injected with different drugs on day 0 (Treatment 1), day 5 (Treatment 2), and day 10 (Treatment 3), respectively, and photothermal treatment (PTT) was performed under 808-nm laser

irradiation at the 12th hour after each drug injection. And the photothermal treatment lasted for 5 min. On the 3rd day after the Treatment 3, these mice were sacrificed with their tumors and adjacent lymph nodes collected and homogenized for flow cytometric analysis. The concentrations of cytokines in the supernatants of tumor lysates were measured by using corresponding ELISA kits according to vendors' protocols.

The fluorescence signals of ICG can be employed for monitoring the dynamic distribution of GP-ICG-SiNPs as well as Trojan bacteria in the body. Supplementary Fig. 14 shows the fluorescence signals of ICG in the brain peaked at ~12-h post injection of Trojan bacteria. Therefore, at the 12-h post injection of Trojan bacteria, the brains of those mice were suffered by an 808 nm irradiation (1.2 W/cm², 5 min). As revealed in the photothermal images recorded by an IR camera (Fig. 6b), the rapid GBM temperature rising only occurs in Trojan EC or Trojan VNP-treated groups. In particular, the GBM-surface temperature can increase to 50.7 °C in Trojan EC group after 5-min irradiation, and 51.7 °C in Trojan VNP group after 5 min irradiation (Fig. 6c). By contrast, significant heating is not observed in other control groups under the same conditions. Next, at 5-day and 10-day post injection, GP-ICG-SiNPs, Trojan EC, Trojan VNP at the same dose were intravenously injected into the mice again. Analogously, at 5-day and 12-h post injection or 10-day and 12-h post injection, the GBM sites of these mice were irradiated by 808 nm laser and the temperature of GBM sites was stabilized at 50 °C for 5 min by adjusting the 808 nm laser power. Afterward, bioluminescence imaging was applied to visualize the antitumor effect every four days. As displayed in Fig. 6d–e, the bioluminescence signals of luc-G422 cells in Trojan EC or Trojan VNP groups are much weaker than that of the other treatment groups. The quantitative analysis shows that Trojan bacteria system has obvious inhibitory effect on tumor growth, and the corresponding inhibition rates are 66.25% of Trojan EC and 70.11% of Trojan VNP, respectively, which are much better than 16.14% of EC, 11.88% of VNP and 8.01% of GP-ICG-SiNPs (Fig. 6f). Also, survival analysis shows that survival time of mice has been significantly prolonged in Trojan bacteria group compared with other control groups (Fig. 6g). To further evaluate the antitumor effects of Trojan bacteria system, H&E staining of GBM tissues were performed. As shown in Fig. 6h, the most apparent tumor cell destruction, tissue necrosis and nuclear pyknosis are found in Trojan bacteria-treated group. As previously reported, the PPT-induced thermolysis of tumor cells would not affect the survival of neighboring non-tumor cells since the NIR light could precisely locate the tumor site, thereby minimizing the damage to surrounding normal tissue[19,30,32,33,38,39,48]. In addition, there was little difference between the trojan EC and VNP in terms of heating curve, fluorescence signal or survival time. This phenomenon might be attributed to the fact that both the EC and VNP are Gram-negative bacteria and have the similar

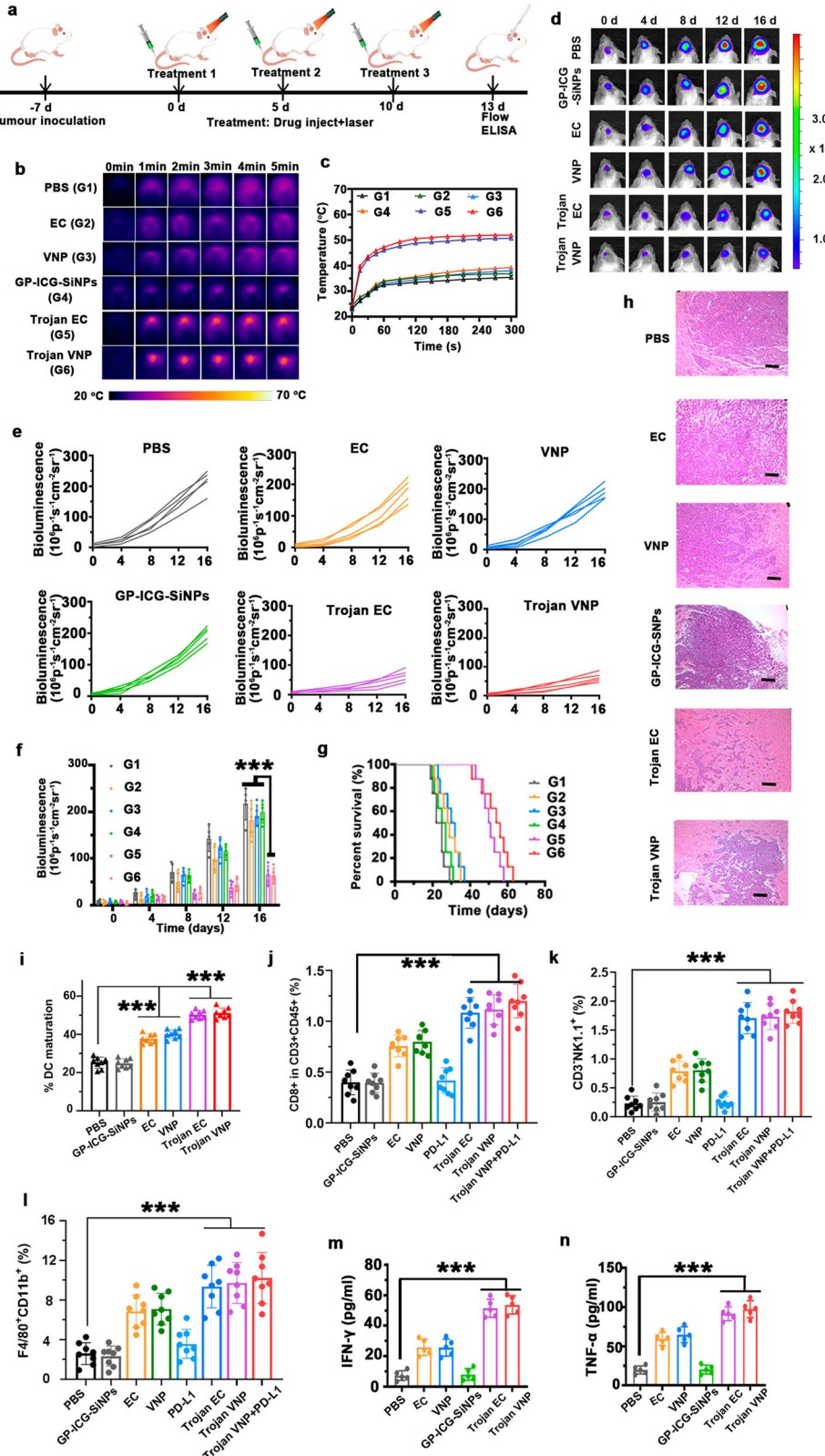

morphology and size, possibly expressing the same number of ABC transporters. Taken together, these therapeutic data demonstrated the adaptable anticancer ability of Trojan bacteria in vivo.

As previously reported, the lysates of tumor and bacterial cells can function as tumor vaccines and initiate potent antitumor immunity[33, 54]. First, we evaluated the in vivo DCs maturation triggered by Trojan bacteria system. After staining with fluorophore-

labeled specific CD11c[+] and MHC II[+] antibodies, the cell suspensions were collected for flow cytometry analysis. As shown in Supplementary Fig. 15 and Fig. 6i, compared with the PBS group, the highest level of DC maturation was observed in the groups of Trojan bacteria+laser (e.g., 47.4% DC maturation for Trojan EC + laser, 52.0% DC maturation for Trojan VNP + laser), which could be attributed to the release of tumor-associated antigens and bacterial residues upon photothermal

**Fig. 6 | Trojan bacteria-induced photothermal immunotherapy. a** Schematic illustrating Trojan bacteria therapy in a GBM-bearing mouse model. **b, c** Representative infrared images of the brains of GBM-bearing mice treated with PBS (G1), EC (G2), VNP (G3), GP-ICG-SiNPs (G4), Trojan EC (G5), and Trojan VNP (G6) under 808 nm laser irradiation (**b**) and the corresponding temperature changes of brains in different treatment groups (**c**). **d–f** Representative in vivo bioluminescence images of GBM-bearing mice (**d**), individual bioluminescence change curves (**e**) and the semi-quantification of bioluminescence intensity of brains in different groups during treatment (**f**). ***$p = 1.71 \times 10^{-7}$ for Trojan EC vs. PBS, ***$p = 2.11 \times 10^{-5}$ for Trojan EC vs. EC, ***$p = 1.87 \times 10^{-6}$ for Trojan EC vs. GP-ICG-SiNPs, ***$p = 1.71 \times 10^{-7}$ for Trojan VNP vs. PBS, ***$p = 5.88 \times 10^{-6}$ for Trojan VNP vs. VNP, ***$p = 1.87 \times 10^{-6}$ for Trojan VNP vs. GP-ICG-SiNPs (**f**). Data are presented as mean values +/− SD ($n = 5$). **g** Kaplan–Meier survival curves. Data are presented as mean values +/− SD ($n = 5$). **h** The H&E staining analysis on the brain tissue of GBM-bearing mice in different groups after therapy. Scale bars: 250 μm. All imaging experiments were repeated three times with similar results. **i** The flow cytometric analysis of matured DC cells (CD11c$^+$ MHC II$^+$) in the lymph nodes of mice post different treatments as indicated. ***$p = 2.10 \times 10^{-8}$ for PBS vs. EC, ***$p = 1.90 \times 10^{-7}$ for PBS vs. VNP, ***$p = 1.00 \times 10^{-11}$ for Trojan EC vs. PBS, ***$p = 2.00 \times 10^{-11}$ for Trojan VNP vs. PBS. Data are presented as mean values +/− SD ($n = 8$). **j** The frequencies of CD8$^+$ T (CD45$^+$CD3$^+$CD8$^+$ CD4$^+$)

cells in the tumors of these mice post different treatments as indicated. ***$p = 1.00 \times 10^{-7}$ for PBS vs. Trojan EC, ***$p = 4.30 \times 10^{-8}$ for PBS vs. Trojan VNP, ***$p = 2.80 \times 10^{-8}$ for Trojan VNP + PDL-1 vs. PBS. Data are presented as mean values +/− SD ($n = 8$). **k** The frequencies of NK cells (CD45$^+$CD3$^+$NK1.1$^+$) in the tumors of these mice post different treatments as indicated. ***$p = 1.90 \times 10^{-9}$ for PBS vs. Trojan EC, ***$p = 1.30 \times 10^{-11}$ for PBS vs. Trojan VNP, ***$p = 2.00 \times 10^{-11}$ for Trojan VNP + PDL-1 vs. PBS. Data are presented as mean values +/− SD ($n = 8$). **l** The frequencies of macrophage cells (CD11b$^+$, F$_4$180$^+$) in the tumors of these mice post different treatments as indicated. ***$p = 1.60 \times 10^{-6}$ for PBS vs. Trojan EC, ***$p = 5.30 \times 10^{-7}$ for PBS vs. Trojan VNP, ***$p = 2.10 \times 10^{-8}$ for Trojan VNP + PDL-1 vs. PBS. Data are presented as mean values +/− SD ($n = 8$). **m** Qualification of IFN-γ in serum of GBM tumor-bearing mice on the 3rd day after the Treatment 3. ***$p = 4.7 \times 10^{-7}$ for PBS vs. Trojan EC, ***$p = 3.30 \times 10^{-7}$ for PBS vs. Trojan VNP. Data are presented as mean values +/− SD ($n = 5$). **n** Qualification of IFN-γ in serum of GBM tumor-bearing mice on the 3rd day after the Treatment 3. ***$p = 2.8 \times 10^{-7}$ for PBS vs. Trojan EC, ***$p = 4.80 \times 10^{-7}$ for PBS vs. Trojan VNP. Data are presented as mean values +/− SD ($n = 5$). Statistical significance was calculated via one-way analysis of variance (ANOVA) with a Tukey post hoc test. Source data are provided as a Source data file.

effect[33,60]. Next, we found that beyond the PTT-induced tumor killing, Trojan bacteria under laser could significantly promote the activation of CD8$^+$ T cells (CD45$^+$ CD3$^+$CD8$^+$CD4$^+$) inside the treated GBM, indicating the effective activation of adaptive antitumor immunity compared to other treatment groups (Supplementary Fig. 16 and Fig. 6j), which was in agreement with the previous reports[33,55]. Furthermore, we have evaluated the role of innate immune cells (e.g., macrophages and natural killer (NK) cells) in therapeutic effects. We found that such Trojan bacteria-induced photothermal treatment could also elicit potent innate antitumor immunity by promoting intratumoral frequencies of NK cells (Supplementary Fig. 17 and Fig. 6k) and macrophages (Supplementary Fig. 18 and Fig. 6l). In addition, we also evaluated the cytokine levels of TNF-α and IFN-γ in serum samples of mice after various treatments. In the cytokine assay of serum samples, levels of both cytokines (IFN-γ and TNF-α) were higher in mice treated with Trojan bacteria than those in mice treated with pure bacteria and nanoagents (Fig. 6m, n). Meanwhile, it was shown that pure bacteria (EC or VNP) treatment could also prime similar antitumor immunity, but mostly lower than those elicited by the Trojan bacteria-induced photothermal treatment. We concluded that the Trojan bacteria under laser irradiation achieved the thermolysis of GBM cells, promoting the release of tumor-associated antigens (TAAs)[33,54,61]. Meanwhile, Trojan bacteria under laser irradiation could kill the host bacterial cells to promote the release of diverse pathogen-associated molecular patterns (PAMPs), which could promote the activation of innate immune cells, such as macrophages and NK cells[33,54,60,62]. On the other aspect, these PAMPs would elicit potent adaptive antitumor immune response by promoting tumor-infiltrating frequencies of activated CD8$^+$ T cells. Taken together, both innate and adaptive antitumor immunity induced by Trojan bacteria under laser irradiation would work together to further suppress tumor growth.

### The elimination of residual bacteria after photothermal immunotherapy

Next, we examined whether the photothermal immunotherapy induced by Trojan bacteria could facilitate the elimination of bacteria from the GBM-bearing mice. In detail, the GBM-bearing mice were intravenously injected with (G1) PBS, (G2) -1 × 10$^7$ CFU mCherry@EC, (G3) -1 × 10$^7$ CFU mCherry@VNP, (G4) Trojan mCherry@EC (e.g., GP-ICG-SiNPs (8 mg/kg ICG) internalized into -1 × 10$^7$ CFU mCherry@EC) or (G5) Trojan mCherry@VNP (e.g., GP-ICG-SiNPs (8 mg/kg ICG) internalized into 1 × 10$^7$ CFU mCherry@VNP), respectively. At the 12-h post injection, the brains of those mice were suffered by an 808 nm irradiation (1.2 W/cm$^2$, 5 min), followed by ex vivo imaging of the main

organs after 5 days or 7 days of photothermal immunotherapy. As shown in Fig. 7a, b, relatively strong red fluorescence signals of mCherry could be detected in GBMs and livers in G2-G5 groups after 5 days of photothermal immunotherapy, while these signals were only observed in GBMs in G2 and G3 groups after 7 days of photothermal immunotherapy. These findings are further confirmed by major organs and GBM tissues harvesting, homogenization and then culturing homogenates on plates (Fig. 7c, d). It is found that sporadic colonies grew from homogenates of GBMs only in G2 and G3 groups after 7 days of photothermal immunotherapy. Accumulating evidence demonstrated the residual Trojan bacteria could be totally eliminated from the body basically after 7 days of photothermal immunotherapy. In addition, the corresponding H&E staining of major organs harvested from the GBM-bearing mice after 30 days of treatments further demonstrated the safety of the Trojan bacteria (Supplementary Fig. 19). On the other aspect, the SiNPs entering the brain might be pumped back into the blood circulation through the efflux pumps such as multidrug resistance protein (MRP) and P-glycoprotein (P-gp)[63]. Ultimately, the SiNPs could be biodegradable into renal clearable components[64]. On the basis of this, SiNPs have received the Food and Drug Administration (FDA)-approved investigational new drug approval for the first-in-human clinical trial in January 2011 (NCT01266096, NCT02106598)[65,66].

## Discussion

Due to the existence of the BBB in GBM, the accumulation of peripherally administered drugs into brain is seriously hindered. It has been reported that bacteria can bypass the BBB transcellularly, paracellularly, and/or in infected phagocytes[34–36]. Intriguingly, cancer therapeutics internalized into bacteria would potentially circumvent drug delivery issues in GBM therapy. However, this progress is still in its infancy due to the lack of a natural connection between therapeutics and the microbiome. Herein, the developed Trojan bacteria system naturally bridge the divide between therapeutics and bacteria. We have previously demonstrated that bacteria including Gram-negative as well as Gram-positive bacteria actively swallowed GP-conjugated nanoparticles through bacteria-specific ABC transporter pathway for ultrasensitive diagnosis of bacterial infections[41]. In this context, we have successfully constructed Trojan bacteria as drug delivery vehicles for GBM therapy, in which the Trojan bacteria system was made of GP-ICG-SiNPs internalized into VNP or EC. Consistently with previous reports, the as-prepared GP-ICG-SiNPs could be internalized into the host bacteria, confirmed by TEM, SEM, and confocal images. We also performed an inhibition assay as well as a competition

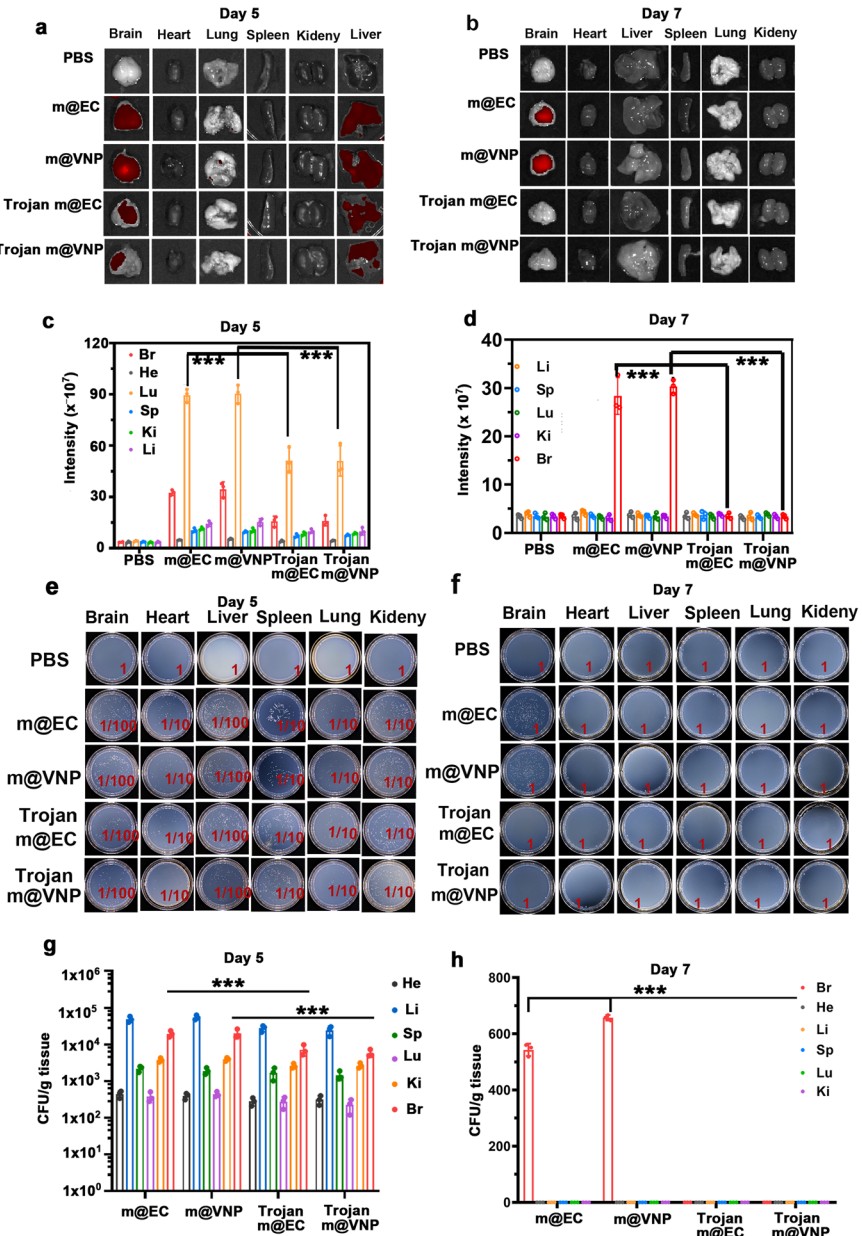

**Fig. 7 | The elimination of residual bacteria after photothermal immunotherapy.** The fluorescence distribution in the main organs (heart, liver, spleen, lung, kidney, and brain) of GBM-bearing mice after 5 days (**a**) or 7 days (**b**) of photothermal immunotherapy. The mice were intravenously injected with PBS, ~1×10⁷ CFU mCherry@EC (m@EC), ~1×10⁷ CFU mCherry@VNP (m@VNP), Trojan m@EC (e.g., GP-ICG-SiNPs (8 mg/kg ICG) internalized into ~1×10⁷ CFU m@EC) or Trojan m@VNP (e.g., GP-ICG-SiNPs (8 mg/kg ICG) internalized into 1×10⁷ CFU m@VNP), respectively. At the 12-h post injection, the brains of those mice were suffered by an 808 nm irradiation (1.2 W/cm², 5 min), followed by ex vivo imaging of the main organs after 5 days or 7 days of photothermal immunotherapy. The corresponding quantitative analysis of fluorescence intensity of main organs in different groups after 5 days (**c**) or 7 days (**d**) of photothermal immunotherapy. ***$p = 5.80 \times 10^{-3}$ for

m@EC vs. Trojan EC, ***$p = 3.20 \times 10^{-3}$ for m@VNP vs. Trojan VNP (lung) (**c**). ***$p = 3.60 \times 10^{-4}$ for m@EC vs. Trojan EC, ***$p = 7.10 \times 10^{-6}$ for m@VNP vs. Trojan VNP (brain) (**d**). Data are presented as mean values +/− SD ($n = 3$). Homogenates of major organs of GBM-bearing mice in different groups after 5 days (**e**) or 7 days (**f**) of photothermal immunotherapy cultured on the solid LB agar. **d** Corresponding quantification of bacterial colonization on LB solid plates in different treatment groups after 5 days (**g**) or 7 days (**h**) of photothermal immunotherapy. ***$p = 3.10 \times 10^{-4}$ for m@EC vs. Trojan EC, ***$p = 5.60 \times 10^{-4}$ for m@VNP vs. Trojan VNP (brain) (**g**). ***$p = 1.8 \times 10^{-6}$ for m@EC vs. Trojan EC, ***$p = 1.10 \times 10^{-7}$ for m@VNP vs. Trojan VNP (brain) (**h**). Data are presented as mean values +/− SD ($n = 3$). Statistical significance was calculated via one-way analysis of variance (ANOVA) with a Tukey post hoc test. Source data are provided as a Source data file.

assay to further verify GP-ICG-SiNP accessing into the bacterial intracellular volume through the ABC transporter pathway.

Subsequently, we demonstrated the notion that the constructed Trojan bacteria could cross BBB, target, and penetrate GBM. As such, in the constructed Trojan bacteria, the bacteria could take therapeutics together to cross BBB, target and penetrate GBM tissues. Consequently, under the irradiation of 808-nm laser, photothermal treatment of ICG can induce tumor cell destruction and bacterial rupture.

The produced lysates of tumor and bacterial cells upon PTT effects can act as immune stimulants to enhance the antitumor immune response. Compared with the use of equivalent free GP-ICG-SiNPs or pure bacteria for GBM therapy, this constructed Trojan bacteria system not only greatly augmented targeted delivery of GP-ICG-SiNPs toward the GBM, but also synergistically promoted antitumor immune responses that prolonged the survival of GBM-bearing mice. In addition, distinguished from pure bacteria therapy which would preserve a lot of

residual bacteria after treatment, that residual bacteria could be effectively eliminated from the body after the Trojan bacteria treatment. Accumulating evidence demonstrated that the therapeutic system of Trojan bacteria could achieve photothermal immunotherapeutic effects and safety profiles under modulation of NIR light irradiation, providing a plausible microbiota-based therapeutic strategy against CNS diseases.

## Methods

### Synthesis of GP-ICG-SiNPs

SiNPs were synthesized by mixing 1,8-naphthalimide and 3-aminopropyltrimethoxysilane, followed by 365 nm UV irradiation at room temperature for 40 min. The resulted solution was then centrifuged at $3381 \times g$ for 20 min to remove unreacted reagents, and further purified by dialysis (MWCO, 1000, Spectra/Pro). The as-synthesized SiNPs (200 μL, 20 mg/mL) were mixed with GP (100 μL, 10 mg/mL) at 70 °C for 6 h, followed by the addition of 0.02 mg of $NaBH_4$. After reacting 12 h at room temperature, the stable GP-modified SiNPs were obtained. To remove unreacted GP, Nanosep centrifugal devices (MW cutoff, 3 kDa; Millipore) were used to filter the reaction solution at $4226 \times g$ for 15 min. After that, ICG were co-incubated with GP-SiNPs, and stirred at 4 °C overnight. Then, the unreacted ICG were centrifuged by Nanosep centrifugal devices (MW cutoff, 3 kDa; Millipore) at $3945 \times g$ for 15 min. Then the products of GP-ICG-SiNPs were harvested and stored at 4 °C in the dark for the following experiments. Transmission electronic microscopy (TEM, Philips CM 200) with 200 kV was used for the characterization of the morphology and size of the nanoagents. UV-vis absorption spectra of nanoagents were measured by A 750 UV-vis near-infrared spectrophotometer (Perkin-Elmer lambda). Photoluminescence (PL) spectra of nanoagents were recorded by a spectrofluorimeter (HORIBA JOBIN YVON FLUORMAX-4). Dynamic light scattering (DLS) of nanoagents was analyzed by a Delsa™ nano submicron particle size analyzer (Beckman Coulter, Inc).

### Bacterial culture

Attenuated *Salmonella* strain VNP20009 (VNP) and *Escherichia coli* 25922 (EC) were purchased from American Type Culture Collection (ATCC). These bacterial cells were harvested at the exponential growth phase when they were cultured in LB liquid medium (250 rpm, 37 °C). Afterward, the bacterial suspensions were washed twice and resuspended in PBS buffer for the following experiments. The bacteria count in solution was determined by the measurement of the corresponding optical density (OD) at 600 nm. The numbers of bacterial colonies were counted by a colony counting instrument (Czone 8).

### Construction of mCherry@VNP and mCherry@EC

To construct mCherry@VNP, monoclonal VNP was selected and inoculated in 2 mL of LB liquid medium without ampicillin. Then, 0.5 mL of bacterial solution was added into 50 mL LB liquid medium and cultured at 37 °C until its $OD_{600}$ reached 0.5. After that, the shake flask was ice-bathed for 30 min, and 25 mL of bacterial solution was added to a precooled 50 mL round bottom centrifuge tube, and centrifuged at 4 °C and $94 \times g$ for 15 min to obtain bacterial pellets. Then the bacterial precipitation was washed twice with precooled sterile water. The bacterial pellet was resuspended with 10 mL of 10% sterile glycerin, and centrifuged at 4 °C and $94 \times g$ for 15 min. Then 50 μL 10% sterile glycerin was added to make the bacterial precipitation fully suspended and transferred to a precooled 0.5 mL centrifuge tube to form electrically transformed competent cells. We took 1 ng of mCherry plasmid and added it to VNP competent cells, mixed it gently, ice bath for 1 min, transferred it into a cold electroporation cup (Bio-Rad), and placed it on the electrode for electric shock transformation (Electric shock parameter: voltage = 2.5 kV, capacitance = 25 μF, resistance = 200 Ω). After the electric shock, we added immediately 1 mL of LB liquid medium to the electroporation cup, mixed, and transferred all to a sterile 1.5 mL centrifuge tube, let it culture at 30 °C and 160 rpm for 1.5 h. Then, we took about 200 μL of bacterial solution and plated it on solid LB agar containing ampicillin, and incubated the LB solid plate at 37 °C until a single colon is formed. To construct mCherry@EC, we added 1 ng mCherry plasmid to 100 μL of EC competent bacteria, and iced bath for 30 min. Then the bacteria liquid was heat shocked at 42 °C for 90 s, and then ice-bathed for 2 min. The bacterial cells were added with 900 μL LB liquid medium without ampicillin and cultured at 180 rpm at 37 °C for 45 min. To obtain the bacterial precipitation, the bacterial solution was centrifuged at $3381 \times g$ for 5 min, and about 100 μL supernatant was retained for mixing. Pipette 100 μL of the resuscitated bacterial solution evenly on the LB solid plate containing ampicillin. Afterward, the plate was cultured upside down in an incubator for 12 to 16 h at 37 °C.

### Construction of Trojan bacteria

The 20 μL of VNP or EC suspension with ~$1.0 \times 10^7$ CFU was purified and resuspended, followed by incubation with GP-ICG-SiNPs (15 mg/mL, 200 μL) in a shaking incubator at 200 rpm, 37 °C for 2 h. The constructed Trojan bacteria were harvested by centrifugation at $4508 \times g$ for 10 min. Afterward, the harvested Trojan bacteria were resuspended again and washed with PBS buffer at least 3 times to remove excess GP-ICG-SiNPs or nonspecifically absorbed GP-ICG-SiNPs. To characterize the constructed Trojan bacteria, the solution containing Trojan bacteria of 10 μL was transferred onto a microscope slide and covered by a coverslip, followed by confocal laser scanning microscope imaging (CLSM, Leica, TCSSP5 II) with 30% power of diode laser. Of note, all fluorescent images were captured under identical optical conditions. The processing and analysis of the region of interest (ROI) were performed by the image analysis software (Leica Application Suite Advanced Fluorescence Lite). The morphology and structure of Trojan VNP and Trojan EC were characterized by TEM and SEM.

### Cellular experiments in vitro

The HBMEC, bend.3, U87MG, HeLa, 4T1, GL261, (Luc)-G422 mouse glioblastoma cell line was obtained from Shanghai Zhong Qiao Xin Zhou Biotechnology and cultured under appropriate conditions. Bone marrow cells from the femurs and tibias of female Balb/c mice were flushed and depleted of red blood cell (RBC) by hypotonic lysis using RBC lysing buffer (Sigma). Cells were grown from precursors at a starting concentration of $1 \times 10^6$ cells/ml in RPMI 1640 for 3 days and then nonadherent cells were washed out. Another 10 ml of fresh complete medium containing 20 ng/ml rmGM-CSF (ProSpec), 20 ng/ml rmIL-4 (ProSpec), was added, and on day 6 half of the medium was replaced. On day 8, nonadherent and loosely adherent cells were harvested by vigorous pipetting and placed in the lower chamber of the transwell system. G422 cells were inoculated in a 96-well plate, co-incubated with EC, VNP, GP-ICG-SiNPs Trojan EC or Trojan VNP for 6 h, irradiated with or without 808-nm laser for 5 min, and then cleaned with sterile PBS. MTT assay was used to determine the cellular viability. On the other aspect, the treated cells were stained with Calcein-AM (CAM) and propidium iodide (PI), and then analyzed by confocal microscopy (CAM: $\lambda_{ex} = 488$ nm, $\lambda_{em} = 500-545$ nm; PI: $\lambda_{ex} = 543$ nm, $\lambda_{em} = 560-620$ nm).

### Human brain microvascular endothelial cell model

A 12-well transwell plate with 2 μm of mean pore size membrane was used to construct the in vitro human brain microvascular endothelial cell (HBMEC) model. The HBMEC cells ($1.0 \times 10^5$ cells/well) were seeded in the transwell insert with 12 mm diameter. The transendothelial electrical resistance (TEER) values were detected by a Millicell-ERS volt-ohmmeter to monitor the cell monolayer integrity during the cell culture process. A TEER value between 150 and 300 Ω cm² was suitable for further experiments.

## Multicellular spheroids model

The 3D tumor spheroids of U87MG cells were obtained using a liquid overlay method. Each well of 96-well plates was pre-coated with 100 μL of the FBS-free medium containing sterile agarose (2%, w:v). Subsequently, U87MG cells (5000 cells/well) were seeded into each well and cultured in the medium containing FBS (10%, v:v). The tumor spheroids were allowed to grow up to attain diameter about 750 μm for 8 days at 37 °C. The Z-stack scanning was performed on the U87MG multicellular spheroids from top to bottom with 40 μm per section by CLSM.

## Animal experiments

All animal experimental procedures were performed according to the Guideline for Animal Experimentation with the approval of the animal care committee of Soochow University.

To construct the GBM-bearing mice model, we used a brain stereotaxic apparatus to determine the location of glioma cells in the mouse brain, and injected luc-G422 cells to monitor the size of the mouse brain tumor by biofluorescence signals (Supplementary Fig. 20). The detailed experimental operations were as follows: (1) Digest luc-G422 cells, dilute Matrigel in a 4-fold concentration gradient in PBS, and use the diluted resuspended cells to place on ice. (2) After the mouse was anesthetized, the head of the mouse was fixed on the brain stereotaxic apparatus. After sterilizing the brain skin with disinfectant, cut the skin with a scalpel, stop the bleeding with a cotton swab, and observe the position of the fontanelle. (3) Determine the coordinates of the fontanelle by the brain locator, and record the values. Afterward, the position of the syringe was moved by the coordinates of the front 0.05 mm (+), the left 0.19 mm (+), and the depth 0.31 mm (+) to determine the injection site and mark it. (4) Use a micro drill to make a hole at the marked site, use a syringe to draw 5 μL of luc-G422 cells, and slowly inject the cell suspension into the hole. (5) Take a small piece of hemostatic cotton to fill the hole in the brain, suture it, apply antibiotics, and wait for the mouse to wake up. The maximum diameter of the mouse tumor volume did not exceed 15 mm, which was approved by the animal care committee of Soochow University.

To study the in vivo distribution of bacteria, mCherry@EC or mCherry@VNP ($-1.0 \times 10^7$ CFU per mouse) was intravenously injected into the healthy mice or GBM-bearing mice after inoculation with Luc-G422 cells ($10^5$/per mouse) for 7 days. Afterward, the mice were sacrificed at specific time points (12, 24, 36, 72, and 120 h). Accordingly, the organs including heart, liver, spleen, lung, and kidney as well as brain were extracted, followed by imaging via an in vivo optical imaging system (IVIS Lumina III). Meanwhile, the excised organs were homogenized in sterile PBS, in which the suspension solution was collected from the tissue dispersions by centrifugation at $94 \times g$ to remove tissue fragments. Finally, the collected suspension solution was diluted by PBS buffer and cultured on LB solid medium at 37 °C for 12 h, followed by counting bacterial colonies with a colony counting instrument (Czone 8).

To study the therapeutic effects in vivo, the GBM-bearing mice were, respectively, injected with (G1) PBS, (G2) $-1 \times 10^7$ CFU EC, (G3) $-1 \times 10^7$ CFU VNP, (G4) GP-ICG-SiNPs (8 mg/kg ICG), (G5) Trojan EC (e.g., GP-ICG-SiNPs (8 mg/kg ICG) internalized into $-1 \times 10^7$ CFU EC), and (G6) Trojan VNP (e.g., GP-ICG-SiNPs (8 mg/kg ICG) internalized into $1 \times 10^7$ CFU VNP) after inoculation for 7 days ($n = 5$, female, 6–7 weeks old). After intravenous injection for 12 h, the GBM sites were irradiated by 808-nm laser (1.2 W/cm², 5 min). The laser spot was adjusted according to the burr hole on the skull. We recorded the infrared thermal images by using a FLIR Ax5 camera, and quantified the temperature by using the BM_IR software. The temperature of GBM sites was stabilized at 50 °C for 5 min by adjusting the 808 nm laser power. Next, at 5-day and 10-day post injection with bacteria, GP-ICG-SiNPs (containing 8 mg/kg of ICG), Trojan EC, Trojan VNP were intravenously injected into the mice again. Analogously, at 5-day and 12-h post injection or 10-day and 12-h post injection, the GBM sites of these mice were irradiated by 808 nm laser and the temperature of GBM sites was stabilized at 50 °C for 5 min by adjusting the 808 nm laser power. Afterward, bioluminescence imaging was applied to visualize the antitumor effect every 4 days. On the other aspect, we harvested the tumor tissues and the carotid lymph nodes after 3 days post-last treatment and homogenized them into single cell suspensions, followed by flow cytometry analysis. The details of antibody panel for spectral flow cytometry analyses were listed in Supplementary Table 1. Meanwhile, serum samples were isolated from mice after various treatments and diluted for analysis. What's more, the tumors were harvested, fixed in 4% paraformaldehyde for 24 h, and embedded in paraffin. Tissues were sectioned, stained with H&E, and then observed with an optical microscope. To examine whether residual bacteria were eliminated from the body after treatment, the excised organs after cancer treatment were imaged by the in vivo optical imaging system (IVIS Lumina III). Meanwhile, the harvested organs were homogenized in sterile PBS and the residual bacterial dispersions were collected by centrifugation. Then the collected bacterial solutions were diluted and cultured on an agarose medium for 12 h. The numbers of bacterial colonies were obtained by a colony counting instrument (Czone 8).

## Human studies

Human blood samples were provided by a healthy volunteer following written informed consent. The study protocols using human blood samples were approved by the ethics committee of Soochow University. The authors state that all human blood experiments were performed in strict accordance with the relevant laws and institutional guidelines.

## Immunofluorescence staining

Female orthotopic glioblastoma tumor-bearing mice were intravenously injected with PBS, EC, VNP, GP-ICG-SiNPs, Trojan EC, or Trojan VNP, respectively. Mice were sacrificed after 12 h of treatment, and GBM tissues were collected and stored at −80 °C. Next, GBM tissue slices were prepared with the assistance of a Leica Microsystems VT1200S. GBM slices were stained with HIF-1α primary antibody for 2 h after blocking nonspecific binding sites in samples with 10% FCS. Then, slices were washed 3 times with 10% FCS. Next, the slices were stained with FITC-labeled secondary antibody of anti-HIF-1α (Merck, clone: H1α67, catalog no. MAB5382) of 5 μg/ml for an additional 2 h. The slices were then washed with 10% FCS for 3 times. Bacteria were stained with 16S rRNA probe. Briefly, slides were stained with Cy7 labeled-FISH probes of 5 ng/μl in hybridization buffer at 56 °C overnight. The sequence of the 16S rRNA probe is GCT GCC TCC CGT AGG AGT. Nuclei were stained with 10 μM DAPI. Cover slides were mounted with Fluoromount-GTM medium (Southern Biotechnologies). Sections were viewed with a wide-field fluorescence microscopy.

## Statistical analysis

The confocal images were processed by the commercial image analysis software (Leica Application Suite Advanced Fluorescence Lite, LAS AF Lite) and common software of ImageJ (NIH Image; http://rsbweb.nih.gov/ij/). Error bars represent the standard deviation obtained from three independent measurements. All the statistical analyses were performed using the Origin and GraphPad Prism 7 software. The statistical significance of differences was determined by a one-way ANOVA analysis.

## Reporting summary

Further information on research design is available in the Nature Research Reporting Summary linked to this article.

## Data availability

The main data supporting the results in this study are available within the paper and its Supplementary Information. Source data are provided with this paper.

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

## Acknowledgements

We thank Dr. Liangzhu Feng (Soochow University, China) and Dr. Fei Peng (Harvard University, USA) for their technical support and valuable suggestions especially in this COVID-19 pandemic period. The authors acknowledge financial support from National Natural Science Foundation of China (No. 21825402 and 22074101), Natural Science Foundation of Jiangsu Province of China (No. BK20191417), the China Postdoctoral Science Foundation (No. 2021M692347) and the Program for Jiangsu Specially-Appointed Professors to the Prof. Yao He, a project funded by the Priority Academic Program Development of Jiangsu Higher Education Institutions (PAPD), 111 Project as well as the Collaborative Innovation Center of Suzhou Nano Science and Technology (NANO-CIC).

## Author contributions

R.S., M.Z.L., H.Y.W., and Y.H. conceived and designed the research. R.S. and M.Z.L. carried out most of the experiments and analyzed the data. J.P.L., B.B.C., Y.M.Y., and B.S. performed additional experiments and characterizations. R.S. and H.Y.W. wrote the manuscript.

## Competing interests

The authors declare no competing interests.
