## [Peer Review File · Nature Communications]

REVIEWER COMMENTS

Reviewer #1 (Expertise: Glioblastoma, micro-environment)- Remarks to the Author:

Rong Sun et al. report on the construction of Trojan bacteria as drug delivery vehicles to enhance the anti-tumor immune response against glioblastoma cells. The study is based on previous work of Houyu Wang and Yao He showing that Gram-positive bacteria actively swallowed GP-conjugated nanoparticles through bacteria-specific ABC transporter pathway (Nat. Commun. 2019). The topic of the present study is interesting, as it extends the initial work to preclinical assays against Glioblastoma and the therapeutic system of Trojan bacteria could potentially achieve photothermal immunotherapeutic effects. The experimental plan is also well designed, and data related to the production and functional assessment of Trojan vectors are convincing. The work is however deserving major revision as it is still lacking rigor to demonstrate Trojan bacteria as a potential new oncolytic treatment for GBMs. Moreover, Trojan bacteria treatment alone only provides limited survival benefit to GBM-bearing mice, that is calling for additional experiments to combine this treatment with immunotherapy and/or other approaches that also boost the anti-tumor immune response. The authors would improve their manuscript by answering the following comments:

Major comments:

- 1 - Construction of Trojan bacteria (Fig 1f). Is the maximal fluorescence obtained at 15 mg/ml of GP-ICG-SiNPs? Does fluorescence decrease above 15 mg/ml? Does increased fluorescence interfere with the process of NIR-induced thermolysis?
- 2 - Trojan bacteria system against tumour in vitro, Fig 3.
 - 2.1 How do G422 tumor cells survive under EC and VNP laser irradiation? The survival assay is missing.
 - 2.2 Alternative glioblastoma lines such as GL261 or CT2A (to G422 cells) and non-GBM tumor cells should also be tested to assess the general/specific efficiency of the Trojan bacteria system on GBM cells.
 - 2.3 What are DCs used in the transwell assay (Fig 3g)? The effect on CD80 and CD86 expression is interesting but CD80 and CD86 are constitutive markers of DCs, not specific of activated DCs. Additional labeling of MHCII would be more accurate.
- 3 - In vivo behavior of Trojan bacteria, Fig 4.
 - 3.1 How does Trojan bacteria injection affect the health of treated mice? Data on internal temperature, blood and CSF cytokine level should be provided.
 - 3.2 Statistics are missing in Figs 4a, c, d, f, g, i, m-p
- 4 - Trojan bacteria crossing BBB, targeting and penetrating GBM, Fig 5.
 - 4.1 TransEC migration assays should use additional model of mouse ECs such as a mouse brain EC line (bEnd.3 cells).
 - 4.2 Control with no leakage is missing (cadaverin or OVA-Alexa).
 - 4.3 Fig 5C shows increased entry of Trojan Bacteria into the brain of GBM-bearing mice/control EC/VNP-injected mice (Fig 4c). But data on control mice injected with Trojan Bacteria-GP-ICG-SiNPs are missing. The authors should also discuss why bacteria better enter GBM brains than control brains.
 - 4.4 Authors use a U87MG (human GBM/astrocytoma) cell line for neurosphere infection experiments. They should justify why they use a new line and not the previous G422 cells.
 - 4.5 Data quantification is missing for Figs 5i, j. Please also provide low magnification of brain tissues in addition to Fig. 5j. This will inform on the extension of brain tissue infection by iv injected Trojan VNP.
- 5 - Trojan bacteria-induced photothermal immunotherapy, Fig 6.
 - 5.1 Injection of mCherry@VNP or mCherry@EC in control female Balb/c mice is missing. A
 - 5.2 Does thermolysis of tumor cells affect the survival of neighboring non-tumor cells? Authors could use a GFP-transfected tumor cell line and Caspase3 labeling to assess cell death in GFP-brain cells around the GFP+ tumor cells.
 - 5.3 T cell activation should be assessed by CD69 labeling

5.4 Assessment of tumor-specific T cell response would require tetramer labeling. This could be performed using the GL261 GBM cell line.

5.5 CD80/86 are constitutive markers of DCs, not of activated DCs.

5.6 How do authors explain that in vivo DC activation is far less boosted by Trojan bacteria than in vitro (Fig 3g)?

5.7 No statistics is provided for the important Fig 6g, but the low survival benefit of Trojan bacteriolysis does not allow to claim that Trojan bacteria offer a potential therapeutic approach against GBM. To increase the interest of their preclinical assays, the authors should consider testing whether prophylactic treatment with Trojan bacteria could improve the effect of immunotherapy with anti-PD-1 and/or CTLA4 antibodies on GBM-bearing mice.

6 - Trojan bacteria clearance, Fig 7. Although reduced at G6 compared to G3, bacterial infection remains in the brain. Could the authors show that remaining bacteria do not compromise brain tissue structure and function?

Minor comments:

1 - Paragraph reorganization:

&1 'Design of Trojan bacteria system' and &2 'Characterization of Trojan bacteria system' should be fused, and description of the Trojan bacteria system model in &2 should go into the Introduction.

&2 'Characterization of Trojan bacteria system' is too long and could be split into 2 &s: & 'Trojan Design' and & 'Trojan bacteria are ABC transporter pathway-dependent'

2 - Replace reference #42 by #41, in the Discussion for the sentence:

'We have previously demonstrated that bacteria including Gram-negative as well as Gram-positive bacteria actively swallowed GP-conjugated nanoparticles through bacteria-specific ABC transporter pathway for ultrasensitive diagnosis of bacterial infections⁴²'.

3 - The number of SFigs (16) could be reduced by gathering data related to the same experiment or the same paragraph of results.

4 - p7: typing mistake, please change 'by 32C' for 'to 32C'

5 - Provide explanations on serum biochemical analysis.

6 - NIR p12 not defined.

7- Exp design of Fig 6 should be better explained.

8. Incorrect sentence p 23 'Accumulating evidence demonstrated only Trojan bacteria-treated mice under NIR irradiation could help to eliminate bacteria from the body'

Reviewer #2 (Expertise: Bacteria based cancer therapy)- Remarks to the Author:

This paper expands a nice tool in brain-targeting delivery across BBB based on Trojan bacteria. It is a topic of interest to researchers in cancer treatment, microbiology, biomaterials, and other related fields. The article is well organized and tells a complete story. Some minor revisions should be considered before publication in Nature Communications.

Comments:

1. Please explain why silicon nanoparticles were chosen in current study and possible elimination pathway after entering the brain.

2. Glucose transporters (such as GLUT 1) are often considered as SLC transporters, rather ABC transporters. Also, the entry of nanoagents can be attenuated by non-specific inhibitor of ATPase, namely NaN₃, which can affect various pathways via inhibiting ATPase or cytochrome c oxidase. It would be reasonable to verify the mechanism using more classical and specific inhibitors of ABC

transporters, or the inhibitor of SLC transporters.

3. It would be instructive to record the TEER of HBMEC BBB model after co-incubating with EC or VNP, which may evidence the possible mechanism of EC or VNP across the BBB (Transcellularly, paracellular or infected phagocytes). It is also possible that VNP induces inflammation to influence the integrity of the BBB.

4. Figure 2C is not clear. Please enlarge the figure as it is important to verify that nanoagents enter bacteria rather than absorb nonspecifically on bacterial surface.

5. In supplementary Figure 7, please explain the reason why temperature did not increase with the concentration of ICG.

6. There is no description for Figure 2f and g in the main text. Also, there were no control values of healthy mice in Fig. 4h-q, while it was claimed that "Compared with untreated healthy mice, all serum biochemical....."

7. It's suggested to mention the recent advances in the utilization of bacteria to improve accumulation of therapeutics in tumor site (such as Adv. Mater. 2021, 2106669; Nat. Commun. 2021, 12:6584).

8. Ref 53 is inappropriate to support that NaN₃ is an inhibitor of ABC transporter.

Reviewer #3 (Expertise: Bacteria based cancer therapy)- Remarks to the Author:

The manuscript entitled "Trojan bacteria cross blood-brain barrier for glioblastoma photothermal immunotherapy" employed bacteria to enhance photothermal effects leading to lysis of Trojan bacterial cells and the adjacent tumor cells which promote anticancer immune responses. Although targeted therapy for aggressive glioblastoma tumor model is an innovative approach, overall data quality (especially immunological data explaining mechanism or very less experimental sample number in vivo) was not enough to be published in Nature Communications.

1. In schematic illustration of Fig. 1. bacterial debris and tumor-associated antigen induce maturation of DCs (mDCs). Then mDCs present the tumor-specific antigens to activate T cells for secreting TNF α and IFN γ . There is no strong evidence for this hypothesis from author's data because no data was presented to demonstrate immature DCs in vitro and in vivo before maturation. For example, author have to show population of immature DCs isolated from bone marrow (all activated markers of DC must be negative in all groups) before co-culture with antigens. Authors only showed population of DCs (CD80+CD86), but DCs should express MHCI/II which is one of three required signals to stimulate T cells.

2. Author detected cytokines (TNF α and IFN γ) in serum by ELISA in the 16th day after treatment. How authors know these secreted cytokines were derived from T cells, not from macrophages, NK cells, or the other immune cells? Once tumor-bearing mice received bacteria, it may also induce secretion of IFN γ from NK cells and TNF α from macrophages.

3. After G422 cells with different treatments + Laser (Fig. 3e), authors checked maturation of dendritic cells (DC) based on CD86 and CD80. Although population of DC increased (CD80+CD86+), it is not enough marker to confirm maturation of DCs. Authors should further analyse more maturation markers of DCs such as CCR7, MHCI/II, IL-12, IL-1 β . Besides, author mentioned CD11c+ marker for DC maturation in Material and Method but the result did not show in vitro and in vivo. I suggest CD11c+CD86+ for DCs gating.

4. There is no information of process for DCs isolation from bone marrow of mice (which protocol author applied for DCs isolation). It should be clarified because it may also contain macrophages or the other cells.

5. For FACS gating (Fig. 3g), there is no Fluorescence Minus One (MFO) control that properly interpret flow cytometry data. It is hard to know the gating is correct or not.

6. There are 8 groups in study (Fig. 3g) but there is no group of G422 + laser? Whether G422 cell debris caused by laser 488 nm (without bacteria) may induce maturation of DCs? What is blank group?

7. The morphology of immaturation and maturation DCs also should be shown.

8. In Fig. 4, the unit of BUN measurement was mmol/LT. What does LT mean?

9. In Fig. 4h-q, normal values should be suggested and normal range should be presented in yellow-shaded areas. All the values measured should represent the quartiles and whiskers mark the 10th and 90th percentiles.

10. Starting points of treatment seem to be different in different groups (Fig 6d). For example, in

Trojan VNP group, no tumor signal was observed when started treatment compared with the other groups (Fig. 6d).

11. Authors carried on the therapeutic experiments with n=5 mice/group. To ensure reliable data of in vivo therapeutic experiments, authors should increase the number of animals.

12. There is no p value in Fig. 5 b, d, e, g, h, Fig. 6, f & g.

13. Authors collect tumors for FACS (T cells) and serum for ELISA (cytokines) at the same time (16 days after treatment). Why FACS data has 3 samples (very less sample to claim role of T cell against tumor) (Fig. 6 k&l) while cytokine data have 5 samples (Fig. 6 m&n)? In addition, the time point for immunological analysis is too late (16 days after treatment). How about early time point? How about the role of innate immune cells in therapeutic effects?

14. The presence of certain bacteria, especially bacterial debris after using laser is associated with inflammation that results in strong recruitment and activation of innate immune cells, especially neutrophils, macrophages or NKs cells (Quibin Lin et al Nat com 2021). In this study, authors mainly focus on T cells. Innate immunity may or may not contribute to any therapeutic effects but author did not show any data regarding the role of innate cells against tumor (innate immune cell data must be required in this study).

15. In Fig. 6 i and j, population of gating is not clear, how authors discriminate cancer cells and DCs (because cancer cells also may express CD80 and CD86, there is no MFO and the dot blot style should be required. Same as in vitro, CD86 marker is not expressed only by DC, but also strong expression on M1-like macrophages thus authors have to show gating strategy with more markers to confirm DC maturation after treatment (such MHCI/II) to discriminate with the other immune cells, especially macrophages. Although author mentioned in materials and methods, there is no data of CD11c marker for DCs gating in main and supplementary data. Through this study, there is no gating strategy of FACS and authors did not use MFO as a control. And the number of samples for FACS analysis is very less (n=3) (at least over 5 mice/group is recommended), thus FACS data is not reliable.

16. Authors demonstrated increased total population of CD8+ T cells to claim role of T cells against tumor is too weak evidence because they may be an exhausted T cells (high population but no function) or these T cell may be specific to bacteria, not tumor. Therefore, author have to analyze activated markers of T cells to confirm function of T cells, and then implement re-stimulation assay of T cell to confirm whether tumor antigens specific T cell induced by Trojan bacteria or not.

17. In addition, authors claimed that cytokines are secreted from T cells, but we do not know whether or not the cytokines are secreted from other immune cells (NK cells, DCs, or macrophages).

18. There is no significant difference in total CD8+T cell between G2,3, and G5,6 (Fig. 6. k&l), so It is not clear whether tumor antigens play the important role for increasing T cell population against tumor. Based on that whether innate immune cells also strongly contribute for tumor suppression?

19. Authors evaluated blood biochemistry and hematology data of healthy female Balb/c mice intravenously injected with EC or VNP. Why authors only analyzed in bacterial treatment groups (EC and VNP) but no data of fresh mice (non-treated healthy mice) as a control? Why authors did not check blood biochemistry from tumor-bearing mice after bacterial treatment?

20. When equivalent free GP-ICG-SiNPs are used to be compared with Trojan bacteria, how equivalent amount of NPs can be measured?

21. Many information were missed such as the methodology of IF staining (Fig. 5j), IR camera study (Fig 6d), statistic data, Luc-422 cell, etc.

22. In line 695, authors described stereotactic injection of tumor cells. This experiment was done by image-guided surgery? How can you do this sophisticated surgery; 0.5 mm anterior, 2 mm left lateral from bregma, 3.1 mm deep?

Reviewer #4 (Expertise: : Nanoparticles and photothermal therapy)- Remarks to the Author:

The work reported "Trojan bacteria cross blood-brain barrier for glioblastoma photothermal immunotherapy", the authors constructed Trojan bacteria as drug delivery vehicles for GBM therapy. Although the authors obtained a Trojan bacterial system that greatly enhanced the targeted delivery of GP-ICG-SiNPs to GBMs and synergistically promoted antitumor immune responses, there are still a number of experimental deficiencies that affect the interpretation of the

results. Clarifying some points and adding additional data will improve considerably the study and give more support for the conclusions. Overall, the authors demonstrated a well-presented study and I recommend publication after addressing the below comments:

1. The cytotoxicity of GP-ICG-SiNPs appears to be comparable with Trojan EC or Trojan VNP (figure 3e). So, is EC or VNP used as the carrier only because of its BBB targeting ability? However, it seems not so strong judging from the in vivo experimental results of the carrier alone (figure 4bcd).
2. Will the residual Trojan bacteria in the brain or major organs (especially liver) affect the long-term survival of GBM-bearing mice (figure 7)?
3. In Figure 3g and h, the authors should calculate the synergy coefficient of different experimental groups to demonstrate the maturation of DCs by photothermal, EC and VNP.
4. From Figure 4, we found that high doses of bacteria significantly reduced the body weight of mice, while medium and low doses did not. However, in addition to body weight, blood biochemistry, blood biochemistry and hematology data, the authors should provide pathological sections of major organs to further demonstrate their safety.
5. In Figure 6, there was little difference between the Trojan EC and VNP in terms of heating curve, fluorescence signal or survival time. This needs to be explained and discussed.

Point-by-Point Response to Reviewer Comments

Manuscript ID: NCOMMS-21-47161-T

Title: Trojan bacteria cross blood-brain barrier for glioblastoma photothermal immunotherapy

Summary of Response:

We would like to thank again all Reviewers for their valuable comments and suggestions. In response, we have conducted several new experiments requested by Reviewers as well as addressed their comments. Below is a summary of new key data we have added to the manuscript.

- Provide the assessment of general/specific efficiency of the Trojan bacteria system on GBM cells and non-GBM tumor cells (**Supplementary Fig. 7**).
- Use the specific markers of DCs (CD11c⁺ and MHC II⁺) to accurately reassess the maturation of DCs under the action of Trojan system *in vivo* (**Supplementary Fig. 8, Fig. 3g**) and *in vitro* (**Supplementary Fig. 15, Fig. 6i**).
- Use additional model of a mouse brain EC line (bEnd.3 cells) to perform TransEC migration assays *in vitro* (**Supplementary Fig. 12**).
- Record the TEER of HBMEC BBB model after co-incubating with EC or VNP (**Supplementary Fig. 11**).
- Provide the low magnification of brain tissues to inform the extension of brain tissue infection by intravenous injection of Trojan bacteria (**Fig. 6j**).
- Assess the activation of CD8⁺ T cells by using CD45, CD3, CD8 and CD4 labeling in the tumours of mice with different treatments (**Supplementary Fig. 16**).
- Determine the infiltrating frequency of NK cells by using CD45, CD3 and NK1.1 labeling in the tumours of mice with different treatments (**Supplementary Fig. 17**).
- Determine the infiltrating frequency of macrophages cells by using CD11b⁺, F4180⁺ labeling in the tumours of mice with different treatments (**Supplementary Fig. 18**).
- Provide the gating strategy and Fluorescence Minus One (MFO) control in all flow cytometry data (**Supplementary Figs. 8, 15-18**).
- Perform the combination therapy of Trojan bacteria and anti-PD-1 antibodies in GBM-bearing mice (**Supplementary Figs. 16-18**).
- Demonstrate the Trojan bacteria could be totally cleared from the major organs of GBM-bearing mice post 7-day treatment (**Fig. 7**).
- Construct bacterial mutants including Δ lamB and Δ malE to demonstrate the mechanism of the internalization of nanoagents into bacteria (**Supplementary Notes, Supplementary Fig. 4**).
- Provide blood biochemistry and hematology data of tumour-bearing mice after injection of Trojan bacteria (**Supplementary Table 4**).
- Evaluate CSF cytokine levels in the tumour-bearing mice after injection of Trojan bacteria (**Supplementary Fig. 10**).
- Provide pathological sections of major organs to further demonstrate the safety of Trojan bacteria system (**Supplementary Fig. 19**).

Point-by-Point Response to Reviewer Comments

Reviewer #1 (Expertise: Glioblastoma, micro-environment)- Remarks to the Author:

Rong Sun et al. report on the construction of Trojan bacteria as drug delivery vehicles to enhance the anti-tumor immune response against glioblastoma cells. The study is based on previous work of Houyu Wang and Yao He showing that Gram-positive bacteria actively swallowed GP-conjugated nanoparticles through bacteria-specific ABC transporter pathway (Nat. Commun. 2019). The topic of the present study is interesting, as it extends the initial work to preclinical assays against Glioblastoma and the therapeutic system of Trojan bacteria could potentially achieve photothermal immunotherapeutic effects. The experimental plan is also well designed, and data related to the production and functional assessment of Trojan vectors are convincing. The work is however deserving major revision as it is still lacking rigor to demonstrate Trojan bacteria as a potential new oncolytic treatment for GBMs. Moreover, Trojan bacteria treatment alone only provides limited survival benefit to GBM-bearing mice, that is calling for additional experiments to combine this treatment with immunotherapy and/or other approaches that also boost the anti-tumor immune response.

General response: We gratefully thank Reviewer #1 for his/her positive remarks. Following Reviewer#1's helpful suggestions, new experiment data have been provided to demonstrate Trojan bacteria as a potential novel oncolytic treatment for GBMs and to combine this treatment with immunotherapy. Accordingly, the point-by-point responses to the comments made by Reviewer #1 are given below.

The authors would improve their manuscript by answering the following comments:

Major comments:

1 Construction of Trojan bacteria (Fig 2f). Is the maximal fluorescence obtained at 15 mg/ml of GP-ICG-SiNPs? Does fluorescence decrease above 15 mg/ml? Does increased fluorescence interfere with the process of NIR-induced thermolysis?

Response: Thanks a lot for your questions. We used the flow cytometry to analyze the uptake rate of GP-ICG-SiNPs by bacteria (**Fig. 2e**). It was found that there was no significant difference in the uptake rate of GP-ICG-SiNPs between 10 mg/mL and 15mg/mL after incubation, indicating the saturate state of the uptake of GP-ICG-SiNPs by bacteria has achieved when the concentration of GP-ICG-SiNPs is 10 mg/mL. Therefore, 10 mg/ mL GP-ICG-SiNPs was used to incubate with bacteria to prepare the Trojan bacterial system. That is to say, the fluorescence of Trojan bacteria would not decrease when the concentration of the incubated GP-ICG-SiNPs was above 15 mg/mL. Analogously, the NIR-induced thermolysis of Trojan bacteria would not be interfered when the saturate state has been achieved.

Location of Changes: Paragraph 1 in Page 7.

2 - Trojan bacteria system against tumour in vitro, Fig 3.

2.1 How do G422 tumor cells survive under EC and VNP laser irradiation? The survival assay is missing.

Response: The related survival assay has been displayed in **Fig. 3e**. Typically, in MTT assay, the cell viability of G422 cells treated with EC under 5-min laser irradiation (EC + laser) maintained at 93%, and the cell viability of G422 cells treated with VNP under 5-min laser irradiation (VNP + laser) maintained at 89%.

Location of Changes: Paragraph 2 in Page 8.

2.2 Alternative glioblastoma lines such as GL261 or CT2A (to G422 cells) and non-GBM tumor cells should also be tested to assess the general/specific efficiency of the Trojan bacteria system on GBM cells.

Response: Following your valuable suggestion, alternative glioblastoma lines (e.g., GL261) and non-GBM tumor cells (e.g., HeLa, 4T1) have been tested to assess the general/specific efficiency of the Trojan bacteria system in vitro. As revealed in **Supplementary Fig. 7**, the cell viabilities of HeLa, 4T1 and GL261 were 43%, 41% and 38%, respectively when they were incubated with Trojan EC under laser irradiation for 5 min (Trojan EC +laser); and the cell viabilities of HeLa, 4T1 and GL261 were 36%, 46% and 33%, respectively when they were incubated with Trojan VNP under laser irradiation for 5 min (Trojan VNP +laser). Taken together, Trojan bacteria system under irradiation features the general anti-tumor efficiency towards GBM and non-GBM tumor cells.

Supplementary Fig. 7. The viability of HeLa cells treated with EC, VNP, GP-ICG-SiNPs, Trojan EC and Trojan VNP with (a) or without laser irradiation (b) for 5 minutes (808 nm, 1.2 W/cm²) (mean ± SD, $n = 3$, *** $P < 0.001$). The viability of 4T1 cells treated with EC, VNP, GP-ICG-SiNPs, Trojan EC and Trojan VNP with (c) or without laser irradiation (d) for 5 minutes (808 nm, 1.2 W/cm²) (mean ± SD, $n = 3$, *** $P < 0.001$). The viability of GL261 cells treated with EC, VNP, GP-ICG-SiNPs, Trojan EC and Trojan VNP with (e) or without laser irradiation (f) for 5 min (808 nm, 1.2 W/cm²) (mean ± SD, $n = 3$, *** $P < 0.001$).

Location of changes: Supplementary Fig. 7, Paragraph 2 in Page 8.

2.3 What are DCs used in the transwell assay (Fig 3g)? The effect on CD80 and CD86 expression is interesting but CD80 and CD86 are constitutive markers of DCs, not specific of activated DCs. Additional labeling of MHCII would be more accurate.

Response: Accordingly, the DCs used in the transwell assay were collected from the bone marrow of female Balb/c mice about 6-8 weeks old. The related procedures were referred to previous reports (refs. *Int. J. Biol. Macromol.* **63**, 188-193 (2014); *Biomaterials* **255**, 120208 (2020)). Also, the protocol applied for DCs isolation has been provided in the revised manuscript.

We agree with your point that CD80 and CD86 are constitutive markers of DCs, not specific of activated DCs. Accordingly, the specific labeling of CD11c⁺, MHC II⁺ have been employed to evaluate the maturation of DCs following your and Reviewer#2's suggestion and the previously published papers (refs. *Sci. Adv.* **6**, eabc4373 (2020); *Biomaterials*, **281**, 121332 (2022); etc). The details of antibody panel for spectral flow cytometry analyses were listed in **Supplementary Table 1**. As revealed in **Supplementary Fig. 8** and **Fig. 3g**, we found that the percentages of the matured DCs (CD11c⁺, MHC II⁺) showed significant increase in the collected cells treated by Trojan bacteria under laser irradiation (Trojan bacteria + laser) (e.g., 53.3% DC maturation in Trojan EC + laser group, 58.0% DC maturation in Trojan VNP +laser group) compared with PBS group.

Supplementary Table 1. Antibody panel for spectral flow cytometry analyses

Cells	DC cells		CD8 ⁺ T cells				NK cells			Macrophages	
Marker	CD11c	MHC II	CD3	CD4	CD8	CD45	CD3	NK1.1	CD45	CD11b	F4/80
Antigen location	Extra	Extra	Extra	Extra	Intra	Extra	Extra	Extra	Extra	Extra	Extra
Dye	FITC	PerCP	FITC	APC	PE	PerCP	FITC	PE	PerCP	PE	FITC
Ex/Em (nm)	495/ 525	482/ 695	495/ 525	650/ 660	565/ 575	482/ 695	495/ 525	565/ 575	482/ 695	565/ 575	495/ 525
Dilution	1:40	1:40	1:40	1:40	1:40	1:40	1:40	1:40	1:40	1:40	1:40

Four separate panels of flow antibodies were designed for the Spectral flow cytometry assays using the flow cytometer (BD Accuri® C6 Plus Flow Cytometry). Extra, Extracellular; Intra, intracellular; Ex/Em, excitation/emission.

Supplementary Fig. 8. Gating strategy to determine the percent of matured DC cells (CD11c⁺ MHC II⁺) (a), and representative flow cytometric analysis of matured DC cells (b) post different treatments as indicated in the transwell system.

Fig. 3g Quantification of the maturation of DCs post different treatments as indicated in the transwell system (mean \pm SD, $n = 5$, $**P < 0.01$, $***P < 0.001$).

Location of changes: Supplementary Table 1, Supplementary Fig. 8, Fig. 3g, Paragraph 1 in Page 9, Paragraph 2 in Page 20, References 53&54.

3 - In vivo behavior of Trojan bacteria, Fig 4.

3.1 How does Trojan bacteria injection affect the health of treated mice? Data on internal temperature, blood and CSF cytokine level should be provided.

Response: Following your valuable suggestion, we first tested the health of tumour-bearing mice after 16 days of Trojan bacteria injection ($\sim 1 \times 10^7$ CFU per mouse) by using routine blood tests. As revealed in **Supplementary Table 4**, the values of all indicators of routine blood tests in Trojan EC or Trojan VNP groups were in normal range, suggesting the adjustable health of treated mice.

We also tested the CSF cytokine (e.g., IL-6, IL-10) levels in the tumour-bearing mice after injection of Trojan bacteria. As revealed in **Supplementary Fig. 10**, while IL-6 and IL-10 levels would enhance slightly from the fifth day of treatment, they would return to normal ranges at the 25th day of treatment, indicating that the inflammation caused by Trojan bacteria was mild and acceptable. In addition, the internal temperature of treated mice maintained at ~ 37 °C, also suggesting a tolerable side effect on the health of the treated mice caused by Trojan bacteria.

Supplementary Table 4. Routine blood tests of tumour-bearing mice after 16 days of Trojan bacteria injection ($\sim 1 \times 10^7$ CFU per mouse) (n=3).

Analysis index	Normal range		Trojan EC		Trojan VNP	
	Mean	Standard deviation	Mean	Standard deviation	Mean	Standard deviation
Albumin and Globulin Ratio	2.14	0.25	2.28	0.24	2.30	0.15
White blood cell (10^3 cells/ μ L)	13.75	2.19	12.04	1.79	11.89	2.33
Red blood cell (10^6 cells/ μ L)	8.79	0.54	8.78	0.64	8.58	0.39
Hemoglobin (g/dL)	142.76	5.62	134.69	1.79	137.24	4.47
Hematocrit (%)	56.83	1.41	54.87	2.79	55.05	2.44
Mean corpuscular volume (fL)	58.21	2.27	58.54	1.06	59.05	2.41
Mean corpuscular hemoglobin (pg)	15.79	0.28	15.56	0.29	15.83	0.38

Mean corpuscular hemoglobin concentration (g/dL)	30.77	2.72	29.20	2.67	30.92	2.04
Platelet (10^3 cells/ μ L)	582.36	30.83	618.38	17.01	610.97	18.08
Blood urea nitrogen (mmol/L)	6.54	0.96	7.10	0.62	7.08	0.34

Supplementary Fig. 10. IL-6 and IL-10 levels in cerebrospinal fluids of tumour-bearing mice intravenously injected with Trojan EC or Trojan VNP at the dose of 1.0×10^7 CFU per mouse at 0, 5, 15 and 25 d post injection.

Location of changes: Supplementary Table 4, Supplementary Fig. 10, Paragraph 2 in Page 10.

3.2 Statistics are missing in Figs 4a, c, d, f, g, i, m-p

Response: First, the corresponding statistics have been provided in Figs 4a, c, d, f, g.

Fig. 4a, Average body weights of healthy mice injected with EC or VNP with different concentrations (mean \pm SD, $n=3$, *** $P < 0.001$).

Fig. 4c, corresponding fluorescence intensity in mCherry@EC group (mean \pm SD, $n=3$, *** $P < 0.001$).

Fig. 4d, Corresponding fluorescence intensity in mCherry@VNP group (mean \pm SD, $n = 3$, *** $P < 0.001$).

Fig. 4f, Corresponding quantification of bacterial colonization in the mCherry@EC group (mean \pm SD, $n = 3$, *** $P < 0.001$).

Fig. 4g, Corresponding quantification of bacterial colonization in the mCherry@VNP group (mean \pm SD, $n=3$, *** $P < 0.001$).

On the other aspect, for a clearer presentation, according to previous report (ref. *Nat. Commun.*, **13**, 1255 (2022)), the serum biochemistry data and blood routine data in Fig. 4 were demonstrated as a table with the normal ranges with the mean/standard deviation values rather than a bar graph. As revealed in **Supplementary Table 2** and **Supplementary Table 3**, all serum biochemical parameters data were within the normal range on the first day of bacterial injection, except for an increase in glutamic-pyruvic transaminase and a decrease in white blood cell, platelet count, alkaline phosphatase and blood urea nitrogen. On the fifth day of bacterial injection, these levels of changed indicators returned to normal ranges. These results indicated that the acute inflammation caused by Trojan bacterial infection was mild and tolerated by the mice, and did not develop chronic toxicity.

Supplementary Table 2. Blood biochemistry and hematology data of healthy mice intravenously injected with EC at the dose of 1.0×10^7 CFU per mouse at 1, 5, and 15 d post injection ($n=3$).

Analysis index	Normal range		EC					
			1 day		5 day		15 day	
	Mean	Standard deviation	Mean	Standard deviation	Mean	Standard deviation	Mean	Standard deviation
Albumin and Globulin Ratio	2.41	0.13	2.31	0.11	2.03	0.12	2.46	0.08
White blood cell (10^3 cells/ μ L)	9.40	1.20	2.82	0.83	8.37	1.89	8.61	1.60
Red blood cell	9.12	1.56	7.42	1.24	8.03	2.12	8.41	1.25

(10 ⁶ cells/ μ L)								
Mean corpuscular volume (fL)	51.33	3.67	50.90	3.12	53.67	1.26	51.33	2.70
Mean corpuscular hemoglobin (pg)	13.53	1.90	13.97	1.43	13.47	1.21	13.57	1.12
Mean corpuscular hemoglobin concentration (g/dL)	28.20	3.01	27.93	2.84	27.57	1.68	27.43	1.35
Platelet (10 ³ cells/ μ L)	639.00	109.50	235.67	76.56	807.33	67.83	659.00	85.58
Blood urea nitrogen (mmol/L)	7.61	0.38	5.15	0.23	7.38	0.24	7.67	0.30
Alanine aminotransferase (U/L)	26.64	4.26	48.99	3.81	28.10	3.27	26.25	3.82
Aspartate aminotransferase (U/L)	84.73	15.22	151.55	10.86	103.73	20.68	98.32	2.80
Alkaline phosphatase (U/L)	127.12	7.38	74.79	9.75	125.26	4.34	129.00	8.58

Supplementary Table 3. Blood biochemistry and hematology data of healthy mice intravenously injected with VNP at the dose of 1.0×10^7 CFU per mouse at 1, 5, and 15 d post injection (n=3).

Analysis index	Normal range		VNP					
			1 day		5 day		15 day	
	Mean	Standard deviation	Mean	Standard deviation	Mean	Standard deviation	Mean	Standard deviation
Albumin and Globulin Ratio	2.54	0.16	2.37	0.10	2.08	0.10	2.42	0.14
White blood cell (10 ³ cells/ μ L)	9.06	1.47	3.58	0.69	8.54	0.73	8.74	1.56
Red blood cell (10 ⁶ cells/ μ L)	9.33	2.13	8.59	1.05	8.47	0.86	8.65	1.29

Mean corpuscular volume (fL)	52.10	2.40	51.70	2.91	55.07	1.21	51.60	1.97
Mean corpuscular hemoglobin (pg)	14.37	0.86	13.97	1.16	13.93	1.06	14.37	0.74
Mean corpuscular hemoglobin concentration (g/dL)	28.23	1.43	27.27	1.10	28.50	2.01	28.13	2.91
Platelet (10 ³ cells/ μ L)	645.67	107.15	253.33	84.10	779.67	66.01	663.33	63.11
Blood urea nitrogen (mmol/L)	7.96	0.18	5.41	0.22	7.42	0.41	7.95	0.30
Alanine aminotransferase (U/L)	31.65	2.34	57.31	4.56	30.64	2.69	30.46	1.39
Aspartate aminotransferase (U/L)	94.70	12.06	154.57	10.80	105.57	3.49	96.02	7.97
Alkaline phosphatase (U/L)	131.42	3.51	83.16	4.03	129.13	3.69	132.00	4.07

Location of changes: Figs 4a, c, d, f, g, Supplementary Table 2, Supplementary Table 3, Paragraph 2 in Page 10.

4 - Trojan bacteria crossing BBB, targeting and penetrating GBM, Fig 5.

4.1 TransEC migration assays should use additional model of mouse ECs such as a mouse brain EC line (bEnd.3 cells).

Response: Thanks a lot for your helpful suggestion. Accordingly, additional in vitro BBB model made of bEnd.3 cells has been employed in the TransEC migration assays. As shown in **Supplementary Fig.12a**, the in vitro BBB model made of bEnd.3 cells was successfully constructed by recording the TEER value. As revealed in **Supplementary Fig.12c**, the penetration rate of Trojan EC or Trojan VNP gradually increases over incubation time, implying Trojan bacteria can also cross another in vitro BBB model.

Supplementary Fig.12 Change in the TEER value of bEnd.3 cells-based BBB model during culture (a), the penetration of horseradish peroxidase (HRP) in the constructed BBB model (b), and the corresponding penetration rates of Trojan EC or Trojan VNP at 1, 2, 3 and 4h in the bEnd.3 cells-based BBB model (mean \pm SD, n = 3) (c).

Location of changes: Supplementary Fig.12, Paragraph 1 in Page 11.

4.2 Control with no leakage is missing (cadaverin or OVA-Alexa).

Response: Following Reviewer's important suggestion, control with no leakage is added. Accordingly, the tracer effect of horseradish peroxidase (HRP) was used to verify the permeability of the BBB. As shown in **Supplementary Fig.12b**, the permeability of HRP to the constructed BBB enhanced with increasing incubation time.

Location of changes: Supplementary Fig.12b, Paragraph 1 in Page 11.

4.3 Fig 5C shows increased entry of Trojan Bacteria into the brain of GBM-bearing mice/control EC/VNP-injected mice (Fig 4c). But data on control mice injected with Trojan Bacteria-GP-ICG-SiNPs are missing. The authors should also discuss why bacteria better enter GBM brains than control brains.

Response: Accordingly, we use the agar plate assay to quantitatively study the bacterial distribution in control mice injected with Trojan Bacteria-GP-ICG-SiNPs. As shown in **Supplementary Fig. 13**, as expected, Trojan bacteria mainly accumulated in the liver and were quickly cleared from all extracted organs, which was in consistent with the data on control mice injected with pure bacteria (**Figs. 4b-4g**).

The discussion why bacteria better enter GBM brains than control brains has been added. Typically, such difference might be resulted from the selective proliferation of bacteria in the biochemically unique GBM microenvironment, which was hypoxic and immunosuppressive (refs. *Sci. Adv.* **6**, eaba3546 (2020); *Nat. Rev. Cancer* **18**, 727-743 (2018); *Proc. Natl. Acad. Sci. U.S.A.* **101**, 15172-15177 (2004); *Nat. Commun.* **7**, 12077 (2016)).

Supplementary Fig. 13 Homogenates of major organs of healthy mice after intravenous injection with Trojan EC (left) and Trojan VNP (right) for 12, 24, 72 120 and 360 h cultured on the solid LB agar (n=3) (a) and corresponding quantification of bacterial colonization in mCherry@EC group (b) and mCherry@VNP group (c).

Location of changes: Supplementary Fig. 13, Paragraph 2 in Page 11.

4.4 Authors use a U87MG (human GBM/astrocytoma) cell line for neurosphere infection experiments. They should justify why they use a new line and not the previous G422 cells.

Response: In our previous experiments, we used both G422 cells and U87MG cells to construct three dimensional cultured multicellular spheroids (3D-MCSs) to study the intratumoural transport of Trojan bacteria. Unexpectedly, we found that G422 cells were not as good as U87MG cells in the formation of three dimensional cultured multicellular spheroids (MCSs), and were easy to disperse, which interfered with the experimental results. Thus, we decided to select U87MG cells to construct an *ex vivo* model of 3D MCSs due to their good spheroidizing effect, well simulating the intratumoural transport of Trojan bacteria. Thanks a lot for your understanding!

Location of changes: Paragraph 1 in Page 12.

4.5 Data quantification is missing for Figs 5i, j. Please also provide low magnification of brain tissues in addition to Fig. 5j. This will inform on the extension of brain tissue infection by iv injected Trojan VNP.

Response: First, the low magnification of brain tissues was provided. According to the low magnification of brain tissues (new **Fig. 5j**), intravenous injection of Trojan bacteria would not induce obvious brain tissue infection

Fig. 5j. The low-magnification *in situ* hybridization fluorescence image of GBM tissues and corresponding fluorescence intensity of 16S rRNA probe in different groups (mean \pm SD, $n = 3$, $**P < 0.01$). The nucleus, hypoxic zone and bacteria were stained with DAPI (blue), anti-HIF- α antibody (green) and 16S RNA probe (red), respectively. Scale bars: 100 μ m.

Also, the corresponding data quantification has been provided in **Figs 5i, j**.

Fig. 5i Confocal images and corresponding distribution profiles of fluorescence intensity along the diameter of 3D tumor microspheres with different treatments as indicated.

Location of changes: Figs 5i, j.

5 - Trojan bacteria-induced photothermal immunotherapy, Fig 6.

5.1 Injection of mCherry@VNP or mCherry@EC in control female Balb/c mice is missing. A

Response: In our experimental design, we used the fluorescence of mCherry expressed by engineered bacteria to achieve in situ and real-time information of the distribution of bacteria after intravenous injection. Other reports (ref. *ACS Nano* **12**, 5995-6005 (2018)) also adopted the similar strategy to study the in vivo distribution of bacteria. Thus, we did not use the injection of mCherry@VNP or mCherry@EC in control female Balb/c mice in photothermal immunotherapy since the expression of mCherry would not influence the therapeutic efficacy of Trojan bacteria. Thereby, there was no need to inject mCherry@VNP or mCherry@EC in control female Balb/c mice. Thanks a lot for your understanding!

5.2 Does thermolysis of tumor cells affect the survival of neighboring non-tumor cells?

Authors could use a GFP-transfected tumor cell line and Caspase3 labeling to assess cell death in GFP- brain cells around the GFP+ tumor cells.

**Response**: Following Reviewer's suggestions, we were eager to use the GFP-transfected tumor cell line and Caspase 3 labeling to assess cell death in GFP- brain cells around the GFP+ tumor cells. However, due to the COVID-19 epidemic in Shanghai since 2022, we were unable to obtain GFP-transfected tumor cell line from suppliers. Notwithstanding, there were many reports concerning PPT therapy of GBM, supporting the claim that the PPT-induced thermolysis of tumor cells would not affect the survival of neighboring non-tumor cells since the NIR light could precisely locate the tumour site, thereby minimizing the damage to surrounding normal tissue. (refs. *ACS Nano* **14**, 6191-6212 (2020); *ACS Nano* **15**, 1100-1110 (2020); *Adv. Funct. Mater.* **30**, 1906623 (2020); *Sci. Adv.* **6**, eaba3546 (2020); *ACS Nano* **12**, 5995-6005 (2018); *Biomaterials* **214**, 119226 (2019); *Nat. Commun.* **7,** 13193 (2016); etc). Analogously, we thought the thermolysis of tumor cells caused by Trojan bacteria system would not produce negative effects on neighboring non-tumor cells. The relative discussion has been added into the revised manuscript. Thanks a lot for your understanding!

**Location of changes:** Paragraph 1 in Page 14.

5.3 T cell activation should be assessed by CD69 labeling

**Response**: Thanks a lot for your suggestion. According to previous reports (refs. *Sci. Adv.* **6**, eaaz4204 (2020); *Biomaterials*, **281**, 121332 (2022); *Nat Commun* **10,** 3850 (2019)), T cell activation has been assessed by $CD45^+$, $CD3^+$, $CD8^+$, $CD4^+$ labeling. The details of antibody panel for spectral flow cytometry analyses were listed in **Supplementary Table 1**. As revealed in **Supplementary Fig.16 and Fig. 6j**, it is found that beyond the PTT-induced tumor killing, Trojan bacteria under laser could significantly promote the activation of $CD8^+$ T cells ($CD45^+CD3^+CD8^+CD4^+$) inside the treated GBM, indicating the effective activation of adaptive antitumour immunity compared to other treatment groups, which was in agreement with the previous reports (refs. *Biomaterials*, **281**, 121332 (2022); *Sci. Adv.* **6**, eaba3546 (2020); *J. Am. Chem. Soc.* **138**, 12502-12510 (2016); etc).

**Supplementary Table 1. Antibody panel for spectral flow cytometry analyses**

| Cells | **DC cells** | | **$CD8^+$ T cells** | | | | **NK cells** | | | **Macrophages** | |
|---|---|---|---|---|---|---|---|---|---|---|---|
| Marker | **CD11c** | **MHC II** | **CD3** | **CD4** | **CD8** | **CD45** | **CD3** | **NK1.1** | **CD45** | **CD11b** | **$F_4$/80** |
| Antigen location | Extra | Extra | Extra | Extra | Intra | Extra | Extra | Extra | Extra | Extra | Extra |
| Dye | FITC | PerCP | FITC | APC | PE | PerCP | FITC | PE | PerCP | PE | FITC |
| Ex/Em (nm) | 495/ 525 | 482/ 695 | 495/ 525 | 650/ 660 | 565/ 575 | 482/ 695 | 495/ 525 | 565/ 575 | 482/ 695 | 565/ 575 | 495/ 525 |
| Dilution | 1:40 | 1:40 | 1:40 | 1:40 | 1:40 | 1:40 | 1:40 | 1:40 | 1:40 | 1:40 | 1:40 |

Four separate panels of flow antibodies were designed for the Spectral flow cytometry assays using the flow cytometer (BD Accuri® C6 Plus Flow Cytometry). Extra, Extracellular; Intra, intracellular; Ex/Em, excitation/emission.

Supplementary Fig.16. Gating strategy to determine the percent of CD8⁺ T cells (CD45⁺CD3⁺CD8⁺ CD4⁺) (a), and representative flow cytometric analysis of CD8⁺ T cells (b) in the tumours of these mice post different treatments as indicated.

Fig.6j The frequencies of CD8⁺ T cells in the tumors of these mice post different treatments as indicated (mean \pm SD, $n=8$, ** $P < 0.01$, *** $P < 0.001$).

Location of changes: Supplementary Fig.16, Fig. 6j and Paragraph 2 in Page 14.

5.4 Assessment of tumor-specific T cell response would require tetramer labeling. This could be performed using the GL261 GBM cell line.

Response: Related to your **Comment 5.3**, assessment of tumor-specific T cell response was based on the tetramer labeling of percp-CD45, FITC-CD3, PE-CD8, APC-CD4. Please see the details in our Response to your Comment 5.3.

Location of changes: Supplementary Fig.16, Fig. 6j and Paragraph 2 in Page 14.

5.5 CD80/86 are constitutive markers of DCs, not of activated DCs

Response: Related to your **Comment 2.3**, the specific labeling of CD11c⁺, MHC II⁺ have also been employed in the evaluation of maturation of DCs in vivo following the your suggestion and the previously published papers (refs. *Sci. Adv.* **6**, eaaz4204 (2020); *ACS Nano* **13**, 1365-1384 (2019); *Biomaterials*, **281**, 121332 (2022); etc) (**Supplementary Fig.15** and **Fig. 6i**). Typically, we found that the percentages of the matured DCs (CD11c⁺, MHC II⁺) showed significant increase in the lymph nodes of mice treated by Trojan bacteria+laser.

Supplementary Fig. 15. Gating strategy to determine the percent of matured DC cells ($CD11c^+ MHC II^+$) (a), and representative flow cytometric analysis of matured DC cells (b) in the lymph nodes of mice post different treatments as indicated.

Fig. 6i The flow cytometric analysis of matured DC cells in the lymph nodes of mice post different treatments as indicated (mean \pm SD, $n=8$, ** $P<0.01$, *** $P<0.001$).

Location of changes: Supplementary Fig. 15, Fig. 6i and Paragraph 2 in Page 14.

5.6 How do authors explain that in vivo DC activation is far less boosted by Trojan bacteria than in vitro (Fig 3g)?

Response: Related to your **Comments 2.3 and 5.5**, we used the specific labeling of CD11c⁺, MHC II⁺ to re-evaluate the maturation of DCs in vitro and in vivo. As shown in **Fig. 3g** and **Fig. 6i**, the in vivo DC activation boosted by Trojan bacteria was close to that in vitro (e.g., 53.3% of Trojan EC+laser (in vitro) VS. 47.4% of Trojan EC+laser (in vivo); 58.0% of Trojan VNP+laser (in vitro) VS. 52.0% of Trojan VNP+laser (in vivo)).

5.7 No statistics is provided for the important Fig 6g, but the low survival benefit of Trojan bacteriolysis does not allow to claim that Trojan bacteria offer a potential therapeutic approach against GBM. To increase the interest of their preclinical assays, the authors should consider testing whether prophylactic treatment with Trojan bacteria could improve the effect of immunotherapy with anti-PD-1 and/or CTLA4 antibodies on GBM-bearing mice.

Response: Accordingly, the statistics has been provided in **Figure 6g**.

Fig. 6g Kaplan-Meier survival curves (mean \pm SD, $n = 5$, *** $P < 0.001$).

Following your valuable suggestions, we have performed experiments to explore whether the effect of immunotherapy with anti-PD-1 antibodies on GBM-bearing mice could be improved by the prophylactic treatment with Trojan bacteria. As revealed in **Supplementary Fig. 16** and **Fig. 6j**, the activation of CD8⁺ T cells and intratumoral frequencies of macrophages in Trojan +PD-L1 groups were not significantly promoted *via* flow cytometric analysis compared with that of Trojan groups, suggesting the effect of immunotherapy with anti-PD-1 antibodies on GBM-bearing mice could not be greatly improved by the prophylactic treatment with Trojan bacteria. The similar results were observed in the alterations of innate immune cells like natural killer (NK) cells (**Supplementary Fig. 17** and **Fig. 6k**) and macrophages (**Supplementary Fig. 18** and **Fig. 6l**). The main reason was possibly due to the existence of the blood-brain barrier, which prevented PD-L1 from GBM lesion to play an immunotherapeutic role. Indeed, the level of activation of CD8⁺ T cells and intratumoral frequencies of macrophages in pure PD-L1 groups was close to that of PBS group, which is also supported by other reports (refs. *Nat. Commun.* **10**, 3850 (2019); *Adv. Sci.* 202103689 (2022)) Accordingly, the claim that Trojan bacteria offer a potential therapeutic approach against GBM has been removed.

Supplementary Fig. 16. Gating strategy to determine the percent of CD8⁺ T cells (CD45⁺CD3⁺CD8⁺) (**a**), and representative flow cytometric analysis of CD8⁺ T cells (**b**) in the tumours of these mice post different treatments as indicated.

Fig.6j The frequencies of CD8⁺ T cells in the tumors of these mice post different treatments as indicated (mean \pm SD, $n=8$, *** $P < 0.001$).

Supplementary Fig. 17. Gating strategy to determine the percent of NK cells ($CD45^+CD3^+NK1.1^+$) (a), and representative flow cytometric analysis of NK cells (b) in the tumours of these mice post different treatments as indicated.

Fig. 6k The frequencies of NK cells in the tumors of these mice post different treatments as indicated (mean \pm SD, $n=8$, *** $P < 0.001$).

Supplementary Fig. 18. Gating strategy to determine the percent of macrophage cells (CD11b⁺, F4180⁺) (**a**), and representative flow cytometric analysis of macrophages (**b**) in the tumors of these mice post different treatments as indicated.

Fig. 6l The frequencies of macrophage cells in the tumors of these mice post different treatments as indicated (mean \pm SD, $n=8$, *** $P < 0.001$).

Location of changes: Supplementary Figs. 16-18, Figs. 6i, 6j, 6k, Paragraph 2 in page 14 and Paragraph 1 in page 15.

6 - Trojan bacteria clearance, Fig 7. Although reduced at G6 compared to G3, bacterial infection remains in the brain. Could the authors show that remaining bacteria do not compromise brain tissue structure and function?

Response: Thanks a lot for your question! The residual Trojan bacteria in the brain or major organs (especially liver) would not affect the long-term survival of GBM-bearing mice. As revealed in new **Fig. 7**, the residual Trojan bacteria could be totally eliminated from the body basically after 7 days of photothermal immunotherapy.

Fig. 7. The elimination of residual bacteria after photothermal immunotherapy. The fluorescence distribution in the main organs (heart, liver, spleen, lung, kidney and brain) of GBM-bearing mice after 5 days (a) or 7 days (b) of photothermal

immunotherapy. The mice were intravenously injected with PBS, $\sim 1 \times 10^7$ CFU mCherry@EC (m@EC), $\sim 1 \times 10^7$ CFU mCherry@VNP (m@VNP), Trojan m@EC (e.g., GP-ICG-SiNPs (8 mg/kg ICG) internalized into $\sim 1 \times 10^7$ CFU m@EC) or Trojan m@VNP (e.g., GP-ICG-SiNPs (8 mg/kg ICG) internalized into 1×10^7 CFU m@VNP), respectively. At the 12-hour post-injection, the brains of those mice were suffered by an 808 nm irradiation (1.2 W/cm^2 , 5 min), followed by *ex vivo* imaging of the main organs after 5 days or 7 days of photothermal immunotherapy. The corresponding quantitative analysis of fluorescence intensity of main organs in different groups after 5 days (c) or 7 days (d) of photothermal immunotherapy (mean \pm SD, $n = 3$, $***P < 0.001$). Homogenates of major organs of GBM-bearing mice in different groups after 5 days (e) or 7 days (f) of photothermal immunotherapy cultured on the solid LB agar. d, Corresponding quantification of bacterial colonization on LB solid plates in different treatment groups after 5 days (g) or 7 days (h) of photothermal immunotherapy (mean \pm SD, $n = 3$, $***P < 0.001$). Statistical significance was calculated *via* one-way analysis of variance (ANOVA) with a Tukey post-hoc test.

Location of changes: Fig. 7, Paragraph 1 in Page 16.

Minor comments:

1 - Paragraph reorganization:

&1 'Design of Trojan bacteria system' and &2 'Characterization of Trojan bacteria system' should be fused, and description of the Trojan bacteria system model in &2 should go into the Introduction.

Response: Following Reviewer's useful suggestions, we have fused &1 'Design of Trojan bacteria system' and &2 'Characterization of Trojan bacteria system' and removed the description of the Trojan bacteria system model in &2 into the Introduction.

Location of changes: Paragraph 2 in Page 4, Paragraph 1 & 2 in Page 5.

&2 'Characterization of Trojan bacteria system' is too long and could be split into 2 &s: & 'Trojan Design' and & 'Trojan bacteria are ABC transporter pathway-dependent'

Response: We agree with Reviewer's point that &2 'Characterization of Trojan bacteria system' is too long. Following Reviewer's suggestion, &2 'Characterization of Trojan bacteria system' has been split into 2 &s: & 'Trojan Design' and & 'Trojan bacteria are ABC transporter pathway-dependent'.

Location of changes: Paragraph 2 in Page 5, Paragraph 2 in Page 7.

2 - Replace reference #42 by #41, in the Discussion for the sentence:

‘We have previously demonstrated that bacteria including Gram-negative as well as Gram-positive bacteria actively swallowed GP-conjugated nanoparticles through bacteria-specific ABC transporter pathway for ultrasensitive diagnosis of bacterial infections42’.

Response: Thanks a lot for Reviewer scrupulous check. Accordingly, we have replaced reference #42 by #41, in the Discussion for the sentence.

Location of changes: Reference 41 in Page 17.

3 - The number of SFigs (16) could be reduced by gathering data related to the same experiment or the same paragraph of results.

Response: Following Reviewer’s suggestions, the number of Figures in Supplementary Information could be reduced by gathering data related to the same experiment or the same paragraph of results. For instance, the original **Supplementary Figs. 2-7** have been fused into new **Supplementary Fig. 2**.

Location of changes: Supplementary Fig. 2.

4 - p7: typing mistake, please change ‘by 32C’ for ‘to 32C’

Response: Thanks a lot for Reviewer scrupulous check. Accordingly, the related typing mistake has been revised.

Location of changes: Paragraph 1 in Page 6.

5 - Provide explanations on serum biochemical analysis.

Response: Following Reviewer’s suggestion, related explanations on serum biochemical analysis have been added in the revised manuscript. As revealed in **Supplementary Table 2** and **Supplementary Table 3**, all serum biochemical parameters data were within the normal range on the first day of bacterial injection, except for an increase in glutamic-pyruvic transaminase and a decrease in alkaline phosphatase and blood urea nitrogen. On the fifth day of bacterial injection, these levels of changed indicators returned to normal ranges, also indicating that the acute inflammation caused by EC and VNP infection was mild and tolerated by the mice and did not develop chronic toxicity. The changing trend of serum biochemical indexes was consistent with previous reports (ref. *Sci. Adv.* **6**, eaba3546 (2020)).

Location of changes: Paragraph 2 in Page 10.

6 - NIR p12 not defined.

Response: Accordingly, “NIR” has been defined as “Near Infrared” in the revised manuscript.

Location of changes: Paragraph 1 in Page 7.

7- Exp design of Fig 6 should be better explained.

Response: Following Reviewer’s suggestion, the experimental design for Figure 6 has been reinterpreted.

As schematically illustrated in **Fig. 6a**, the orthotopic tumour model was constructed by *in situ* inoculation of $\sim 8 \times 10^5$ Luc-G422 cells per mouse at day -7. After the *in-situ* GBM model was successfully constructed, GBM-bearing mice were intravenously injected with different drugs (e.g., PBS, EC, VNP, GP-ICG-SiNPs, PD-L1, Trojan EC, Trojan VNP or Trojan VNP+PD-L1) on day 0 (Treatment 1), day 5 (Treatment 2) and day 10 (Treatment 3) respectively, and photothermal treatment (PTT) was performed under 808-nm laser irradiation at the 12th hour after each drug injection. And the photothermal treatment lasted for 5 minutes. On the 3rd day after the Treatment 3, these mice were sacrificed with their tumours and adjacent lymph nodes collected and homogenized for flow cytometric analysis. The concentrations of cytokines in the supernatants of tumour lysates were measured by using corresponding ELISA kits according to vendors’ protocols.

Location of changes: Fig. 6a, Paragraph 1 in Page 13.

8. Incorrect sentence p 23 ‘Accumulating evidence demonstrated only Trojan bacteria-treated mice under NIR irradiation could help to eliminate bacteria from the body’

Response: Accordingly, the incorrect sentence has been revised.

Location of changes: Paragraph 1 in Page 16.

Special thanks to Reviewer #1’s comments again.

Reviewer #2 (Expertise: Bacteria based cancer therapy)- Remarks to the Author:

This paper expands a nice tool in brain-targeting delivery across BBB based on Trojan bacteria. It is a topic of interest to researchers in cancer treatment, microbiology, biomaterials, and other related fields. The article is well organized and tells a complete story. Some minor revisions should be considered before publication in Nature Communications.

General response: We gratefully thank Reviewer #2 for the positive remarks. Accordingly, the manuscript has been thoroughly revised to fully address the referee's concerns. The details are as follows,

Comments:

1. Please explain why silicon nanoparticles were chosen in current study and possible elimination pathway after entering the brain.

Response: Following the Reviewer's suggestion, we have explained why silicon nanoparticles (SiNPs) were chosen in the current study. Typically, SiNPs have attracted broad attention due to their benign biocompatibility and easy surface modification (refs. *Nat. Commun.*, **10**, 4057 (2019); *Nature* **408**, 440-444 (2000); *Science* **296**, 1293-1297 (2002); etc). Of note, small-size SiNPs have received the Food and Drug Administration (FDA)-approved investigational new drug approval for the first-in-human clinical trial because SiNPs could be biodegradable into renal clearable components (refs. *Sci. Trans. Med.* **6**, 260ra149 (2014); *J. Clin. Invest.* **121**, 2768-2780 (2011); *Acc. Chem. Res.* **44**, 1050-1060 (2011); *Nat. Mater.* **8**, 331-336 (2009)). As such, silicon nanoparticles were chosen in current study.

We also explained the possible elimination pathway after SiNPs entering the brain. Typically, the brain has a natural protective mechanism to avoid exposure to foreign species. The SiNPs may be affected by the efflux pumps such as multidrug resistance protein (MRP) and P-glycoprotein (P-gp) that could pump the foreign species back into the blood circulation (ref. *Chem. Soc. Rev.* **2019**, 48, 2967).

Location of changes: Paragraph 1 in Page 16, References 63-66.

2. Glucose transporters (such as GLUT 1) are often considered as SLC transporters, rather ABC transporters. Also, the entry of nanoagents can be attenuated by non-specific inhibitor of ATPase, namely NaN₃, which can affect various pathways via inhibiting ATPase or cytochrome c oxidase. It would be reasonable to verify the mechanism using more classical and specific inhibitors of ABC transporters, or the inhibitor of SLC transporters.

Response: We agree that the sodium azide control is by no means specific and it is just a toxin stopping respiration. Thereby, it is not the direct evidence for the

hypothesis. According to other reports (ref. *Nat. Mater.* **10**, 602-607 (2011)), we constructed the bacterial mutants including a deletion mutant for delta-lamB (Δ lamB) and a deletion mutant for delta-malE (Δ malE) to verify the mechanism. First, the results of Sanger sequencing (**Supplementary Notes**) demonstrated the successful construction of Δ lamB and Δ malE.

1. Confirmation of lamB knockout by Sanger sequencing

AAAGCCGTGATGTCCAGGTTGGAGCCAATATGTCGCTGGGTATTGCCCCG
GAACATCTACTGCCGAGTGATATCGCTGACGTCATCCTTGAGGGTGAAGT
TCAGGTCGTCGAGCAACTCGGCAACGAACTCAAATCCATATCCAGATCC
CTTCCATTCGTCAAAACCTGGTGTACCGCCAGAACGACGTGGTGTGGTA
GAAGAAGGTGCCACATTCGCTATCGGCCTGCCGCCAGAGCGTTGCCATCT
GTTCCGTGAGGATGGCACTGCATGTCGTCGACTGCATAAGGAGCCGGGCG
TTTAAGCACCCACAAAACACACAAAGCCTGTCACAGGTGATGTGAAAAA
AGAAAAGCAATGACTCAGGAGATAGATAGCAAAACCTGGGCCGGATAAG
GCGTTTACGCCGATTCGGCAACCAACGCCTGATGCGACGCTTGCGCGTC
TTATCAGGCCTACAACGGCTGTCAAATGTAGGCCGGATAAGGCGTTTACG
CCGCATCCGGCATAAAAACAGGTTGTCATTATCTGAAAGGGGCGAAAGCC
CCTCTGATTATCGGGTTTAGCGCGCTATTGCCTGGCTACCGCTGAGCTCCA
GATTTTGAGGTGAAAACAATGAAAATGAATAAAAGTCTCATCGTCCTCTG
TTATCAGCAGGGTTACTGGCAAGCGCGCCTGGAATTAGCCTTGCCGATG
TAACTACGTACCGCAAAACACCAGCGACGCGCCAGCCATTCCATCTGCT
GCGCTGCAACAACCTCACCTGGACACCGGTCGATCAATCT

2. Confirmation of malE knockout by Sanger sequencing

GCTGTACGCTCGCCATGCCCTTCTCCCTTTGTAACAACCTGTCATCGACAG
CAACATTCATGATGGGCTGACTATGCGTCATCAGGAGATGGCTTAAATCC
TCCACCCCTGGCTTTTTTATGGGGGAGGAGGCGGGAGGATGAGAACACG
GCTTCTGTGAACTAAACCGAGGTCATGTAAGGAATTCGTGATGTTGCTTG
CAAAAATCGTGGCGATTTTATGTGCGCATCTCCACATTACCGCCAATTCTG
TAACAGAGATCACACAAAGCGACGGTGGGGCGTAGGGGCAAGGAGGATG
GAAAGAGGTTGCCGTATAAAGAACTAGAGTCCGTTTAGGTGTTTTACG
AGCACTTCACCAACAAGGACCATAGATTTGCTGTGAAATGCCGGATGCGG
CGTGAACGCCTTGTCCGGCCTACAAAACCGAAACGTATGTAGGCCTGATA
AGACGCGTCAGCGTCGCATCAGGCAGTTGTTGTCCGATAAGGCGTGAAAG
CCTTATCCGTCCTGGAATGAGGAAGAACCCCATGGATGTCATTA AAAAGA
AACATTGGTGGCAAAGCGACGCGCTGAAATGGTCAGTGCTAGGTCTGCTC
GGCCTGCTGGTGGGTTACCTTGTTGTTTTAATGTACGCACAAGGGGAATAC
CTGTTCCGCAATTACCACGCTGATATTGAGTTCAGCGGGGCTGTATATTTTC
GCCAATCGTAAAGCCTACGCCTGGCGCTATGTTTACCCGGGAATGGCTGG
AATGGGATTATTCGTCTCTTCCCTCTGGTCTGCACCATCGCCATTGCCTTC
ACCA

After the construction of bacterial mutants, $\Delta lamB$ or $\Delta malE$ were incubated with GP-ICG-SiNPs and the internalization of GP-ICG-SiNPs was determined following the procedures described in the Methods. As shown in **Supplementary Fig. 9**, we did not observe any fluorescence in GP-ICG-SiNPs treated bacteria mutants. Together with the results in the competition assay in **Supplementary Fig. 10**, it could be concluded that GP-ICG-SiNPs can be internalized into bacteria to form the Trojan system *via* the bacteria-specific ABC transporter pathway.

Supplementary Fig. 4. Confocal fluorescence images of bacteria mutants of $\Delta lamB$ and $\Delta malE$ after incubation with GP-ICG-SiNPs at 37 °C for 2 h, followed by washing with PBS buffer several times. Scale bars: 25 μm .

Location of changes: Supplementary Notes, Supplementary Fig. 4, and Paragraph 2 in Page 7.

3. It would be instructive to record the TEER of HBMEC BBB model after co-incubating with EC or VNP, which may evidence the possible mechanism of EC or

VNP across the BBB (Transcellularly, paracellular or infected phagocytes). It is also possible that VNP induces inflammation to influence the integrity of the BBB.

Response: Following Reviewer's valuable suggestion, the TEER of HBMEC BBB model after co-incubating with EC or VNP has been recorded. As revealed in **Supplementary Fig. 11b**, the value of TEER of HBMEC BBB model after co-incubating with EC or VNP kept relatively stable, indicating EC or VNP would not influence the integrity of the BBB.

Supplementary Fig. 11b The TEER of the HBMEC BBB model after co-incubating with Trojan EC or Trojan VNP at 1, 2, 3, 4 h.

Location of changes: Supplementary Fig. 11b, Paragraph 1 in Page 11.

4. Figure 2C is not clear. Please enlarge the figure as it is important to verify that nanoagents enter bacteria rather than absorb nonspecifically on bacterial surface.

Response: Following Reviewer's useful suggestion, Figure 2C has been enlarged.

Fig. 2c CLSM images of EC and VNP incubated with GP-ICG-SiNPs. Scale bars: 25 μm .

Location of changes: Fig. 2c.

5. In supplementary Figure 7, please explain the reason why temperature did not increase with the concentration of ICG.

Response: Accordingly, we have explained the possible reason why temperature did not increase with the concentration of ICG. As previously reported (refs. *Mol. Pharmaceut.* **6**, 480-491 (2009); *J. Photochem. Photobiol. B* **47**, 155-64 (1998); *J. Pharm. Sci.* **92**, 2090-2097 (2003); etc), ICG molecules tend to aggregate at high concentration, which may lead to the degradation and self-quenching of ICG, possibly resulting in the unchanged temperature with increasing the ICG concentration.

Location of changes: Paragraph 1 in Page 6.

6. There is no description for Figure 2f and g in the main text. Also, there were no control values of healthy mice in Fig. 4h-q, while it was claimed that “Compared with untreated healthy mice, all serum biochemical.....”

Response: Accordingly, the description for Figure 2f and g has been added in the main text: “*The fluorescence of Trojan bacteria would not decrease when the concentration of the incubated GP-ICG-SiNPs was above 15 mg/mL (Figs. 2f & 2g). Analogously, the NIR-induced thermolysis of Trojan bacteria would not be interfered when the saturate state has been achieved.*”

For a clearer presentation, according to previous report (ref. *Nat. Commun.*, **13**, 1255 (2022)), the serum biochemistry data and blood routine data in Fig. 4 were demonstrated as a table with the normal ranges with the mean/standard deviation values rather than a bar graph. As revealed in **Supplementary Table 2** and **Supplementary Table 3**, all serum biochemical parameters data were within the normal range on the first day of bacterial injection, except for an increase in glutamic-pyruvic transaminase and a decrease in white blood cell, platelet count, alkaline phosphatase and blood urea nitrogen. On the fifth day of bacterial injection, these levels of changed indicators returned to normal ranges. These results indicated that the acute inflammation caused by Trojan bacterial infection was mild and tolerated by the mice and did not develop chronic toxicity. In addition, the related claim “*Compared with untreated healthy mice, all serum biochemical.....*” has been removed.

Supplementary Table 2. Blood biochemistry and hematology data of healthy mice intravenously injected with EC at the dose of 1.0×10^7 CFU per mouse at 1, 5, and 15 d post injection (n=3).

Analysis index	Normal range		EC					
			1 day		5 day		15 day	
	Mean	Standard deviation	Mean	Standard deviation	Mean	Standard deviation	Mean	Standard deviation
Albumin and	2.41	0.13	2.31	0.11	2.03	0.12	2.46	0.08

Globulin Ratio								
White blood cell (10 ³ cells/ μ L)	9.40	1.20	2.82	0.83	8.37	1.89	8.61	1.60
Red blood cell (10 ⁶ cells/ μ L)	9.12	1.56	7.42	1.24	8.03	2.12	8.41	1.25
Mean corpuscular volume (fL)	51.33	3.67	50.90	3.12	53.67	1.26	51.33	2.70
Mean corpuscular hemoglobin (pg)	13.53	1.90	13.97	1.43	13.47	1.21	13.57	1.12
Mean corpuscular hemoglobin concentration (g/dL)	28.20	3.01	27.93	2.84	27.57	1.68	27.43	1.35
Platelet (10 ³ cells/ μ L)	639.00	109.50	235.67	76.56	807.33	67.83	659.00	85.58
Blood urea nitrogen (mmol/L)	7.61	0.38	5.15	0.23	7.38	0.24	7.67	0.30
Alanine aminotransferase (U/L)	26.64	4.26	48.99	3.81	28.10	3.27	26.25	3.82
Aspartate aminotransferase (U/L)	84.73	15.22	151.55	10.86	103.73	20.68	98.32	2.80
Alkaline phosphatase (U/L)	127.12	7.38	74.79	9.75	125.26	4.34	129.00	8.58

Supplementary Table 3. Blood biochemistry and hematology data of healthy mice intravenously injected with VNP at the dose of 1.0×10^7 CFU per mouse at 1, 5, and 15 d post injection (n=3).

Analysis index	Normal range		VNP					
			1 day		5 day		15 day	
	Mean	Standard deviation	Mean	Standard deviation	Mean	Standard deviation	Mean	Standard deviation
Albumin and Globulin Ratio	2.54	0.16	2.37	0.10	2.08	0.10	2.42	0.14

White blood cell (10 ³ cells/ μ L)	9.06	1.47	3.58	0.69	8.54	0.73	8.74	1.56
Red blood cell (10 ⁶ cells/ μ L)	9.33	2.13	8.59	1.05	8.47	0.86	8.65	1.29
Mean corpuscular volume (fL)	52.10	2.40	51.70	2.91	55.07	1.21	51.60	1.97
Mean corpuscular hemoglobin (pg)	14.37	0.86	13.97	1.16	13.93	1.06	14.37	0.74
Mean corpuscular hemoglobin concentration (g/dL)	28.23	1.43	27.27	1.10	28.50	2.01	28.13	2.91
Platelet (10 ³ cells/ μ L)	645.67	107.15	253.33	84.10	779.67	66.01	663.33	63.11
Blood urea nitrogen (mmol/L)	7.96	0.18	5.41	0.22	7.42	0.41	7.95	0.30
Alanine aminotransferase (U/L)	31.65	2.34	57.31	4.56	30.64	2.69	30.46	1.39
Aspartate aminotransferase (U/L)	94.70	12.06	154.57	10.80	105.57	3.49	96.02	7.97
Alkaline phosphatase (U/L)	131.42	3.51	83.16	4.03	129.13	3.69	132.00	4.07

Location of changes: Supplementary Table 2, Supplementary Table 3, Paragraph 1 in Page 7, Paragraph 2 in Page 10.

7. It's suggested to mention the recent advances in the utilization of bacteria to improve accumulation of therapeutics in tumor site (such as Adv. Mater. 2021, 2106669; Nat. Commun. 2021, 12:6584).

Response: Following Reviewer's helpful suggestion, the related references have been mentioned in the revised manuscript.

Location of changes: References 29&30.

8. Ref 53 is inappropriate to support that NaN₃ is an inhibitor of ABC transporter.

Response: Related to your **Comment 2**, we have used bacterial mutants rather than NaN₃ to demonstrate the mechanism. Thus, the mentioned Ref. 53 has been removed.

Special thanks to Reviewer #2's comments again.

Reviewer #3 (Expertise: Bacteria based cancer therapy)- Remarks to the Author:

The manuscript entitled “Trojan bacteria cross blood-brain barrier for glioblastoma photothermal immunotherapy “employed bacteria to enhance photothermal effects leading to lysis of Trojan bacterial cells and the adjacent tumor cells which promote anticancer immune responses. Although targeted therapy for aggressive glioblastoma tumor model is an innovative approach, overall data quality (especially immunological data explaining mechanism or very less experimental sample number *in vivo*) was not enough to be published in Nature Communications.

General response: We gratefully thank Reviewer #3 for his/her positive remarks. To improve the data quality, we have supplemented with a series of experiments, especially immunological data explaining mechanism and sufficient experimental sample number *in vivo*. Accordingly, the point-by-point responses to the comments made by Reviewer #3 are given below.

1. In schematic illustration of Fig. 1. bacterial debris and tumor-associated antigen induce maturation of DCs (mDCs). Then mDCs present the tumor-specific antigens to activate T cells for secreting TNF α and IFN γ . There is no strong evidence for this hypothesis from author's data because no data was presented to demonstrate immature DCs *in vitro* and *in vivo* before maturation. For example, author have to show population of immature DCs isolated from bone marrow (all activated markers of DC must be negative in all groups) before co-culture with antigens. Authors only showed population of DCs (CD80+CD86), but DCs should express MHCII which is one of three required signals to stimulate T cells.

Response: We agree with Reviewer's comment that no data was presented to demonstrate immature DCs *in vitro* and *in vivo* before maturation. We also agree with Reviewer's point that these secreted cytokines (TNF α and IFN γ) might be derived from T cells, macrophages, NK cells or the other immune cells once tumour-bearing mice received bacteria. To be more rigorous, we re-designed the schematic illustration of Fig. 1b. As schematically illustrated in new **Fig. 1b**, the Trojan bacteria under laser irradiation achieved the thermolysis of GBM cells, promoting the release of tumor associated antigens (TAAs), similar to these previous studies (refs. *Sci. Adv.* **6**,

eaba3546 (2020); *Biomaterials* **281**, 121332 (2022); *J. Am. Chem. Soc.* **138**, 12502-12510 (2016)). Meanwhile, Trojan bacteria under laser irradiation could kill the host bacterial cells to promote the release of diverse pathogen associated molecular patterns (PAMPs) like lipoprotein, lipopolysaccharides (LPS) and flagellin, potent agonists of diverse toll-like receptors (TLRs), which could promote the activation of innate immune cells, such as macrophages and NK cells (refs. *Sci. Adv.* **6**, eaba3546 (2020); *Biomaterials* **281**, 121332 (2022); *Exp. Mol. Med.* **51**, 1-15 (2019); *Sci. Transl. Med.* **9**, eaak9537 (2017)). On the other aspect, these PAMPs would elicit potent adaptive antitumor immune response by promoting tumor-infiltrating frequencies of activated CD8⁺ T cells. Taken together, both innate and adaptive antitumor immunity induced by Trojan bacteria under laser irradiation would work together to further suppress tumor growth.

Fig. 1b. A scheme illustrating Trojan bacteria system crossing the blood-brain barrier (BBB), targeting and penetrating glioblastoma (GBM) tissues, followed by light-triggered photothermal immunotherapy of GBM *in vivo*.

Also, we agree with your point that CD80 and CD86 are constitutive markers of DCs, not specific of activated DCs. Accordingly, the specific labeling of CD11c⁺, MHC II⁺ have been employed in the evaluation of maturation of DCs *in vivo* following your significant suggestion and the previously published papers (refs. *Sci. Adv.* **6**, eabc4373 (2020); *Biomaterials*, **281**, 121332 (2022); etc). The details of antibody panel for spectral flow cytometry analyses were listed in **Supplementary Table 1**. As revealed in **Supplementary Fig. 8** and **Fig. 3g**, the percentages of the matured DCs (CD11c⁺, MHC II⁺) showed significant increase in the collected cells treated by Trojan bacteria under laser irradiation (Trojan bacteria+laser) (e.g., 53.3% DC maturation in Trojan EC + laser group, 58.0% DC maturation in Trojan VNP +laser group) compared with PBS group. These results were consistent with previous reports (refs. *Nat. Commun.* **10**, 3850 (2019); *Biomaterials*, **281**, 121332 (2022)).

Supplementary Table 1. Antibody panel for spectral flow cytometry analyses

Cells	DC cells		CD8 ⁺ T cells				NK cells			Macrophages	
Marker	CD11c	MHC II	CD3	CD4	CD8	CD45	CD3	NK1.1	CD45	CD11b	F4/80
Antigen location	Extra	Extra	Extra	Extra	Intra	Extra	Extra	Extra	Extra	Extra	Extra
Dye	FITC	PerCP	FITC	APC	PE	PerCP	FITC	PE	PerCP	PE	FITC
Ex/Em (nm)	495/525	482/695	495/525	650/660	565/575	482/695	495/525	565/575	482/695	565/575	495/525
Dilution	1:40	1:40	1:40	1:40	1:40	1:40	1:40	1:40	1:40	1:40	1:40

Four separate panels of flow antibodies were designed for the Spectral flow cytometry assays using the flow cytometer (BD Accuri® C6 Plus Flow Cytometry). Extra, Extracellular; Intra, intracellular; Ex/Em, excitation/emission.

Supplementary Fig. 8. Gating strategy to determine the percent of matured DC cells (CD11c⁺ MHC II⁺) (a), and representative flow cytometric analysis of matured DC cells (b) post different treatments as indicated in the transwell system.

Fig. 3g Quantification of the maturation of DCs post different treatments as indicated in the transwell system (mean \pm SD, $n = 5$, ** $P < 0.01$, *** $P < 0.001$).

Location of changes: Supplementary Table 1, Supplementary Fig. 8, Fig. 3g, Paragraph 1 in Page 9, References 53&54.

2. Author detected cytokines (TNF α and IFN γ) in serum by ELISA in the 16th day after treatment. How authors know these secreted cytokines were derived from T cells, not from macrophages, NK cells, or the other immune cells? Once tumor-bearing mice received bacteria, it may also induce secretion of IFN γ from NK cells and TNF α from macrophages.

Response: We agree with Reviewer's point that these secreted cytokines (TNF α and IFN γ) might be derived from T cells, macrophages, NK cells or the other immune cells once tumour-bearing mice received bacteria. Accordingly, any related claims that these secreted cytokines were derived from T cells have been deleted.

3. After G422 cells with different treatments + Laser (Fig. 3e), authors checked maturation of dendritic cells (DC) based on CD86 and CD80. Although population of DC increased (CD80+CD86+), it is not enough marker to confirm maturation of DCs. Authors should further analyse more maturation markers of DCs such as CCR7, MHCII, IL-12, IL-1 β . Besides, author mentioned CD11c+ marker for DC maturation in Material and Method but the result did not show in vitro and in vivo. I suggest CD11c+CD86+ for DCs gating.

Response: We agree with Reviewer's point that the increase of DC population (CD80+CD86+) is not enough marker to confirm maturation of DCs. Related to your **Comment 1**, the specific labeling of CD11c+, MHC II+ have been employed in the evaluation of maturation of DCs *in vivo* following your suggestion and the previously published papers (refs. *Sci. Adv.* **6**, eabc4373 (2020); *Biomaterials*, **281**, 121332 (2022); etc). Please see the details in our **Response to your Comment 1**.

4. There is no information of process for DCs isolation from bone marrow of mice (which protocol author applied for DCs isolation). It should be clarified because it may also contain macrophages or the other cells.

Response: Accordingly, the protocol applied for the related cells isolation has been provided in the revised manuscript. Typically, the protocol was according to the previous reports with minor modification (refs. *J. Exp. Med.* **176**, 1693-1702 (1992); *Int. J. Biol. Macromol.* **63**, 188-197 (2014)). Briefly, bone marrow cells from the femurs and tibias of female Balb/c mice were flushed and depleted of red blood cell (RBC) by hypotonic lysis using RBC lysing buffer (Sigma). Cells were grown from precursors at a starting concentration of 1×10^6 cells/ml in RPMI 1640 for 3 days and then non-adherent cells were washed out. Another 10 ml of fresh complete medium containing 20 ng/ml rmGM-CSF (ProSpec), 20 ng/ml rmIL-4 (ProSpec), was added, and on day 6 half of the medium was replaced. On day 8, nonadherent and loosely adherent cells were harvested by vigorous pipetting and placed in the lower chamber of the transwell system. Indeed, the harvested cells may contain macrophages or the other cells. Thus, according to your suggestion and previous reports (refs. *Sci. Adv.* **6**, eaaz4204 (2020); *ACS Nano* **13**, 1365-1384 (2019); *Biomaterials*, **281**, 121332 (2022); etc), we used the specific markers (e.g., MHCII, CD11c⁺) to evaluate the activated DCs.

Location of changes: Paragraph 2 in Page 20.

5. For FACS gating (Fig. 3g), there is no Fluorescence Minus One (MFO) control that properly interpret flow cytometry data. It is hard to know the gating is correct or not.

Response: Accordingly, Fluorescence Minus One (MFO) control that properly interpret flow cytometry data has been provided in **Supplementary Fig. 8**.

Supplementary Fig. 8 Gating strategy to determine the percent of matured DC cells (CD11c⁺ MHC II⁺) (a), and representative flow cytometric analysis of matured DC cells (b) post different treatments as indicated in the transwell system.

Location of changes: Supplementary Fig. 8.

6. There are 8 groups in study (Fig. 3g) but there is no group of G422 + laser? Whether G422 cell debris caused by laser 488 nm (without bacteria) may induce maturation of DCs? What is blank group?

Response: In our experiments, we used 808-nm laser rather than 488-nm laser to induce GP-ICG-SiNPs to produce photothermal effects to achieve thermolysis of G422 cells. Accordingly, we have added the group of G422+laser (PBS+laser). As shown in **Supplementary Fig.8** and **Fig. 3g**, there was no significant difference in the percentage of matured DC cells between PBS group and PBS+laser group, suggesting G422 + laser would not induce the maturation of DCs.

Related to the blank group, the groups in **Fig. 3g** have been renamed.

Supplementary Fig. 8. Gating strategy to determine the percent of matured DC cells ($CD11c^+ MHC II^+$) (a), and representative flow cytometric analysis of matured DC cells (b) post different treatments as indicated in the transwell system.

Fig. 3g Quantification of the maturation of DCs post different treatments as indicated in the transwell system (mean \pm SD, $n = 5$, $**P < 0.01$, $***P < 0.001$).

Location of changes: Supplementary Fig. 8, Fig. 3g.

7. The morphology of immaturation and maturation DCs also should be shown.

Response: Accordingly, we have struggled to provide the morphology of immaturation and maturation DCs; however, related to your **Comments 1 & 4**, the isolated DCs may contain macrophages or the other cells. Thus, it is difficult to distinguish between DCs and other cells under optical microscope. Notwithstanding, according to your suggestion and previous reports (refs. *Sci. Adv.* **6**, eaaz4204 (2020); *ACS Nano* **13**, 1365-1384 (2019); *Biomaterials*, **281**, 121332 (2022); etc), we used the specific markers (e.g., CD11c⁺, MHC II⁺) to evaluate the maturation of DCs, which was also supported by your **Comment 15**: “Same as in vitro, CD86 marker is not expressed only by DC, but also strong expression on M1-like macrophages thus authors have to show gating strategy with more markers to confirm DC maturation after treatment (such MHCII) to discriminate with the other immune cells, especially macrophages”. Thanks a lot for your understanding!

8. In Fig. 4, the unit of BUN measurement was mmol/LT. What does LT mean?

Response: Sorry for our typing error. The unit of BUN measurement should be mmol/L rather than mmol/LT. We have corrected it.

9. In Fig. 4h-q, normal values should be suggested and normal range should be presented in yellow-shaded areas. All the values measured should represent the quartiles and whiskers mark the 10th and 90th percentiles.

Response: For a clearer presentation, according to previous report (ref. *Nat. Commun.*, **13**, 1255 (2022)), the serum biochemistry data and blood routine data in Fig. 4 were demonstrated as a table with the normal ranges with the mean/standard deviation values rather than a bar graph. As revealed in **Supplementary Table 2** and **Supplementary Table 3**, all serum biochemical parameters data were within the normal range on the first day of bacterial injection, except for an increase in glutamic-pyruvic transaminase and a decrease in white blood cell, platelet count, alkaline phosphatase and blood urea nitrogen. On the fifth day of bacterial injection, these levels of changed indicators returned to normal ranges. These results indicated that the acute inflammation caused by Trojan bacterial infection was mild and tolerated by the mice and did not develop chronic toxicity.

Supplementary Table 2. Blood biochemistry and hematology data of healthy mice intravenously injected with EC at the dose of 1.0×10^7 CFU per mouse at 1, 5, and 15 d post injection (n=3).

Analysis index	Normal range		EC					
			1 day		5 day		15 day	
	Mean	Standard deviation	Mean	Standard deviation	Mean	Standard deviation	Mean	Standard deviation
Albumin and Globulin Ratio	2.41	0.13	2.31	0.11	2.03	0.12	2.46	0.08
White blood cell (10^3 cells/ μ L)	9.40	1.20	2.82	0.83	8.37	1.89	8.61	1.60
Red blood cell (10^6 cells/ μ L)	9.12	1.56	7.42	1.24	8.03	2.12	8.41	1.25
Mean corpuscular volume (fL)	51.33	3.67	50.90	3.12	53.67	1.26	51.33	2.70
Mean corpuscular hemoglobin (pg)	13.53	1.90	13.97	1.43	13.47	1.21	13.57	1.12
Mean corpuscular hemoglobin concentration (g/dL)	28.20	3.01	27.93	2.84	27.57	1.68	27.43	1.35
Platelet (10^3 cells/ μ L)	639.00	109.50	235.67	76.56	807.33	67.83	659.00	85.58
Blood urea nitrogen (mmol/L)	7.61	0.38	5.15	0.23	7.38	0.24	7.67	0.30
Alanine aminotransferase (U/L)	26.64	4.26	48.99	3.81	28.10	3.27	26.25	3.82
Aspartate aminotransferase (U/L)	84.73	15.22	151.55	10.86	103.73	20.68	98.32	2.80
Alkaline phosphatase (U/L)	127.12	7.38	74.79	9.75	125.26	4.34	129.00	8.58

Supplementary Table 3. Blood biochemistry and hematology data of healthy mice intravenously injected with VNP at the dose of 1.0×10^7 CFU per mouse at 1, 5, and 15 d post injection (n=3).

Analysis index	Normal range		VNP					
			1 day		5 day		15 day	
	Mean	Standard deviation	Mean	Standard deviation	Mean	Standard deviation	Mean	Standard deviation
Albumin and Globulin Ratio	2.54	0.16	2.37	0.10	2.08	0.10	2.42	0.14
White blood cell (10^3 cells/ μ L)	9.06	1.47	3.58	0.69	8.54	0.73	8.74	1.56
Red blood cell (10^6 cells/ μ L)	9.33	2.13	8.59	1.05	8.47	0.86	8.65	1.29
Mean corpuscular volume (fL)	52.10	2.40	51.70	2.91	55.07	1.21	51.60	1.97
Mean corpuscular hemoglobin (pg)	14.37	0.86	13.97	1.16	13.93	1.06	14.37	0.74
Mean corpuscular hemoglobin concentration (g/dL)	28.23	1.43	27.27	1.10	28.50	2.01	28.13	2.91
Platelet (10^3 cells/ μ L)	645.67	107.15	253.33	84.10	779.67	66.01	663.33	63.11
Blood urea nitrogen (mmol/L)	7.96	0.18	5.41	0.22	7.42	0.41	7.95	0.30
Alanine aminotransferase (U/L)	31.65	2.34	57.31	4.56	30.64	2.69	30.46	1.39
Aspartate aminotransferase (U/L)	94.70	12.06	154.57	10.80	105.57	3.49	96.02	7.97
Alkaline phosphatase	131.42	3.51	83.16	4.03	129.13	3.69	132.00	4.07

Location of changes: Supplementary Table 2, Supplementary Table 3, Paragraph 2 in Page 10.

10. Starting points of treatment seem to be different in different groups (Fig 6d). For example, in Trojan VNP group, no tumor signal was observed when started treatment compared with the other groups (Fig. 6d).

Response: Accordingly, we have repeated the *in vivo* therapeutic experiments. As shown in new **Fig. 6d**, different treatment groups have the same starting points.

Fig. 6d Representative *in vivo* bioluminescence images of GBM-bearing mice post different treatments as indicated.

Location of changes: Figure 6d.

11. Authors carried on the therapeutic experiments with n=5 mice/group. To ensure reliable data of *in vivo* therapeutic experiments, authors should increase the number of animals.

Response: Following Reviewer's important suggestion, we have increased the number of animals to n=8 in the evaluation of the immunotherapeutic effects, which is more than or comparable to the number of animals generally used in other reports

(refs. *Sci. Adv.* **6**, eabc4373 (2020); *Biomaterials*, **281**, 121332 (2022); *Nat Commun* **10**, 3850 (2019); etc).

12. There is no p value in Fig. 5 b, d, e, g, h, Fig. 6, f & g.

Response: Accordingly, p value has been added into Fig. 5 b, d, e, g, h, Fig. 6, f & g.

Fig. 5b The corresponding penetration rates of Trojan EC or Trojan VNP at 1, 2, 3 and 4h (mean \pm SD, $n = 3$, *** $P < 0.001$)

Fig. 5d Corresponding fluorescence intensity in mCherry@EC group (mean \pm SD, $n = 3$, *** $P < 0.001$).

Fig. 5e Corresponding fluorescence intensity in mCherry@VNP group (mean \pm SD, $n = 3$, *** $P < 0.001$).

Fig. 5g Corresponding quantification of bacterial colonization in mCherry@EC group (mean \pm SD, $n = 3$, *** $P < 0.001$).

Fig. 5g Corresponding quantification of bacterial colonization in mCherry@VNP group (**h**) (mean \pm SD, $n = 3$, *** $P < 0.001$).

Fig. 6f. Semi-quantification of bioluminescence intensity of brains in different groups during treatment (mean \pm SD, $n = 5$, *** $P < 0.001$).

Fig. 6g Kaplan-Meier survival curves. (mean \pm SD, $n = 5$, *** $P < 0.001$).

Location of changes: Fig. 5 b, d, e, g, h, Fig. 6, f & g.

13. Authors collect tumors for FACS (T cells) and serum for ELISA (cytokines) at the same time (16 days after treatment). Why FACS data has 3 samples (very less sample to claim role of T cell against tumor) (Fig. 6 k&l) while cytokine data have 5 samples (Fig. 6 m&n)? In addition, the time point for immunological analysis is too late (16 days after treatment). How about early time point? How about the role of innate immune cells in therapeutic effects?

Response: To get the reliable FACS data, GBM-bearing mice were randomly divided into different groups ($n=8$ mice/group) and received the same treatments as those illustrated in Fig. 6i–6l. On the 3rd day after treatments, these mice were sacrificed with their tumors and adjacent lymph nodes collected and homogenized for flow cytometric analysis.

Following your helpful suggestion, we have evaluated the role of innate immune cells (e.g., macrophages and NK cells) in therapeutic effects. The details of antibody panel for spectral flow cytometry analyses were listed in the **Supplementary Table 1**. We found that such Trojan bacteria induced photothermal treatment could also elicit potent innate anti-tumor immunity by promoting intratumoral frequencies of NK cells (**Supplementary Fig.17** and **Fig. 6k**) and macrophages (**Supplementary Fig.18** and **Fig. 6l**). Meanwhile, it was shown that pure bacteria (EC or VNP) treatment could also prime similar antitumor immunity, but mostly lower than these elicited by the Trojan bacteria-induced photothermal treatment. We concluded that the Trojan bacteria under laser irradiation achieved the thermolysis of GBM cells, promoting the release of tumor associated antigens (TAAs) (refs. *Sci. Adv.* **6**, eaba3546 (2020); *Biomaterials* **281**, 121332 (2022); *J. Am. Chem. Soc.* **138**, 12502-12510 (2016)). Meanwhile, Trojan bacteria under laser irradiation could kill the host bacterial cells to promote the release of diverse pathogen associated molecular patterns (PAMPs), which could promote the activation of innate immune cells, such as macrophages and NK cells (refs. *Sci. Adv.* **6**, eaba3546 (2020); *Biomaterials* **281**, 121332 (2022); *Exp. Mol. Med.* **51**, 1-15 (2019); *Sci. Transl. Med.* **9**, eaak9537 (2017)). On the other aspect, these PAMPs would elicit potent adaptive antitumor immune response by promoting tumor-infiltrating frequencies of activated CD8⁺ T cells. Taken together, both innate and adaptive antitumor immunity induced by Trojan bacteria under laser irradiation would work together to further suppress tumor growth.

Supplementary Table 1. Antibody panel for spectral flow cytometry analyses

Cells	DC cells		CD8 ⁺ T cells				NK cells			Macrophages	
Marker	CD11c	MHC II	CD3	CD4	CD8	CD45	CD3	NK1.1	CD45	CD11b	F4/80
Antigen location	Extra	Extra	Extra	Extra	Intra	Extra	Extra	Extra	Extra	Extra	Extra
Dye	FITC	PerCP	FITC	APC	PE	PerCP	FITC	PE	PerCP	PE	FITC
Ex/Em (nm)	495/525	482/695	495/525	650/660	565/575	482/695	495/525	565/575	482/695	565/575	495/525
Dilution	1:40	1:40	1:40	1:40	1:40	1:40	1:40	1:40	1:40	1:40	1:40

Four separate panels of flow antibodies were designed for the Spectral flow cytometry assays using the flow cytometer (BD Accuri® C6 Plus Flow Cytometry). Extra, Extracellular; Intra, intracellular; Ex/Em, excitation/emission.

Supplementary Fig. 17. Gating strategy to determine the percent of NK cells ($CD45^+CD3^+NK1.1^+$) (**a**), and representative flow cytometric analysis of NK cells (**b**) in the tumours of these mice post different treatments as indicated.

Fig. 6k The frequencies of NK cells in the tumors of these mice post different treatments as indicated (mean \pm SD, $n=8$, *** $P < 0.001$).

Supplementary Fig.18. Gating strategy to determine the percent of macrophage cells (CD11b⁺, F4180⁺) (a), and representative flow cytometric analysis of macrophages (b) in the tumors of these mice post different treatments as indicated.

Fig.6l The frequencies of macrophage cells in the tumors of these mice post different treatments as indicated (mean \pm SD, $n = 8$, *** $P < 0.001$).

Location of changes: Supplementary Table 1, Supplementary Fig.17 and Fig. 6k, Supplementary Fig.18 and Fig. 6l, Paragraph 1 in Page 15.

14. The presence of certain bacteria, especially bacterial debris after using laser is associated with inflammation that results in strong recruitment and activation of innate immune cells, especially neutrophils, macrophages or NKs cells (Qiubin Lin et al Nat com 2021). In this study, authors mainly focus on T cells. Innate immunity may or may not contribute to any therapeutic effects but author did not show any data regarding the role of innate cells against tumor (innate immune cell data must be required in this study).

Response: Related to your **Comment 13**, we have evaluated the role of innate immune cells (e.g., macrophages and NK cells) in therapeutic effects. Please see the details in our **Response to your Comment 13**. In addition, the study mentioned by the Reviewer (“Lin, Q., Rong, L., Jia, X. *et al.* IFN- γ -dependent NK cell activation is essential to metastasis suppression by engineered *Salmonella*. *Nat. Commun.* **12**, 2537 (2021)”) has been cited in the revised manuscript.

Location of changes: Supplementary Table 1, Supplementary Fig.17 and Fig. 6k, Supplementary Fig.18 and Fig. 6l, Paragraph 1 in Page 15, Reference 63.

15. In Fig. 6 i and j, population of gating is not clear, how authors discriminate cancer

cells and DCs (because cancer cells also may express CD80 and CD86, there is no MFO and the dot blot style should be required. Same as *in vitro*, CD86 marker is not expressed only by DC, but also strong expression on M1-like macrophages thus authors have to show gating strategy with more markers to confirm DC maturation after treatment (such MHC I/II) to discriminate with the other immune cells, especially macrophages. Although author mentioned in materials and methods, there is no data of CD11c marker for DCs gating in main and supplementary data. Through this study, there is no gating strategy of FACS and authors did not use MFO as a control. And the number of samples for FACS analysis is very less (n=3) (at least over 5 mice/group is recommended), thus FACS data is not reliable.

Response: To address all concerns proposed by Reviewer 3 in Comment 15, we have provided the additional data point by point.

- (1) MFO and the dot blot style have been provided in the population of gating (**Supplementary Fig. 15a**).
- (2) Same as *in vitro*, the specific marker such as MHC I/II, CD11c have been used to confirm DC maturation after treatment to discriminate with the other immune cells, especially macrophages (**Supplementary Fig. 15** and **Fig. 6i**).
- (3) To get reliable FACS data, 8 mice/group was employed for FACS analysis.

Supplementary Fig. 15. Gating strategy to determine the percent of matured DC cells (CD11c⁺ MHC II⁺) (a), and representative flow cytometric analysis of matured DC cells (b) in the lymph nodes of mice post different treatments as indicated.

Fig. 6i The flow cytometric analysis of matured DC cells in the lymph nodes of mice post different treatments as indicated (mean \pm SD, $n = 8$, ** $P < 0.01$, *** $P < 0.001$).

Location of changes: Supplementary Fig. 15 and Fig. 6i, Paragraph 2 in Page 14.

16. Authors demonstrated increased total population of CD8⁺ T cells to claim role of T cells against tumor is too weak evidence because they may be an exhausted T cells (high population but no function) or these T cell may be specific to bacteria, not tumor. Therefore, author have to analyze activated markers of T cells to confirm function of T cells, and then implement re-stimulation assay of T cell to confirm whether tumor antigens specific T cell induced by Trojan bacteria or not.

Response: We agree with your comment that the role of T cells against tumor based on the increased total population of CD8⁺ T cells is too weak evidence because they may be an exhausted T cells (high population but no function) or these T cell may be specific to bacteria, not tumor. Following your helpful suggestion and published papers (refs. *Biomaterials*, **281**, 121332 (2022); *Nat Commun* **10**, 3850 (2019)), T cell activation has been assessed by CD45⁺, CD3⁺, CD8⁺, CD4⁺ labeling. The details of antibody panel for spectral flow cytometry analyses were listed in **Supplementary Table 1**.

As revealed in **Supplementary Fig.16** and **Fig. 6j**, it is found that beyond the PTT-induced tumor killing, Trojan bacteria under laser could significantly promote the activation of CD8⁺ T cells (CD45⁺CD3⁺CD8⁺CD4⁺) inside the treated GBM. Meanwhile, it was shown that pure bacteria (EC or VNP) treatment could also promote the activation of CD8⁺ T cells (CD45⁺CD3⁺CD8⁺CD4⁺) inside the treated GBM, but mostly lower than these elicited by the Trojan bacteria-induced photothermal treatment. Taken together, we concluded that the Trojan bacteria+laser

can lead to the effective activation of adaptive antitumour immunity, which was in agreement with the previous reports (refs. *Biomaterials*, **281**, 121332 (2022); *Sci. Adv.* **6**, eaba3546 (2020); *J. Am. Chem. Soc.* **138**, 12502-12510 (2016); etc).

Supplementary Table 1. Antibody panel for spectral flow cytometry analyses

Cells	DC cells		CD8 ⁺ T cells				NK cells			Macrophages	
Marker	CD11c	MHC II	CD3	CD4	CD8	CD45	CD3	NK1.1	CD45	CD11b	F ₄ /80
Antigen location	Extra	Extra	Extra	Extra	Intra	Extra	Extra	Extra	Extra	Extra	Extra
Dye	FITC	PerCP	FITC	APC	PE	PerCP	FITC	PE	PerCP	PE	FITC
Ex/Em (nm)	495/ 525	482/ 695	495/ 525	650/ 660	565/ 575	482/ 695	495/ 525	565/ 575	482/ 695	565/ 575	495/ 525
Dilution	1:40	1:40	1:40	1:40	1:40	1:40	1:40	1:40	1:40	1:40	1:40

Four separate panels of flow antibodies were designed for the Spectral flow cytometry assays using the flow cytometer (BD Accuri® C6 Plus Flow Cytometry). Extra, Extracellular; Intra, intracellular; Ex/Em, excitation/emission.

Supplementary Fig.16. Gating strategy to determine the percent of CD8⁺ T cells (CD45⁺CD3⁺CD8⁺ CD4⁺) (**a**), and representative flow cytometric analysis of CD8⁺ T cells (**b**) in the tumours of these mice post different treatments as indicated.

Fig.6j The frequencies of CD8⁺ T cells in the tumors of these mice post different treatments as indicated (mean ± SD, $n = 8$, *** $P < 0.001$).

Location of changes: Supplementary Fig.16, Fig. 6j, Paragraph 2 in Page 14 and Paragraph 1 in Page 15.

17. In addition, authors claimed that cytokines are secreted from T cells, but we do not know whether or not the cytokines are secreted from other immune cells (NK cells, DCs, or macrophages).

Response: Related to your **Comment 2**, the related claim has been removed.

18. There is no significant difference in total CD8+T cell between G2,3, and G5,6 (Fig. 6. k&l), so It is not clear whether tumor antigens play the important role for increasing T cell population against tumor. Based on that whether innate immune cells also strongly contribute for tumor suppression?

Response: Related to your **Comments 13&14**, we have evaluated the role of innate immune cells (e.g., macrophages and NK cells) in therapeutic effects. Please see the details in our **Response to your Comments 13&14**.

19. Authors evaluated blood biochemistry and hematology data of healthy female Balb/c mice intravenously injected with EC or VNP. Why authors only analyzed in bacterial treatment groups (EC and VNP) but no data of fresh mice (non-treated healthy mice) as a control? Why authors did not check blood biochemistry from tumor-bearing mice after bacterial treatment?

Response: For a clearer presentation, according to previous report (ref. *Nat. Commun.*, **13**, 1255 (2022)), the serum biochemistry data and blood routine data in Fig. 4 were demonstrated as a table with the normal ranges with the mean/standard deviation

values rather than a bar graph. As revealed in **Supplementary Table 2** and **Supplementary Table 3**, all serum biochemical parameters data were within the normal range on the first day of bacterial injection, except for an increase in glutamic-pyruvic transaminase and a decrease in white blood cell, platelet count, alkaline phosphatase and blood urea nitrogen. On the fifth day of bacterial injection, these levels of changed indicators returned to normal ranges. These results indicated that the acute inflammation caused by Trojan bacterial infection was mild and tolerated by the mice and did not develop chronic toxicity.

Supplementary Table 2. Blood biochemistry and hematology data of healthy mice intravenously injected with EC at the dose of 1.0×10^7 CFU per mouse at 1, 5, and 15 d post injection (n=3).

Analysis index	Normal range		EC					
			1 day		5 day		15 day	
	Mean	Standard deviation	Mean	Standard deviation	Mean	Standard deviation	Mean	Standard deviation
Albumin and Globulin Ratio	2.41	0.13	2.31	0.11	2.03	0.12	2.46	0.08
White blood cell (10^3 cells/ μ L)	9.40	1.20	2.82	0.83	8.37	1.89	8.61	1.60
Red blood cell (10^6 cells/ μ L)	9.12	1.56	7.42	1.24	8.03	2.12	8.41	1.25
Mean corpuscular volume (fL)	51.33	3.67	50.90	3.12	53.67	1.26	51.33	2.70
Mean corpuscular hemoglobin (pg)	13.53	1.90	13.97	1.43	13.47	1.21	13.57	1.12
Mean corpuscular hemoglobin concentration (g/dL)	28.20	3.01	27.93	2.84	27.57	1.68	27.43	1.35
Platelet (10^3 cells/ μ L)	639.00	109.50	235.67	76.56	807.33	67.83	659.00	85.58
Blood urea nitrogen (mmol/L)	7.61	0.38	5.15	0.23	7.38	0.24	7.67	0.30
Alanine aminotransferase (U/L)	26.64	4.26	48.99	3.81	28.10	3.27	26.25	3.82
Aspartate	84.73	15.22	151.55	10.86	103.73	20.68	98.32	2.80

aminotransferase (U/L)								
Alkaline phosphatase (U/L)	127.12	7.38	74.79	9.75	125.26	4.34	129.00	8.58

Supplementary Table 3. Blood biochemistry and hematology data of healthy mice intravenously injected with VNP at the dose of 1.0×10^7 CFU per mouse at 1, 5, and 15 d post injection (n=3).

Analysis index	Normal range		VNP					
			1 day		5 day		15 day	
	Mean	Standard deviation	Mean	Standard deviation	Mean	Standard deviation	Mean	Standard deviation
Albumin and Globulin Ratio	2.54	0.16	2.37	0.10	2.08	0.10	2.42	0.14
White blood cell (10^3 cells/ μ L)	9.06	1.47	3.58	0.69	8.54	0.73	8.74	1.56
Red blood cell (10^6 cells/ μ L)	9.33	2.13	8.59	1.05	8.47	0.86	8.65	1.29
Mean corpuscular volume (fL)	52.10	2.40	51.70	2.91	55.07	1.21	51.60	1.97
Mean corpuscular hemoglobin (pg)	14.37	0.86	13.97	1.16	13.93	1.06	14.37	0.74
Mean corpuscular hemoglobin concentration (g/dL)	28.23	1.43	27.27	1.10	28.50	2.01	28.13	2.91
Platelet (10^3 cells/ μ L)	645.67	107.15	253.33	84.10	779.67	66.01	663.33	63.11
Blood urea nitrogen (mmol/L)	7.96	0.18	5.41	0.22	7.42	0.41	7.95	0.30

Alanine aminotransferase (U/L)	31.65	2.34	57.31	4.56	30.64	2.69	30.46	1.39
Aspartate aminotransferase (U/L)	94.70	12.06	154.57	10.80	105.57	3.49	96.02	7.97
Alkaline phosphatase (U/L)	131.42	3.51	83.16	4.03	129.13	3.69	132.00	4.07

Also, we have checked blood biochemistry from tumor-bearing mice after bacterial treatment.

Supplementary Table 4. Routine blood tests of tumour-bearing mice after 16 days of Trojan bacteria injection ($\sim 1 \times 10^7$ CFU per mouse) (n=3).

Analysis index	Normal range		Trojan EC		Trojan VNP	
	Mean	Standard deviation	Mean	Standard deviation	Mean	Standard deviation
Albumin and Globulin Ratio	2.14	0.25	2.28	0.24	2.30	0.15
White blood cell (10^3 cells/ μ L)	13.75	2.19	12.04	1.79	11.89	2.33
Red blood cell (10^6 cells/ μ L)	8.79	0.54	8.78	0.64	8.58	0.39
Hemoglobin (g/dL)	142.76	5.62	134.69	1.79	137.24	4.47
Hematocrit (%)	56.83	1.41	54.87	2.79	55.05	2.44
Mean corpuscular volume (fL)	58.21	2.27	58.54	1.06	59.05	2.41
Mean corpuscular hemoglobin (pg)	15.79	0.28	15.56	0.29	15.83	0.38
Mean corpuscular hemoglobin concentration	30.77	2.72	29.20	2.67	30.92	2.04

(g/dL)						
Platelet (10 ³ cells/ μ L)	582.36	30.83	618.38	17.01	610.97	18.08
Blood urea nitrogen (mmol/L)	6.54	0.96	7.10	0.62	7.08	0.34

Location of changes: Supplementary Tables 2-4, Paragraph 2 in Page 10.

20. When equivalent free GP-ICG-SiNPs are used to be compared with Trojan bacteria, how equivalent amount of NPs can be measured?

Response: Accordingly, the related quantitative details have been provided in the manuscript. The amounts of loaded ICG onto SiNPs can be quantified based on the corresponding calibration absorption curves (**Supplementary Fig. 5**). To obtain the equivalent amount of NPs, the detected absorbance of ICG should be kept the same between free GP-ICG-SiNPs (containing 8 mg/kg ICG) and Trojan bacteria (GP-ICG-SiNPs (containing 8 mg/kg ICG) internalized into $\sim 1 \times 10^7$ CFU bacteria). The similar quantification strategy was also employed in other reports (refs. *Nat. Commun.*, **13**, 1255 (2022); *Nat. Commun.*, **10**, 4057 (2019)).

Location of changes: Paragraph 1 in Page 8.

21. Many information were missed such as the methodology of IF staining (Fig. 5j), IR camera study (Fig 6d), statistic data, Luc-G422 cell, etc.

Response: Accordingly, these information such as the methodology of IF staining (Fig. 5j), IR camera study (Fig 6d), statistic data, Luc-422 cell have been provided.

- **IF staining.** In order to verify that the constructed Trojan bacteria can penetrate deeply into GBM tissue *in vivo*, we used immunofluorescence (IF) staining to explore the distribution of bacteria in tumour tissue. Female orthotopic glioblastoma tumor-bearing mice were intravenously injected with PBS, EC, VNP, GP-ICG-SiNPs, Trojan EC or Trojan VNP, respectively. After 12 hours of treatment, mice were sacrificed and the GBM tissues were collected to be stored at -80 °C. Next, GBM tissue slices were prepared with a Leica Microsystems VT1200S. After blocking non-specific binding site in samples with 10% FCS, GBM sections were stained with HIF-1 α primary antibodies for 2 h. Then, samples were washed with 10% FCS for 3 times, and FITC labeled secondary antibodies (Molecular Probes-Life Technologies Inc.) (5 μ g/ml) were also added and stained for another 2 h. Samples were then washed with 10% FCS for 3 times. Bacteria were stained with 16S rRNA probe. Briefly, slides were hybridized with FISH probes 5' labeled with Cy7 at a concentration of 5 ng/ μ l in hybridization buffer (20 mM Tris-HCl, 0.1% sodium

dodecyl sulfate (SDS), 0.9% NaCl [pH 7.2]) and incubated at 56°C overnight. The following probes were used to detect general bacteria (EUB 338, GCT GCC TCC CGT AGG AGT). Nuclei were stained with 10 µM DAPI. Cover slides were mounted with Fluoromount-GTM medium (Southern Biotechnologies). Sections were observed with a wide-field fluorescence microscopy.

- **IR camera.** We used the IR camera to study the photothermal conversion efficiency of the constructed Trojan bacteria under NIR laser irradiation. Infrared thermal images were recorded by using a FLIR Ax5 camera under irradiation with an 808 nm laser at a power density of 1.2 W cm⁻², and the temperature was quantified by BM_IR software.

- **Statistic data.** Error bars represent the standard deviation obtained from three independent measurements. All the statistical analyses were performed using the Origin and GraphPad Prism 7 software. The statistical significance of differences was determined by a one-way ANOVA analysis. $p < 0.05$ (*), $p < 0.01$ (**), and $p < 0.001$ (***) were used to indicate statistical difference.

- **Luc-G422 cell.** The luc-G422 cell line was obtained from Shanghai Zhong Qiao Xin Zhou Biotechnology and cultured under appropriate conditions. The orthotopic GBM model was constructed by *in situ* inoculation of $\sim 8 \times 10^5$ Luc-G422 cells per mouse at day -7. The detailed procedures of the construction of GBM model by using Luc-G422 cells was given in the Response to your **Comment 22**.

Location of changes: Paragraph 5 in Page 21, Paragraph 1&3 in Page 22, Paragraph 2 in Page 23, Paragraph 1&2 in Page 24.

22. In line 695, authors described stereotactic injection of tumor cells. This experiment was done by image-guided surgery? How can you do this sophisticated surgery; 0.5 mm anterior, 2 mm left lateral from bregma, 3.1 mm deep?

Response: Accordingly, the details of stereotactic injection of tumor cells have been provided into the revised manuscript. Typically, the construction of the orthotopic GBM-bearing mouse model was according to the previous reports (refs. *Nat. Commun.* 5, 4196 (2014); *Nat. Cell Biol.* 17, 1556-1568 (2015)). As schematically illustrated in **Supplementary Fig. 20**, we used a brain stereotaxic apparatus to determine the location of glioma cells in the mouse brain, and injected luc-G422 cells to monitor the size of the mouse brain tumor by biofluorescence signals. The specific experimental operations were as follows:

- (1) Digest luc- G422 cells, dilute Matrigel in a 4-fold concentration gradient in PBS, and use the diluted resuspended cells to place on ice.
- (2) After the mouse was anesthetized, the head of the mouse was fixed on the brain stereotaxic apparatus. After sterilizing the brain skin with disinfectant, cut the skin with a scalpel, stop the bleeding with a cotton swab, and observe the position of the fontanelle.

- (3) Determine the coordinates of the fontanelle by the brain locator, and record the values. Afterwards, the position of the syringe was moved by the coordinates of the front 0.05 mm (+), the left 0.19 mm (+), and the depth 0.31 mm (+) to determine the injection site and mark it.
- (4) Use a micro drill to make a hole at the marked site, use a micro syringe to draw 5 μ L of luc-G422 cells, and slowly inject the cell suspension into the hole.
- (5) Take a small piece of hemostatic cotton to fill the hole in the brain, suture it, apply antibiotics, and wait for the mouse to wake up.

Supplementary Fig. 20 The construction of the orthotopic GBM-bearing mouse model by using the brain stereotaxic apparatus.

Location of changes: Supplementary Fig. 20, Paragraph 5 in Page 21, Paragraph 1 in Page 22.

Special thanks to Reviewer #3's comments again.

Reviewer #4 (Expertise: Nanoparticles and photothermal therapy)- Remarks to the Author:

The work reported “Trojan bacteria cross blood-brain barrier for glioblastoma photothermal immunotherapy”, the authors constructed Trojan bacteria as drug delivery vehicles for GBM therapy. Although the authors obtained a Trojan bacterial system that greatly enhanced the targeted delivery of GP-ICG-SiNPs to GBMs and synergistically promoted antitumor immune responses, there are still a number of experimental deficiencies that affect the interpretation of the results. Clarifying some points and adding additional data will improve considerably the study and give more support for the conclusions. Overall, the authors demonstrated a well-presented study and I recommend publication after addressing the below comments:

General response: We gratefully thank Reviewer #4 for his/her positive remarks. Accordingly, the manuscript has been thoroughly revised to fully address Reviewer’s concerns. The details are as follows,

1. The cytotoxicity of GP-ICG-SiNPs appears to be comparable with Trojan EC or Trojan VNP (figure 3e). So, is EC or VNP used as the carrier only because of its BBB targeting ability? However, it seems not so strong judging from the in vivo experimental results of the carrier alone (figure 4bcd).

Response: We agree with Reviewer #4’s comment that the cytotoxicity of GP-ICG-SiNPs is comparable with Trojan EC or Trojan VNP in figure 3e ascribed to their equivalent photothermal effects. Of note, EC or VNP is used as the carrier because of its BBB targeting ability and photothermal effects, which could lead to lysis of Trojan bacterial cells and the adjacent tumor cells, thus promoting anticancer immune responses.

2. Will the residual Trojan bacteria in the brain or major organs (especially liver) affect the long-term survival of GBM-bearing mice (figure 7)?

Response: The residual Trojan bacteria in the brain or major organs (especially liver) would not affect the long-term survival of GBM-bearing mice. As revealed in new **Fig. 7**, the residual Trojan bacteria could be totally eliminated from the body basically at 7 days after treatment.

Fig. 7. The elimination of residual bacteria after photothermal immunotherapy. The fluorescence distribution in the main organs (heart, liver, spleen, lung, kidney and brain) of GBM-bearing mice after 5 days (a) or 7 days (b) of photothermal

immunotherapy. The mice were intravenously injected with PBS, $\sim 1 \times 10^7$ CFU mCherry@EC (m@EC), $\sim 1 \times 10^7$ CFU mCherry@VNP (m@VNP), Trojan m@EC (e.g., GP-ICG-SiNPs (8 mg/kg ICG) internalized into $\sim 1 \times 10^7$ CFU m@EC) or Trojan m@VNP (e.g., GP-ICG-SiNPs (8 mg/kg ICG) internalized into 1×10^7 CFU m@VNP), respectively. At the 12-hour post-injection, the brains of those mice were suffered by an 808 nm irradiation (1.2 W/cm^2 , 5 min), followed by *ex vivo* imaging of the main organs after 5 days or 7 days of photothermal immunotherapy. The corresponding quantitative analysis of fluorescence intensity of main organs in different groups after 5 days (c) or 7 days (d) of photothermal immunotherapy (mean \pm SD, $n = 3$, $***P < 0.001$). Homogenates of major organs of GBM-bearing mice in different groups after 5 days (e) or 7 days (f) of photothermal immunotherapy cultured on the solid LB agar. **d**, Corresponding quantification of bacterial colonization on LB solid plates in different treatment groups after 5 days (g) or 7 days (h) of photothermal immunotherapy (mean \pm SD, $n = 3$, $***P < 0.001$). Statistical significance was calculated *via* one-way analysis of variance (ANOVA) with a Tukey post-hoc test.

Location of changes: Fig. 7, Paragraph 1 in Page 16.

3. In Figure 3g and h, the authors should calculate the synergy coefficient of different experimental groups to demonstrate the maturation of DCs by photothermal, EC and VNP.

Response: Following Reviewer's helpful suggestion, we have calculated the synergy coefficient of different experimental groups to demonstrate the maturation of DCs by Trojan bacteria + laser according to the previous papers (refs. *Macromol. Biosci.*, **13**, 1648-1660 (2013); *Clin. Cancer Res.*, **10**, 7994-8004 (2004)). Typically, the synergy coefficient was calculated to be ~ 0.75 for the combination of Trojan EC and laser; and ~ 0.78 for the combination of Trojan VNP and laser, indicating this combination exhibits synergy in the maturation of DCs.

Location of changes: Paragraph 1 in Page 9.

4. From Figure 4, we found that high doses of bacteria significantly reduced the body weight of mice, while medium and low doses did not. However, in addition to body weight, blood biochemistry, blood biochemistry and hematology data, the authors should provide pathological sections of major organs to further demonstrate their safety.

Response: Following Reviewer's helpful suggestion, we have provided pathological sections of major organs to further demonstrate their safety. As presented in **Supplementary Fig. 19**, no hydropic degeneration occurred in the heart tissues; no inflammatory infiltrates appeared in the liver tissues; no hyperplasia existed in the spleen tissues; no pulmonary fibrosis was found in the lung tissues; glomerula structures were easily identified in the kidney tissues. Together, no obvious histopathological abnormalities were found in biopsy sections in all resected organs, suggesting feeble *in vivo* toxicity of the Trojan bacteria.

Supplementary Fig. 19. H&E staining of histological evaluation of different organs (brain, heart, liver, spleen, lung and kidney) harvested from the GBM-bearing mice with different treatments. The mice were intravenously injected with PBS, GP-ICG-SiNPs (8 mg/kg ICG), $\sim 1 \times 10^7$ CFU EC, $\sim 1 \times 10^7$ CFU VNP, Trojan EC (GP-ICG-SiNPs (8 mg/kg ICG) internalized into $\sim 1 \times 10^7$ CFU EC) and Trojan VNP (GP-ICG-SiNPs (8 mg/kg ICG) internalized into 1×10^7 CFU VNP), respectively. At the 12-hour post-injection, the brains of those mice were suffered by an 808 nm irradiation (1.2 W/cm^2 , 5 min). At 30-day post-injection, the main organs were harvested for H&E staining. Scale bars, 50 μm .

Location of changes: Supplementary Fig. 19 and Paragraph 1 in page 16.

5. In Figure 6, there was little difference between the trojan EC and VNP in terms of heating curve, fluorescence signal or survival time. This needs to be explained and discussed.

Response: Accordingly, the related discussion has been added into the revised manuscript. Typically, the heating curve, fluorescence signal or survival time of Trojan bacteria was determined by the amount of the nanoagents taken by bacteria. Herein, we used the flow cytometry to analyze the uptake rate of GP-ICG-SiNPs by EC or VNP (**Fig. 2e**). As revealed in **Fig. 2e**, the uptake rates of GP-ICG-SiNPs by EC were close to the ones by VNP. This phenomenon might be attributed to the fact that both the EC and VNP are Gram-negative bacteria and have the similar morphology and size, possibly expressing the same number of ABC transporters. Thereby, there was little difference between the trojan EC and VNP in terms of heating curve, fluorescence signal or survival time.

Location of changes: Paragraph 1 in page 14.

Special thanks to Reviewer #4's comments again.

Finally, we thank you very much for the editor's and referees' valuable comments, which vastly facilitate improvement of the quality of this manuscript, making it possible to satisfy requirement of the esteemed journal--- Nature Communications. Thank you very much!

REVIEWERS' COMMENTS

Reviewer #1 (Remarks to the Author):

The authors have performed a very careful revision of their manuscript and provided detailed responses to my comments. I agree with their explanations to address points #4.4, 5.1, and 5.2. Congratulation on this very nice study that deserves publication in Nature Communications.

Reviewer #2 (Remarks to the Author):

My concerns have been addressed appropriately by the authors. The manuscript could be accepted for publication now.

Reviewer #3 (Remarks to the Author):

The manuscript entitled "Trojan bacteria cross blood-brain barrier for glioblastoma photothermal immunotherapy" employed bacteria to enhance photothermal effects leading to lysis of Trojan bacterial cells and the adjacent tumor cells which promote anticancer immune responses. This revised draft satisfied most of the points I came up with, so I suggest the publication of this manuscript on this journal.

Just one point I'd like to suggest is that this draft still has some English errors. Nouns are used like verbs. This draft should undergo linguistic revision.

Reviewer #4 (Remarks to the Author):

After this round of revision, the quality of the article has been improved, and I recommend it for publication.

Response to Reviewers' comments

Reviewer #1 (Remarks to the Author):

The authors have performed a very careful revision of their manuscript and provided detailed responses to my comments. I agree with their explanations to address points #4.4, 5.1, and 5.2. Congratulation on this very nice study that deserves publication in Nature Communications.

Response: Thank you very much for the reviewer's positive recommendation.

Reviewer #2 (Remarks to the Author):

My concerns have been addressed appropriately by the authors. The manuscript could be accepted for publication now.

Response: Thank you very much for the reviewer's positive recommendation.

Reviewer #3 (Remarks to the Author):

The manuscript entitled "Trojan bacteria cross blood-brain barrier for glioblastoma photothermal immunotherapy" employed bacteria to enhance photothermal effects leading to lysis of Trojan bacterial cells and the adjacent tumor cells which promote anticancer immune responses. This revised draft satisfied most of the points I came up with, so I suggest the publication of this manuscript on this journal.

Just one point I'd like to suggest is that this draft still has some English errors. Nouns are used like verbs. This draft should undergo linguistic revision.

Response: We gratefully thank the reviewer for his/her positive remarks. Following the reviewer's helpful suggestion, the draft has been undergone linguistic revision.

Reviewer #4 (Remarks to the Author):

After this round of revision, the quality of the article has been improved, and I recommend it for publication.

Response: Thank you very much for the reviewer's positive recommendation.

Finally, we thank you very much for the editor's and all referees' valuable comments again!